# *miR-125-chinmo* pathway regulates dietary restriction-dependent enhancement of lifespan in *Drosophila*

Manish Pandey[1], Sakshi Bansal[1], Sudipta Bar[2], Amit Kumar Yadav[3], Nicholas S Sokol[4], Jason M Tennessen[4], Pankaj Kapahi[2]*, Geetanjali Chawla[1]*

[1]RNA Biology Laboratory, Regional Centre for Biotechnology, Faridabad, India; [2]Buck Institute for Research on Aging, Novato, United States; [3]Translational Health Science and Technology Institute, Faridabad, India; [4]Department of Biology, Indiana University, Bloomington, United States

**Abstract** Dietary restriction (DR) extends healthy lifespan in diverse species. Age and nutrient-related changes in the abundance of microRNAs (miRNAs) and their processing factors have been linked to organismal longevity. However, the mechanisms by which they modulate lifespan and the tissue-specific role of miRNA-mediated networks in DR-dependent enhancement of lifespan remains largely unexplored. We show that two neuronally enriched and highly conserved microRNAs, *miR-125* and *let-7* mediate the DR response in *Drosophila melanogaster*. Functional characterization of *miR-125* demonstrates its role in neurons while its target *chinmo* acts both in neurons and the fat body to modulate fat metabolism and longevity. Proteomic analysis revealed that Chinmo exerts its DR effects by regulating the expression of *FATP, CG2017, CG9577, CG17554, CG5009, CG8778, CG9527*, and *FASN1*. Our findings identify *miR-125* as a conserved effector of the DR pathway and open the avenue for this small RNA molecule and its downstream effectors to be considered as potential drug candidates for the treatment of late-onset diseases and biomarkers for healthy aging in humans.

**\*For correspondence:**
Pkapahi@buckinstitute.org (PK);
gchawla@rcb.res.in (GC)

**Competing interests:** The authors declare that no competing interests exist.

## Introduction

Aging is characterized by a progressive decline in physiological function, which leads to an increased risk of chronic degenerative diseases and disabilities (*Harman, 2003*). Deregulated nutrient signaling is one of the key hallmarks of aging, and restricting nutrient intake or dietary restriction (DR) has been shown to enhance health and longevity in most species (*Fontana and Partridge, 2015*; *Kapahi et al., 2017*; *Klass, 1977*; *Lin et al., 2002*; *López-Otín et al., 2013*; *McCay et al., 1989*). More significantly, DR delays age-related pathologies such as diabetes, cardiovascular diseases, cancer, and neurodegenerative disorders (*Anson et al., 2003*; *Klebanov, 2007*; *Kraus et al., 2019*; *Maswood et al., 2004*; *Mattison et al., 2012*; *Mattson and Wan, 2005*; *Wang et al., 2005*). Emerging evidence from diverse model systems has implicated micro-ribonucleic acids (miRNAs) as critical components of signaling pathways that modulate lifespan by regulating mRNA turnover and translation (*Boehm and Slack, 2005*; *Chawla et al., 2016*; *Liu et al., 2012*; *Verma et al., 2015*). These 19–24 nucleotides long, single-stranded RNAs function by directing effector complexes to target mRNAs (*Bartel, 2018*). This recruitment of the miRNA ribonucleoprotein complexes (miRNPs) is facilitated by interactions between the miRNA and its target and results in silencing of the target mRNA (*Guo et al., 2010*; *Kim, 2005*). Since the interaction of a miRNA and its target occurs by imperfect base-pairing interactions, a single miRNA can target several mRNAs in a given context. Thus, these evolutionary conserved and dosage-sensitive effectors possess the key attributes to facilitate the complex metabolic reprogramming that occurs during DR. While studies in the *C. elegans*,

mammalian cell culture, mouse, and primate model systems have reported regulation of miRNAs and their targets upon DR, there is no evidence to indicate whether the DR-mediated expression changes of the miRNAs and their downstream targets occur in the same tissue and whether modulating miRNA and their target mRNAs can result in lifespan extension by DR (*Mercken et al., 2013*; *Pandit et al., 2014*; *Schneider et al., 2017*; *Zhang et al., 2019*).

Our small RNAseq analysis of wild-type flies that were fed an AL and DR diet for 40 days identified *let-7* and *miR-125* to be upregulated by DR (to be published elsewhere). Here, we report that nutrient restriction in *D. melanogaster* upregulates *let-7-Complex* miRNAs (*miR-100*, *let-7*, and *miR-125*). Furthermore, *let-7* and *miR-125* loss-of-function mutations dampen the DR-dependent lifespan extension. The DR phenotype associated with loss of *miR-125* is due to the derepression of its target, <u>*Ch*ronologically *I*nappropriate *M*orphogenesis</u> (*chinmo*). Our analysis reveals that *chinmo* codes for a nutrient-regulated transcription factor and its upregulation in the nervous system results in altered fat metabolism. Our analysis has also uncovered a previously unknown mechanism of nutrient-dependent post-translational control of Chinmo that may be linked to a novel nutrient-dependent non-nuclear role for this protein. Consistent with the *miR-125* loss of function DR phenotype, increasing the dosage of human *miR-125* in the fat body increased longevity. In summary, we have identified a conserved miRNA that mediates the effects of DR by promoting tissue-tissue communication demonstrating its potential as a DR mimetic agent.

## Results

### DR-dependent upregulation of *let-7* and *miR-125* increases lifespan

To examine whether *let-7-Complex* miRNAs mediate the effects of DR, we investigated whether DR affects the expression of these miRNAs in wild type ($w^{1118}$) and *let-7-C* hypomorphic (*let-7-C$^{hyp}$*) mutant. The *let-7-C* hypomorph (*let-7-C$^{hyp}$*/Δ*let-7-C*: $w^{1118}$; *let-7-C$^{GKI}$*/*let-7-C$^{KO2}$*, P{neoFRT}40A; P{w+, *let-7-Cp$^{3.3kb}$::cDNA*} / {v+, *let-7-C $^{Δlet-7-C miRNAs}$*}attP2) encodes the *let-7-Complex* cDNA driven by a 3.3 kb promoter fragment and a single copy of the ~18 kb *let-7-C* that lacks the mature *miR-100*, *let-7* and *miR-125* sequences in a transheterozygous *let-7-C$^{null}$* mutant background (*let-7-C$^{KO2}$*/*let-7-C$^{GKI}$*) (*Chawla et al., 2016*; *Chawla and Sokol, 2012*; *Sokol et al., 2008*; *Figure 1A*). The 3.3 kb enhancer fragment harbors three ecdysone response elements (EcREs) that we previously showed to be responsible for the developmental expression of the *let-7-C* primary transcript (*Chawla and Sokol, 2012*; *Figure 1A*). The *let-7-C$^{hyp}$*/Rescue ($w^{1118}$; *let-7-C$^{GKI}$*/*let-7-C$^{KO2}$*, P{neoFRT}40A; P{w+, *let-7-Cp$^{3.3kb}$::cDNA*} / {v+, *let-7-C*}attP2) line encodes a single copy of the *let-7-C$^{hyp}$* and a single copy of the *let-7-C* wild-type transgene (*Figure 1A*). The *let-7-C$^{hyp}$*/Δ*let-7-C* and *let-7-C$^{hyp}$*/Rescue strains were generated by utilizing the crossing scheme represented in the supplement figure (*Figure 1—figure supplement 1*). Wild type ($w^{1118}$), *let-7-C$^{hyp}$*/Δ*let-7-C*, and the *let-C$^{hyp}$* rescue flies were exposed to DR or AL conditions for 7 or 20 days, and quantitative reverse transcription polymerase chain reaction (qRT-PCR) analysis was performed with RNA extracted from whole animals. In young $w^{1118}$ flies exposed to DR for 7 days, an increase in only *miR-125* levels (176 ± 50%) was observed. Increasing the dietary restriction for 20 days led to a statistically significant increase in the levels of *miR-100* (223 ± 80%), *let-7* (171 ± 22%), and *miR-125* (255 ± 54%) in $w^{1118}$ flies (*Figure 1B–D*). Under AL conditions, young *let-7-C$^{hyp}$*/Δ*let-7-C* mutants displayed lower levels of *miR-100* (*let-7-C$^{hyp}$ Rescue*: AL = 100% and DR = 83.8 ± 19%; *let-7-C$^{hyp}$*: AL = 24 ± 4% and DR = 30 ± 8%), *let-7* (*let-7-C$^{hyp}$ Rescue*: AL = 100% and DR = 110 ± 28.5%; *let-7-C$^{hyp}$*: AL = 51 ± 7% and DR = 74 ± 11.5%) and *miR-125* (*let-7-C$^{hyp}$ Rescue*: AL = 100% and DR = 145.8 ± 32%; *let-7-C$^{hyp}$*: AL = 34 ± 3.4% and DR = 39 ± 20%) relative to the *let-C$^{hyp}$* rescue strain in AL and DR conditions (*Figure 1E–G*). Increasing the DR for 20 days led to an increase in the levels of *let-7* (144 ± 32.5%) and *miR-125* (178 ± 28%) in the *let-7-C$^{hyp}$ rescue* line. However, there was a significant decrease in the levels of *miR-100* (AL = 5.7 ± 3.8% and DR = 3.6 ± 1.6%), *let-7* (AL = 5.6 ± 2.9% and DR = 4.4 ± 3.6%) and *miR-125* (AL = 3.3 ± 2.1% and DR = 4.5 ± 2.7%) in the *let-7-C$^{hyp}$* mutants that were administered either AL or DR diet (*Figure 1—figure supplement 2B–D*). To examine whether the increase in *let-7-C* miRNAs was required for lifespan extension upon DR, we examined the survival of $w^{1118}$, *let-7-C$^{hyp}$*/Rescue, and *let-7-C$^{hyp}$*/Δ*let-7-C* mutants fed an AL and DR diet (*Figure 1H–I*) (*Figure 1—figure supplement 2A*). Wild-type ($w^{1118}$) flies fed a DR diet (blue line) lived significantly longer than wild-type flies that were fed an AL diet (red line) (p=0.00E + 00; $X^2$ = 156)

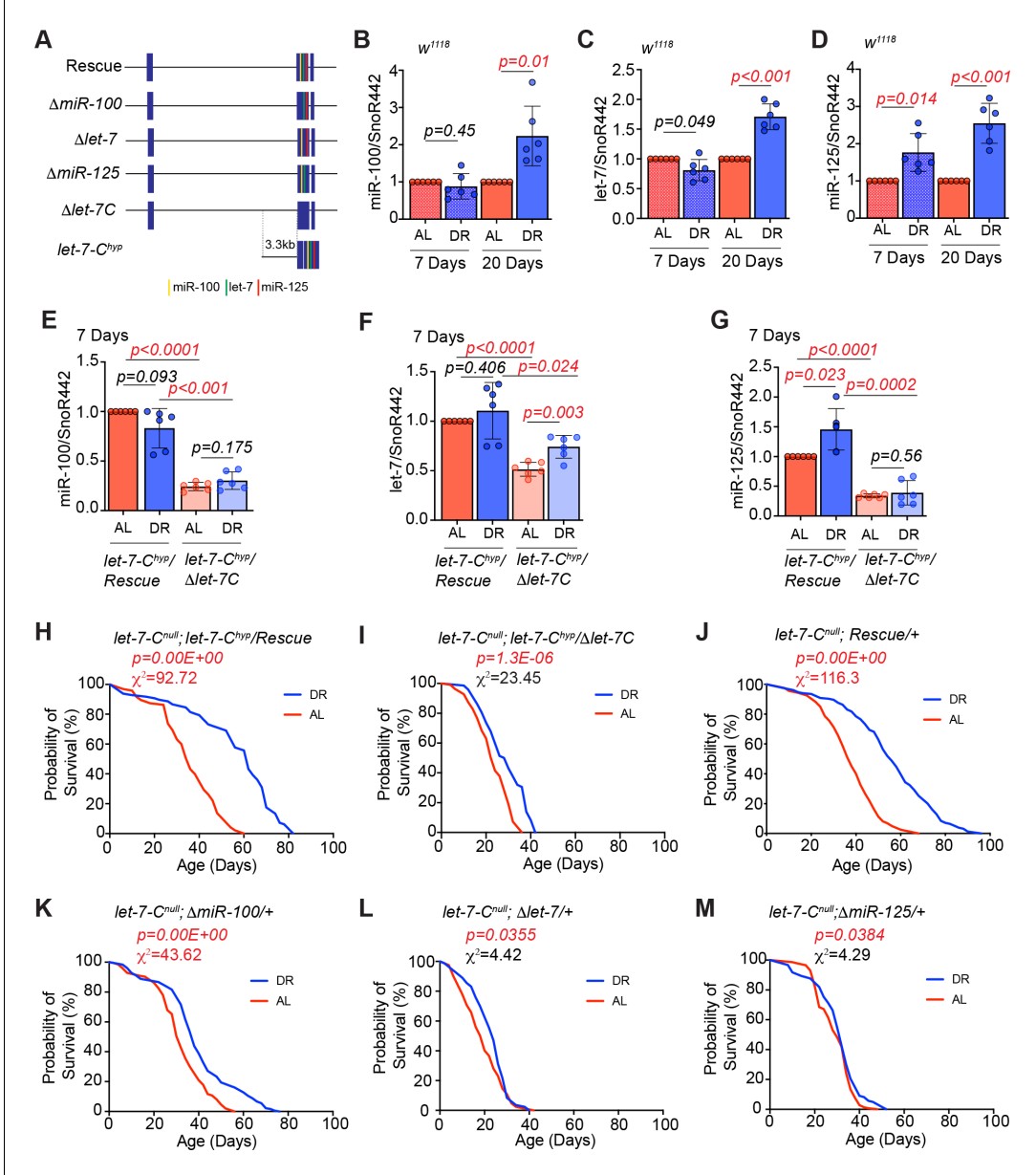

**Figure 1.** *let-7* and *miR-125* are required for DR-dependent enhancement of lifespan. (**A**) Schematic of the *let-7-Complex* (*let-7-C*) rescuing transgenes. The *Drosophila let-7-Complex* (*let-7-C*) locus is located on chromosome two and encodes a 2435 nucleotide long primary transcript that is processed into three evolutionary conserved miRNAs: *miR-100*, *let-7* and *miR-125*. The *let-7-C* rescuing transgene includes a 17983 base-pair genomic fragment containing the *let-7-C* locus. The Δ*miR-100*, Δ*let-7*, and Δ*miR-125* are derivatives of the *let-7-C* transgenes that lack the mature *miR-100*, *let-7*, and *miR-125* sequences, respectively. The *let-7-C* hypomorph (*let-7-C*^*hyp*) encodes the *let-7-Complex* cDNA driven by a 3.3 kb promoter fragment in a transheterozygous *let-7-C* null mutant background (*let-7-C*^*KO2*/*let-7-C*^*GKI*). The *let-7-C*^*hyp*/*Rescue* line encodes a single copy of the *let-7-C*^*hyp* and a single copy of the *let-7-C* wild-type transgene. The *let-7-C*^*hyp*/Δ*let-7-C* line encodes a single copy of the *let-7-C*^*hyp* and a single copy of the ~18 kb *let-7-C* that lacks the mature *miR-100*, *let-7* and *miR-125* sequences. The *let-7-C*^*null* rescue line encodes a single copy of the ~18 kb *let-7-C* transgene in a *let-7-C* null mutant. (**B-D**) Quantitative RT-PCR of *miR-100* (**B**), *let-7* (**C**) and *miR-125* (**D**) in wild-type (*w*^*1118*) flies that were fed *Ad libitum* (AL) (pink pattern and pink solid bar) or DR diet (blue pattern and blue solid bar) for 7 days or 20 days. The levels of *miR-100*, *let-7* and *miR-125* increase upon 20d exposure to DR, while a significant increase in *miR-125* is detected upon 7d of DR. Expression levels were normalized to *SnoR442.* Values are mean ± SD, n = 6. (**E-G**) *Let-7-C* ^*hyp*/Δ*let-7-C* flies express significantly reduced levels of *miR-100*, *let-7*, and *miR-125* as compared to *Let-7-C* ^*hyp* rescue flies that were fed AL or DR diets for 7 days. For expression analysis of the rescue and mutant lines that were fed AL or DR diet for 20 days, please refer to **Figure 1—figure supplement 2B-D**. (**H**) The *let-7-C*^*hyp*/*Rescue* flies show a significant increase in lifespan upon dietary restriction ( blue line; n = 97) as compared to *let-7-C*^*hyp*/*Rescue* flies that were fed an '*ad libitum*' diet ( red line; n = 95). (**I**) The *let-7-C*^*hyp* mutants (AL, red line; n = 111 and DR, blue line; n = 72) display a reduced DR-dependent increase in lifespan compared to the rescue line. (**J-M**) Compared to the *let-7-C*^*null* rescue flies (**J**) and Δ*miR-100* (**K**) flies, Δ*let-7* (**L**) and Δ*miR-125* (**M**) flies displayed a significantly dampened DR-dependent lifespan extension upon DR (blue line) when

*Figure 1 continued on next page*

Figure 1 continued

compared to flies that were fed an AL diet (red line). For statistical comparison of survival curves, p values and $X^2$ values were calculated with log rank test. Genotypes of strains used in this figure (B-D) $w^{1118}$; (E-G, H) *let-7-C$^{null}$*; *let-7-C$^{hyp}$/Rescue: w$^{1118}$*; *let-7-C$^{GKI}$/let-7-C$^{KO2}$, P{neoFRT}40A; P{w+, let-7-Cp$^{3.3kb}$::cDNA} / {v+, let-7-C}attP2*; (E-G, I) *let-7-C$^{hyp}$//Δlet-7-C: w$^{1118}$*; *let-7-C$^{GKI}$/let-7-C$^{KO2}$, P{neoFRT}40A; P{w+, let-7-Cp$^{3.3kb}$::cDNA} / {v+, let-7-C $^{Δlet-7-C miRNAs}$}attP2*; (J) *let-7-C$^{null}$*; *Rescue/+: w$^{1118}$*; *let-7-C$^{GKI}$/let-7-C$^{KO2}$, P{neoFRT}40A; {v+, let-7-C} attP2 /+*; (K) *let-7-C$^{null}$*; *ΔmiR-100/+: w$^{1118}$*; *let-7-C$^{GKI}$/let-7-C$^{KO2}$, P{neoFRT}40A; {v+, let-7-C ΔmiR-100} attP2 / +*; (L) *let-7-C$^{null}$*; *Δlet-7: w$^{1118}$*; *let-7-C$^{GKI}$/let-7-C$^{KO2}$, P{neoFRT}40A; {v+, let-7-C $^{Δlet-7}$} attP2 / +*; (M) *let-7-C$^{null}$*; *ΔmiR-125: w$^{1118}$*; *let-7-C$^{GKI}$/let-7-C$^{KO2}$, P{neoFRT}40A; {v+, let-7-C $^{ΔmiR-125}$} attP2 / +*.

The online version of this article includes the following source data and figure supplement(s) for figure 1:

**Source data 1.** Lifespan analysis of *let-7-Complex$^{hyp}$, let-7-Complex$^{hyp}$ rescue, let-7-Complex$^{null}$ rescue, ΔmiR-100, Δlet-7,* and *ΔmiR-125* mutant lines.

**Source data 2.** Survival proportions for strains used in *Figure 1*.

**Figure supplement 1.** Scheme for generation of experimental strains used in *Figures 1* and *2*.

**Figure supplement 2.** Lifespan analysis controls and expression analysis of strains used in *Figure 1*.

**Figure supplement 2—source data 1.** Lifespan analysis of *let-7-Complex$^{hyp}$, let-7-Complex$^{hyp}$ rescue, let-7-Complex$^{null}$ rescue, ΔmiR-100, Δlet-7,* and *ΔmiR-125* mutant lines.

**Figure supplement 2—source data 2.** Survival proportions for strains used in *Figure 1*.

(*Figure 1—figure supplement 2A*). DR increased median lifespan by 50% in $w^{1118}$ flies, by 42% in *let-7-C$^{hyp}$ Rescue* flies (p=0.00E + 00; $X^2$ = 92.72) and by 17% in the *let-7-C$^{hyp}$* mutant flies (p=1.30E-06; $X^2$ = 23.45) (*Figure 1, Figure 1—figure supplement 2A* and *Figure 1—source data 1*). To confirm whether the DR phenotype was not due to the genetic background of the fly stocks used to prepare the experimental strains, the survival of the background strains (*let-7-C$^{GKI}$/+, let-7-C$^{KO2}$/+, let-7-C$^{KO2}$/+; let-7-C$^{hyp}$/+* and *let-7-C$^{GKI}$/+; let-7-C/+*) was measured under AL and DR conditions (*Figure 1—figure supplement 2E–H* and *Figure 1—figure supplement 2—source data 1*). Taken together, these data indicated that one or more of the *let-7-C* miRNAs were required for DR-mediated lifespan extension and a reduction in the levels of one or more of the miRNAs caused dampening of the DR-dependent increase in longevity.

To determine the specific contribution of *miR-100, let-7,* and *miR-125* in DR-mediated enhancement of lifespan, we measured survival of genetically identical control and *miR-100, let-7,* and *miR-125* null mutant flies reared on AL or DR diet (*Figure 1J–M*). The rescue and the mutant lines were generated by expressing a single copy of the wild type, Δ*miR-100,* Δ*let-7,* or Δ*miR-125* transgene in a transheterozygous *let-7-C$^{null}$* mutant background (*let-7-C$^{KO2}$/let-7-C$^{GKI}$*). The *let-7-C* rescuing (wild type) transgene encodes a 17983 base-pair genomic fragment containing the *let-7-C* locus. The Δ*miR-100,* Δ*let-7* and Δ*miR-125* are derivatives of the *let-7-C* transgenes that lack the mature *miR-100, let-7* and *miR-125* sequences, respectively. The genetic background of the control and mutant flies that were compared were identical and the crossing scheme used for the generation of these strains is described in our previous publication and the supplement figure (*Figure 1—figure supplement 1; Chawla et al., 2016*). *let-7-C$^{null}$* Rescue (p=0.00E + 00; $X^2$ = 116.3) and Δ*miR-100* (p=0.00E + 00; $X^2$ = 43.62) female flies fed a DR diet (blue line) lived significantly (*let-7-C$^{null}$* Rescue DR: 47% increase median lifespan; Δ*miR-100* DR: 18.75% increase in median lifespan) longer than flies that were fed an AL diet (*Figure 1J,K* and *Figure 1—source data 1*). Δ*let-7* mutants exhibited a significantly dampened lifespan extension when fed a DR diet (p=0.0355; $X^2$ = 4.42). Though a 33% increase in median lifespan was observed, the DR-fed flies had a 4.7% decrease in maximum lifespan compared to the AL-fed flies (*Figure 1L* and *Figure 1—source data 1*). In contrast, Δ*miR-125* mutants failed to exhibit lifespan extension when fed a DR diet, and a 0% increase in median lifespan was observed for Δ*miR-125* flies (p=0.0384; $X^2$ = 4.29) that were fed a DR diet (*Figure 1M* and *Figure 1—source data 1*). To confirm that the reduction in DR-dependent lifespan extension was specifically due to loss of *let-7* and *miR-125* and not an effect of the genetic background of the strains used, the survival of the background strains (*let-7-C$^{GKI}$/+, let-7-C$^{KO2}$/+, let-7-C$^{GKI}$/+; let-7-C/+, let-7-C$^{GKI}$/+; ΔmiR-100, let-7-C$^{GKI}$/+; Δlet-7, let-7-C$^{GKI}$/+; ΔmiR-125, let-7-C$^{GKI}$/+; Δlet-7-C*) was measured under AL and DR conditions (*Figure 1—figure supplement 2E–L* and *Figure 1—figure supplement 2—source data 1*). A significant DR-dependent increase in lifespan was observed for all the lines tested. These data confirmed the requirement of *miR-125* and *let-7* in DR-dependent extension of lifespan.

## *miR-125* regulates DR by repressing *chinmo*

MicroRNAs function by repressing their target mRNAs. We had previously shown that two of the *let-7-Complex* miRNAs, *miR-125* and *let-7* differentially target the mRNA of a BTB-zinc finger protein-coding gene referred to as C̲hronologically I̲nappropriate M̲orphogenesis (*chinmo*) (*Chawla et al., 2016*; *Zhu et al., 2006*). The 3′ untranslated region of *chinmo* mRNA has several *let-7* and *miR-125*-binding sites (*Wu et al., 2012*). The *let-7-C* miRNAs are not detected during early development but their upregulation during larval-to-pupal transition is mediated by transcriptional activation of the *let-7-C* primary transcript by the Ecdysone receptor and coincides with the downregulation of Chinmo (*Chawla and Sokol, 2012*; *Wu et al., 2012*). *Let-7-C* null mutants display defects in temporal transitions in mushroom body neuron identities due to elevated Chinmo levels (*Wu et al., 2012*). Our previously published functional and expression analysis of *let-7* and *miR-125* mutants showed that *chinmo* is differentially regulated by these two *let-7-C* miRNAs: with *let-7* being the predominant repressor of *chinmo* during metamorphosis and *miR-125* silencing *chinmo* in adulthood (*Figure 2A*; *Chawla et al., 2016*). Consistent with our previously published data, Chinmo is upregulated in the dissected adult brains of *miR-125* null mutants that are fed an AL or DR diet for 5 days (*Figure 2B*, upper panels). To test whether the DR-dependent phenotypes of the *miR-125* loss of function mutant were due to lack of post-transcriptional silencing of its previously validated target *chinmo*, we measured survival of strains in which dosage of *chinmo* was reduced genetically by a *let-7-C* Gal4 (*let-7-C^GKI*) driven UAS *chinmo^RNAi* transgene or a *chinmo^1* loss-of-function mutant in the *let-7-C* rescue flies and *miR-125* mutants (*Figure 2C–F* and *Figure 2—source data 1*). Lowering *chinmo* levels suppressed the DR-dependent lifespan phenotypes of *miR-125* mutants (Compare *Figure 1M* with *Figure 2D and F*). Although, reducing *chinmo* specifically in *let-7-C* expressing (*let-7-C Gal4*) cells was able to increase the median lifespan by 25% (compared to 0% in *miR-125* mutants), a 41.3% increase in median lifespan was observed when *chinmo* levels were reduced genetically (*chinmo^1*) in all *chinmo* expressing cells (*Figure 2F* and *Figure 2—source data 2*). The *let-7-C Gal4* driver alone control flies displayed a statistically significant DR-dependent increase in lifespan (*Figure 1—figure supplement 2F*). The UAS *chinmo^RNAi* control with and without RU-486 indicated that RU-486 had no significant effect on lifespan and *let-7-C^KO2*, *chinmo^1* control crosses displayed a DR-dependent increase in lifespan (*Figure 2—figure supplement 1A–B*, *Figure 2—figure supplement 1—source data 1–2*). These data demonstrated a role for *chinmo* in DR-dependent lifespan extension by *miR-125*. Furthermore, these data indicated that *chinmo* played a wider role in regulating DR-dependent lifespan extension which is not just limited to *miR-125* expressing cells.

DR imposed by restricting yeast in the diet enhances lipid content (*Bradley and Simmons, 1997*; *Katewa et al., 2012*) and this increase in lipid content enhances lipid turnover under DR and is required for the DR-dependent lifespan extension in *Drosophila* (*Katewa et al., 2012*). Since *miR-125* mutants displayed a DR phenotype, we quantitated the levels of total triglycerides (TAG) in whole bodies of rescue and *miR-125* mutants that were exposed to AL or DR diet for 20 days. In contrast to the rescue strain, the *miR-125* mutants displayed a significant drop in TAG levels in both AL (83 ± 8.5%) and DR (40.5 ± 21.6%) conditions (*Figure 2G*). In a parallel experiment, we also examined lipid droplets in the fat body of adult flies by staining with nile red. Consistent with the TAG analysis, the *miR-125* mutants displayed a significant drop in the diameter of lipid droplets compared to the rescue strain in both AL (35% decrease in the median) and DR (59% decrease in median) diet (*Figure 2H–I*). Thus, our analysis demonstrates that *miR-125* regulates the shift in lipid metabolism upon DR. Consistent with the lifespan data, reducing *chinmo* dosage in Δ*miR-125* flies resulted in an increase in the TAG levels and lipid droplet diameter (*Figure 2G–I*). However, a much greater increase in TAG levels was observed in *let-7-C Gal4> ΔmiR-125, UAS chinmo^RNAi* (113.3 ± 8.7% in AL and 446 ± 86.9% increase in DR relative to Δ*miR-125*) flies as compared to the Δ*miR-125*, *chinmo^1* flies (108.3 ± 6.6% in AL and 214.8 ± 90.96% increase in DR) (*Figure 2G*). However, while RNAi for *chinmo* in *let-7-C Gal4> ΔmiR-125, UAS chinmo^RNAi* flies doubled the TAG levels that were observed in the Rescue line under DR conditions (93.65 ± 7.2% in AL and 180.7 ± 32.5.9% increase in DR relative to Rescue flies), genetically reducing *chinmo* levels in Δ*miR-125*, *chinmo^1* restored the TAG to the control/rescue levels in DR conditions (89.5 ± 5.45% in AL and 87.04 ± 32.9% increase in DR relative to Rescue flies). These differences could have arisen due to other unknown *miR-125* targets operating in the DR pathway or due to *mir-125*-independent regulation of *chinmo* upon DR in cells that do not express *miR-125* (*Figure 2G*). Reducing *chinmo* levels in *miR-125* mutant flies that

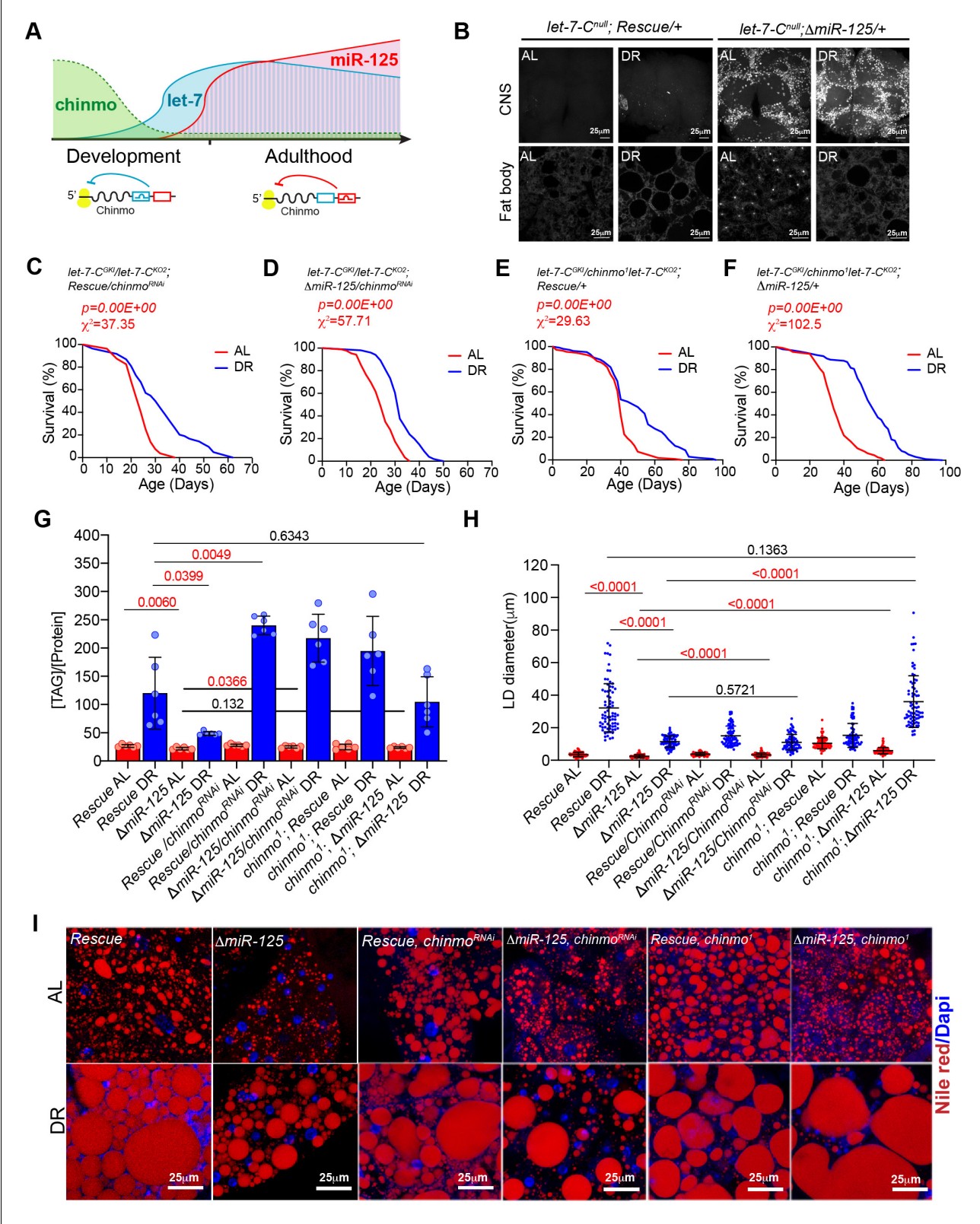

**Figure 2.** Reducing *chinmo* (*ΔmiR-125, chinmo^RNAi^* and *ΔmiR-125, chinmo^1^*) levels in *miR-125* mutants suppresses the loss of DR-mediated increase in lifespan extension and decrease in triglyceride levels. (**A**) Schematic representing post-transcriptional repression of *chinmo* mRNA by *let-7* and *miR-125* adapted from our previously published study (*Chawla et al., 2016*). *chinmo* mRNA 3'untranslated region (3'UTR) encodes multiple binding sites for *let-7* and *miR-125* miRNAs and is differentially regulated by *let-7* and *miR-125*. *Let-7* is the predominant repressor of *chinmo* during development and *miR-*

*Figure 2 continued*

*125* silences *chinmo* in the adult stage. **(B)** Loss of *miR-125* leads to derepression of *chinmo* in the adult brain and a concomitant increase in Chinmo signal in the adult fat tissue (Compare top two left panels with top two right panels). Confocal images of dissected tissues (brain and fat body) from adult *let-7-C* $^{null}$ rescue and *miR-125* mutant flies that were fed *ad libitum* (AL) or nutrient restricted (DR) diet for 5 days. **(C, D)** Rescue, UAS *chinmo*$^{RNAi}$ flies and Δ*miR-125*, UAS *chinmo*$^{RNAi}$ flies show a 25% increase in median lifespan upon DR ( blue line) (Rescue, UAS *chinmo*$^{RNAi}$, $\chi^2$ = 37.35; Δ*miR-125*, UAS *chinmo*$^{RNAi}$, $\chi^2$ = 57.71). **(E-F)** *Rescue, chinmo*$^1$ flies display a 20% increase in median lifespan upon DR (blue line) and Δ*miR-125, chinmo*$^1$ flies display a 41% increase in DR-dependent increase in median lifespan (compare red and blue curves) (*Rescue, chinmo*$^1$ $\chi^2$ = 29.63; Δ*miR-125, chinmo*$^1$ $\chi^2$ = 102.5). For statistical comparison of survival curves, p values and $\chi^2$ were calculated with log rank test. **(G)** Quantitation of triglyceride (TAG) stored levels in Rescue line, Δ*miR-125* line, Rescue, UAS *chinmo*$^{RNAi}$ line, Δ*miR-125*, UAS *chinmo*$^{RNAi}$ line, *Rescue, chinmo*$^1$ line and Δ*miR-125, chinmo*$^1$ line, AL (Red bars) and DR (Blue bars) fed 20-day-old flies. The bars represent mean ± SD, n = 6, p value was calculated with unpaired t-test with Welch's correction. p Values are indicated in the graph. The TAG values were normalized to protein levels. The Δ*miR-125* flies display significantly lower stored triglycerides compared to the genetically identical rescue line in both AL and DR diets. Reducing *chinmo* levels by RNAi or by reducing a copy of *chinmo* led to an increase in TAG levels upon DR. Consistent with the survival analysis, the increase in TAG levels in Δ*miR-125, chinmo*$^1$ flies TAG was similar to the increase in rescue flies (p value=0.6343), indicating that reducing a copy of *chinmo* was sufficient to rescue the TAG levels and the rescue was better than what was obtained with knockdown of *chinmo* by RNAi in cells that express *miR-125*. **(H, I)** Fat bodies/abdomens of female flies were dissected and stained for the content and diameter of lipid droplet (LD) (red are lipid droplets stained with Nile red and blue is Dapi). Scale bar, 25 µm. **(I)** Quantitation of lipid droplet (LD) diameter in **(H)**. Quantitation of 15 largest LDs in five samples per condition. Error bars represent mean ± SD and p values are indicated in the figure. The reduction in lipid droplet size in Δ*miR-125* mutants that are fed an AL and DR diet is rescued by reducing the dosage of *chinmo* in Δ*miR-125, chinmo*$^1$ and not by *chinmo*$^{RNAi}$. Genotypes of strains used in this figure: **(B, G-I)** *let-7-C*$^{null}$; *Rescue/+*: $w^{1118}$; *let-7-C*$^{GKI}$/*let-7-C*$^{KO2}$, P{neoFRT}40A; {v+, let-7-C} attP2 /+; **(B, G-I)** *let-7-C*$^{null}$; Δ*miR-125*: $w^{1118}$; *let-7-C*$^{GKI}$/*let-7-C*$^{KO2}$, P{neoFRT}40A; {v+, let-7-C $^{ΔmiR-125}$} attP2 / +; **(C, G-I)** *let-C*$^{GKI/KO2}$; *Rescue, chinmo*$^{RNAi}$: $w^{1118}$; *let-7-C*$^{GKI}$/*let-7-C*$^{KO2}$, P{neoFRT}40A; {v+, let-7-C} attP2/P{w+, UAS-chinmo$^{RNAi\ 148}$}VK00033; **(D, G-I)** *let-C*$^{GKI/KO2}$; Δ*miR-125, chinmo*$^{RNAi}$: $w^{1118}$; *let-7-C*$^{GKI}$/*let-7-C*$^{KO2}$, P{neoFRT}40A; {v+, let-7-C $^{ΔmiR-125}$} attP2/P{w+, UAS-chinmo$^{RNAi\ 148}$}VK00033; **(E, G-I)** *let-7-C*$^{GKI}$/*chinmo*$^1$*let-7-C*$^{KO2}$; *Rescue/+*: $w^{1118}$; *let-7-C*$^{GKI}$/*chinmo*$^1$, *let-7-C*$^{KO2}$, P{neoFRT}40A; {v+, let-7-C} attP2 /+; **(F, G-I)** *let-7-C*$^{GKI}$/*chinmo*$^1$*let-7-C*$^{KO2}$; Δ*miR-125/+*: $w^{1118}$; *let-7-C*$^{GKI}$/*chinmo*$^1$, *let-7-C*$^{KO2}$, P{neoFRT}40A; {v+, let-7-C $^{ΔmiR-125}$} attP2 / +.

The online version of this article includes the following source data and figure supplement(s) for figure 2:

**Source data 1.** Lifespan analysis of *rescue, chinmo*$^{RNAi}$ and Δ*miR-125, chinmo*$^{RNAi}$ strains.

**Source data 2.** Lifespan analysis of *rescue, chinmo*$^1$ and Δ*miR-125, chinmo*$^1$ strains.

**Source data 3.** Survival proportions for strains used in *Figure 2*.

**Figure supplement 1.** Effect of genetic background and/or RU-486 in *UAS chinmo*$^{RNAi}$/+ and *chinmo*$^1$/+ strains used for analysis in *Figure 2*.

**Figure supplement 1—source data 1.** Lifespan analysis of flies encoding a single copy of the *UAS chinmo*$^{RNAi}$ transgene and a strain that is heterozygous for *let-7-C* and *chinmo*.

**Figure supplement 1—source data 2.** Survival proportions for strains used in *Figure 2—figure supplement 1*.

were fed a DR diet rescued the decrease in TAG levels observed in *miR-125* mutants. The lipid droplet size in Δ*miR-125, chinmo*$^1$ flies that were fed a DR diet did not vary significantly from the lipid droplet size in the Rescue flies upon DR (*Figure 2H–I*). However, the lipid droplet size was not rescued in *let-7-C Gal4> ΔmiR-125, UAS chinmo*$^{RNAi}$ flies (*Figure 2H–I*). Thus, reducing *chinmo* dosage genetically rescued the DR phenotype of *miR-125* mutants more effectively than knocking down *chinmo* levels in *miR-125* expressing cells. Taken together, these data revealed that *chinmo* acts downstream of *miR-125* to regulate lifespan extension upon DR.

## DR induces *chinmo* in the adult fat tissue

To examine whether DR-mediated regulation of *chinmo* in the adult fat tissue was required for extension of lifespan, we examined whether Chinmo protein was upregulated in the fat tissue of *miR-125* mutants, Rescue and $w^{1118}$ flies that were fed an AL and DR diet (*Figure 2B*, *bottom panels* and *Figure 3A–C*). Immunostaining of dissected abdominal fat tissue with anti-Chinmo antibody indicated an increased nuclear signal for Chinmo in *miR-125* mutant adult fat tissue dissected from flies that were fed an AL diet (*Figure 2B*). In contrast, an increased non-nuclear/peripheral signal was detected in both Rescue and *miR-125* mutant fat tissue of flies that were fed a DR diet (*Figure 2B*). These data indicated that nutrient restriction caused redistribution of Chinmo to the non-nuclear compartments. To explore whether Chinmo was upregulated by DR in the fat tissue, we examined Chinmo protein and mRNA in flies that expressed wild-type levels of *miR-125*. Abdominal fat tissue was dissected from $w^{1118}$ flies that were fed an AL or DR diet for 10 days. The tissues were fixed and immunostained with an anti-Chinmo antibody to detect Chinmo and anti-La antibody was used as a control for immunostaining along with Dapi for nuclear staining (*Figure 3A*). $w^{1118}$ flies that were fed a DR diet displayed a much stronger immunostaining signal for Chinmo in the fat tissue than flies

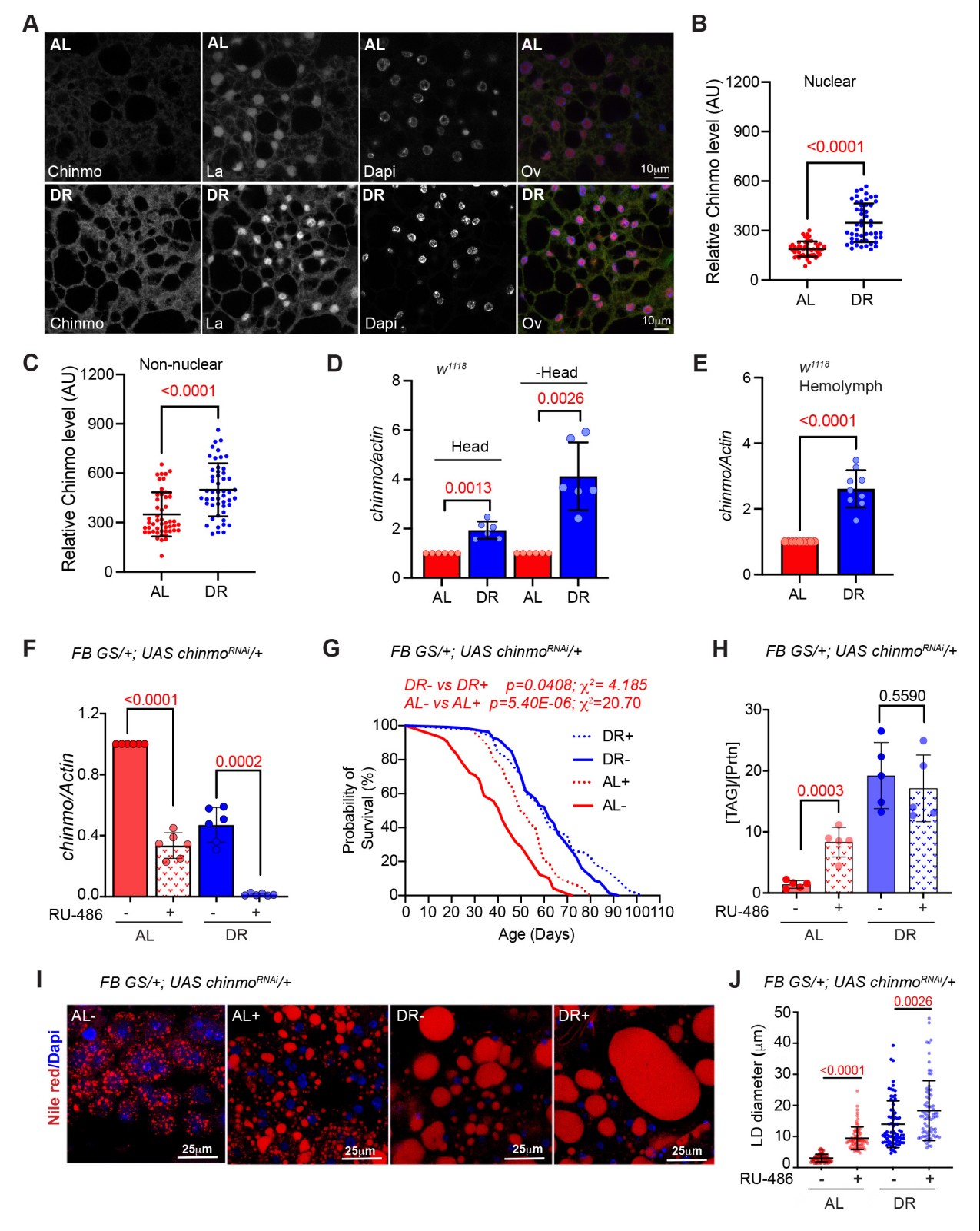

**Figure 3.** Chinmo is nutrient-dependent downstream effector of the DR-pathway. (**A**) Chinmo protein levels increase in adult fat tissue of $w^{1118}$ flies that were fed a DR diet for 10 days. Confocal images of dissected fat body from adult $w^{1118}$ flies that were fed an AL or DR diet for 10 days. Immunostaining was performed with anti-Chinmo, anti-La (nuclear protein), and Dapi. Scale bar, 10 μm. (**B, C**) Quantitation of Chinmo protein levels in adult fat tissue nuclear and peripheral (non-nuclear) regions as determined by measuring pixel intensity in 10 cells in five samples per condition. Error bars represent

*Figure 3 continued on next page*

*Figure 3 continued*

mean ± SD and p values were calculated by unpaired t test with Welch's correction. (**D**) DR induces expression of *chinmo* in $w^{1118}$ flies. RT-PCR quantitation of *chinmo* mRNA in head and decapitated body tissue (-Head) of $w^{1118}$ flies that were fed an AL or DR diet for 10 days. (**E**) DR increases circulating levels of *chinmo* mRNA in $w^{1118}$ flies. RT-PCR quantitation of *chinmo* mRNA in the hemolymph of $w^{1118}$ flies that were fed an AL or DR diet for 10 days. (**F-J**) Reducing *chinmo* levels in the adult fat body increases lifespan and enhances TAG levels under AL conditions. (**F**) A transgene expressing a short hairpin to knockdown *chinmo* was expressed in the adult fat body using the steroid (RU-486) inducible gene switch Gal4 driver. Quantitative RT-PCR of *chinmo* from abdominal fat tissue of *FB GS >UAS chinmo^{RNAi}* flies in presence of RU-486 (bars with red and blue pattern) or in absence of RU-486 (bars with solid red and blue color) in flies that were fed Ad libitum (AL) (red) or DR diet (blue) for 10 days. Expression levels were normalized to *Actin*. Values are mean ± SD, n = 6. p Values are indicated in the graph and were calculated by unpaired t test with Welch's correction. (**G**) Knockdown of *chinmo* in the adult fat tissue resulted in a 16% increase in the median lifespan of flies that were fed an AL diet (compare red solid line with the red dotted line) (*Figure 3—source data 1A–B* has details of experimental repeat). For statistical comparison of survival curves, p values and $\chi^2$ were calculated with log rank test. (**H**) Quantitation of triglyceride (TAG) stored levels in AL-RU-486 (Solid red bars), AL + RU-486 (Red pattern bars), DR-RU-486 (Solid blue bars) and DR +RU-486 (Blue pattern bars) fed 20-day-old *FB GS >UAS chinmo^{RNAi}* flies. The bars represent mean ± SD, n = 5, p value was calculated with two tailed t-test. Significance levels: ***p<0.001. (**I**) Fat bodies of *FB GS >UAS chinmo^{RNAi}* female flies were dissected and stained for the content and diameter of lipid droplet (LD) (red are lipid droplets stained with Nile red and blue is Dapi). Scale bar, 25 μm. (**J**) Quantitation of lipid droplet (LD) diameter in (**I**). Quantitation of 15 largest LDs in five samples per condition. Error bars represent mean ± SD and p values are indicated in the graph and were calculated by using unpaired t test with Welch's correction. Genotypes of strains used in this figure: (**A-E**) $w^{1118}$; (**F-J**) *FB GS/+; UAS chinmo^{RNAi}/+; w^{1118}; P{w[+mW.hs]=Switch1}106/+; P{w+, UAS-chinmo^{RNAi 148}}VK00033/+.*

The online version of this article includes the following source data and figure supplement(s) for figure 3:

**Source data 1.** Lifespan analysis of *FBGS >UAS chinmo^{RNAi}* strain.
**Source data 2.** Survival proportions of lifespan curves of *FBGS >UAS chinmo^{RNAi}* strain.
**Figure supplement 1.** Nutrient-dependent regulation of Chinmo in the adult fat tissue.

that were fed an AL diet for 10 days (*Figure 3A*). These data were recapitulated in fat tissue obtained from flies that were fed an AL or DR diet for 20 days and with two different antibodies to detect Chinmo (*Figure 3—figure supplement 1A*). To distinguish whether the increase in the intensity of Chinmo immunostaining occurs in the nuclear or peripheral compartments of the cell, the intensity of the signal generated by immunostaining was quantitated in nuclear and non-nuclear regions (*Figure 3B–C*; *Figure 3—figure supplement 1B–E*). DR increased the expression of Chinmo in both nuclear and non-nuclear regions of the fat tissue. While $w^{1118}$ flies that were fed an AL diet expressed 188.7 ± 46.17 Arbitrary units (AUs) of Chinmo in the nuclear region, feeding flies a DR diet increased nuclear expression of Chinmo to 347.58 ± 117.18 AUs (*Figure 3B*; *Figure 3—figure supplement 1B and D*). An increase in the intensity of Chinmo signal was detected in non-nuclear regions upon DR (Non-nuclear AL: 349.19 ± 133.90 AU; Non-nuclear DR: 498.56 ± 160.89 AU) (*Figure 3C*; *Figure 3—figure supplement 1C and E*). Wild-type ($w^{1118}$) flies that were fed a DR diet for 10 days expressed higher levels of *chinmo* RNA than the AL-fed flies (*Figure 3D*). The magnitude of increase was higher in tissues other than head despite lower levels of the transcript in decapitated tissue as DR led to a 1.94 ± 0.35 fold increase in head tissue and a 4.11 ± 1.37 fold increase in decapitated body tissue (*Figure 3D*). We previously found *miR-125* to be upregulated upon DR (*Figure 1D*).

To examine whether *miR-125* and *chinmo* were differentially upregulated by DR in a tissue-specific manner, the expression of *chinmo* and *miR-125* was quantitated by extracting total RNA from the head and dissected fat tissue of $w^{1118}$ flies that were exposed to AL or DR diet for 20 days (*Figure 3—figure supplement 1F–G*). Both *chinmo* and *miR-125* were expressed at much higher levels in the adult head tissue and DR induced *chinmo* by 1.29 ± 0.22 fold and *miR-125* by 2.33 ± 0.28 fold in this tissue (*Figure 3—figure supplement 1F–G*). Thus, the greater increase in *miR-125* upon DR ensures that *chinmo* continues to be repressed in the brain. Both *chinmo* and *miR-125* are expressed at much lower levels in the fat body tissue as compared to the head tissue (*chinmo*: 5.5% of the levels in the head tissue; *miR-125*: 39.8% of the levels in head tissue). However, DR induces *chinmo* by 2.5 ± 0.98 fold and *miR-125* by 1.89 ± 0.6 fold in the adult fat body (*Figure 3—figure supplement 1F–G*). *Chinmo* mRNA levels also increased by 2.6 ± 0.19 fold in the hemolymph of $w^{1118}$ adult flies that were fed a DR diet for 10 days (*Figure 3E*). Hemolymph is responsible for circulating nutrients that are absorbed by the midgut to the fat body where carbohydrate and lipid storage, mobilization, and de novo lipogenesis occurs. A DR-dependent increase in the levels of circulating *chinmo* mRNA hinted at a possible mechanism by which *chinmo* could function in a non-autonomous manner. To examine whether a similar increase in Chinmo protein was detected in the adult brain, dissected

brain samples from flies that were exposed to AL or DR conditions for 20 days were immunostained with anti-chinmo antibody to detect Chinmo, anti-Dacshund antibody as a control for immunostaining, and Dapi for DNA staining. As expected, the high levels of *miR-125* in brain tissue repressed *chinmo* and no Chinmo signal was detected in adult $w^{1118}$ brain tissue even when flies were exposed to DR for 20 days (*Figure 3—figure supplement 1H*). Together these data provided evidence for a *miR-125*-independent regulation of *chinmo* in the DR pathway in tissues other than the brain and raised the possibility of a possible non-nuclear role for this protein under conditions of nutrient restriction (*Figure 3—figure supplement 1I*).

## Knockdown of *chinmo* in the adult fat body enhances lifespan under AL conditions

To address the contribution of *chinmo* in DR-dependent increase in lifespan, we examined the survival of flies expressing a *chinmo^RNAi^* transgene in the adult fat tissue (*Figure 3F–J*). A fat body (FB) gene switch Gal4 driver was used for overexpressing *UAS-chinmo^RNAi^* in an inducible manner (*Figure 3F*). Knockdown of *chinmo* resulted in a 16% increase in median lifespan in flies that were fed an AL diet and a 3.2% decrease in median lifespan in DR diet (*Figure 3G*). Since no increase in lifespan upon DR was observed and flies that were fed an AL diet showed a significant increase in lifespan, it confirmed that the lifespan extension mediated by reducing *chinmo* operates predominantly through the DR pathway and reducing *chinmo* does not enhance it further. RU-486 did not show a difference in lifespan of the *UAS chinmo^RNAi^* flies (*Figure 2—figure supplement 1A* and *Figure 2—source data 1C–D*) and FBGS driver (*Bolukbasi et al., 2017*; *Huang et al., 2014*). Reducing *chinmo* in the adult FB also resulted in an increase in stored triglyceride levels and an increase in lipid droplet diameter in flies that were fed an AL diet (*Figure 3H–J*). These data confirmed that *chinmo* functioned as an effector of DR-mediated extension of lifespan.

## Chinmo redistributes to the cytoplasm upon nutrient deprivation

Chinmo's role as a transcriptional repressor is well-documented and consistent with this function, endogenous and exogenous Chinmo is predominantly localized to the nucleus (*Flaherty et al., 2010*; *Flaherty et al., 2009*). To examine whether the non-autonomous role of Chinmo is due to its ability to shuttle between the nucleus and cytoplasm we examined the expression of tagged Chinmo in a *Drosophila* embryonic cell line, Kc167. Kc167 is one of the cell lines used by modENCODE and expresses very low levels of endogenous *chinmo* (*Cherbas et al., 2011*). A GFP-tagged version of Chinmo (*UAS GFP-chinmo*) was generated for examining the sub-cellular localization of Chinmo with GFP antibody. In fed cells (CCM3 medium), GFP-tagged Chinmo predominantly localized to the nucleus (*Figure 4A*). In contrast, the protein relocalized to the cytoplasm when Kc167 cells were transferred to starvation conditions (2 mg/ml glucose in PBS) for 10 hr (*Figure 4A*). These data indicated that Chinmo protein was being modified upon nutrient deprivation to a form that was able to relocalize to the cytoplasm.

## Deacetylation of Chinmo is required for its nuclear export upon nutrient deprivation

Protein acetylation is a dynamic and reversible post-translational modification that has been implicated in the nutrient-dependent subcellular relocalization of non-histone proteins (*Huang et al., 2015*; *Narita et al., 2019*). Lysine is the most common residue at which protein acetylation and deacetylation occur and one major family of enzymes that regulate lysine deacetylation in a nutrient-dependent manner are Sirtuin enzymes (*Bao and Sack, 2010*). Sir2/SIRT1 is a conserved nuclear-localized, NAD+-dependent deacetylase that senses energy status and protects cells against metabolic stresses (*Chang and Guarente, 2014*; *Lee et al., 2019*). dSir2 is regulated by diet and environmental stress and regulates the aging process (*Banerjee et al., 2012*; *Lee et al., 2019*). Hence, we tested whether increasing the levels of the nuclear-localized Sirt1/dSir2 led to relocalization of GFP:: Chinmo to the cytoplasm (*Figure 4B–C*). Kc167 cells were co-transfected with Tubulin-Gal4, UAS GFP-Chinmo, and Flag-dSir2, and cells were fixed 48 hr after transfection and immunostained with anti-GFP to detect Chinmo, anti-flag antibody to detect dSir2 and Dapi to detect nuclear DNA (*Figure 4B–C*). Overexpression of Flag-dSir2 led to export of Chinmo into the cytoplasm under fed conditions. In absence of dSir2, 84.9 ± 4.69% of the cells had Chinmo localized to the nucleus and in

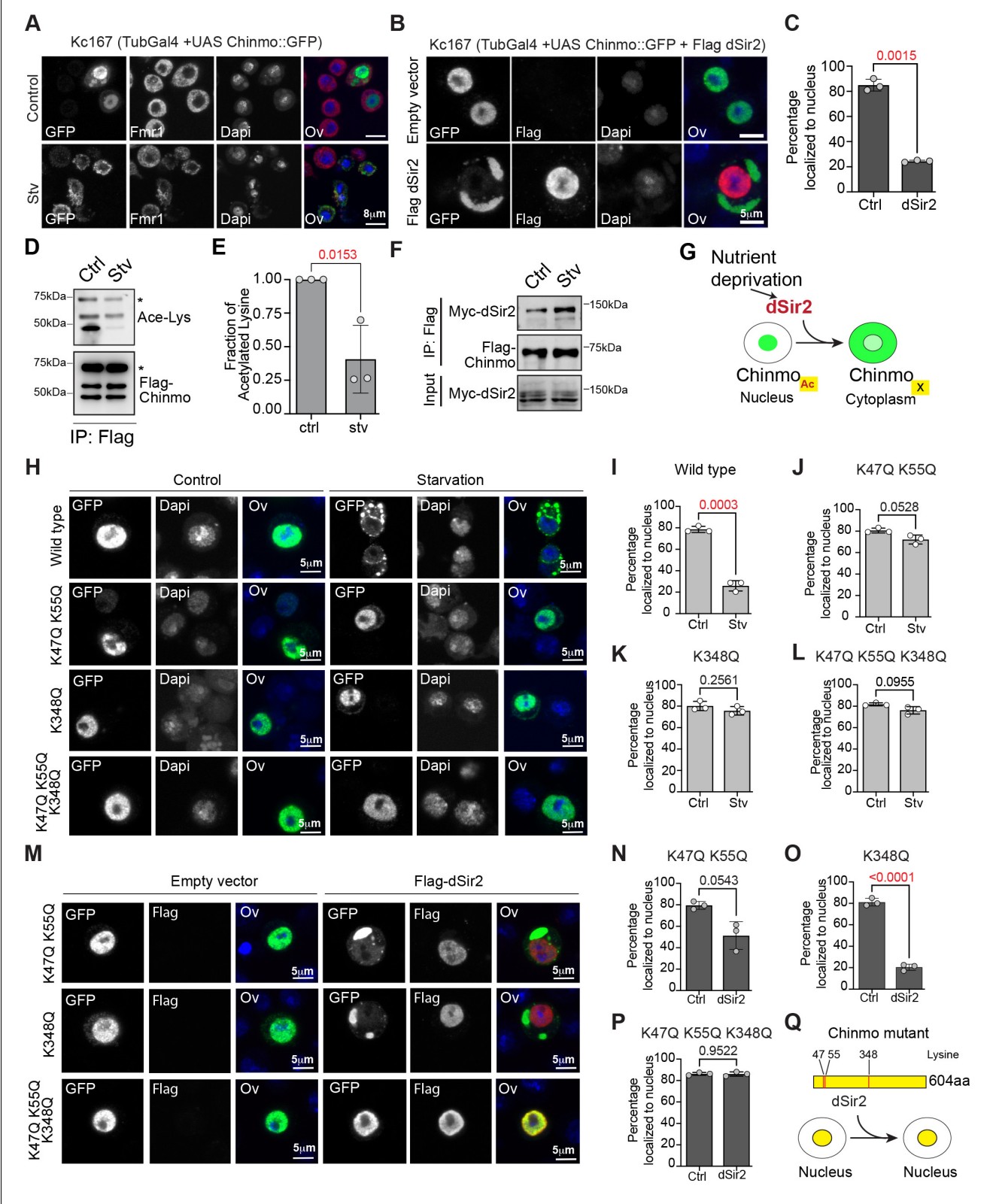

**Figure 4.** Export of Chinmo from the nucleus is dependent on deacetylation of Chinmo by dSir2. (**A**) Chinmo redistributes to the cytoplasm upon cell starvation. Kc167 cells were transfected with Tubulin Gal4 and UAS-GFP::Chinmo constructs and redistribution of Chinmo in fed (*upper panel*) and starved (*lower panel*) cells was examined by immunostaining with GFP antibody (Chinmo), Fmr1 (cytoplasmic marker) and Dapi (blue). Chinmo is predominantly localized to the nucleus in the fed state and redistributes to the cytoplasm in starved Kc167 cells. (**B, C**) Localization of GFP::Chinmo in

*Figure 4 continued on next page*

*Figure 4 continued*

cells overexpressing Flag dSir2 under fed/control (Ctrl) conditions. Kc167 cells were transfected with Tubulin Gal4 and UAS-Chinmo:: GFP and dSir2 pAFW constructs and redistribution of Chinmo was examined by immunostaining cells with GFP (Chinmo), Flag (dSir2), and Dapi. Overexpression of dSir2 led to redistribution of Chinmo to the cytoplasm under fed conditions. (**C**) Bar graph showing quantitation of the percentage of cells in which Chinmo::GFP was localized to the nucleus in cells transfected with either empty vector or dSir2 expressing plasmid. A total of 100 cells were quantitated in a minimum of three independent experiments. (**D, E**) Acetylation of Flag-Chinmo in fed (Ctrl) and Starvation (Stv) conditions. Acetylation of Flag-Chinmo was detected by immunoprecipitation with a Flag antibody followed by western blot analysis with an anti-acetyl-lysine antibody. Three bands of Chinmo were detected in the Flag immunoprecipitates reproducibly with the top-most band corresponding to the expected size of the full-length protein. (**E**) Bar graph showing the fraction of acetylated lysine in full-length Flag-Chinmo (indicated by * in panel **D**) under Ctrl and Stv conditions, determined by three independent immunoprecipitation experiments. (**F**) Co-immunoprecipitation of Myc-dSir2 with Flag-Chinmo under control and starvation conditions. Flag antibody was used for immunoprecipitation of Flag-Chinmo followed by western blot analysis with Myc antibody to detect dSir2 in the immunoprecipitates. (**G**) Schematic representing the nutrient-dependent Chinmo relocalization by dSir2. (**H-L**) K47, K55, and K348 are the main acetylation sites of Chinmo. The GPS-PAIL online prediction tool was used to identify potential Lysine residues in Chinmo that were acetylated. For table with all predicted acetylation Lysine residues please refer to *Figure 4—figure supplement 1A*. Kc167 cells were transfected with Tubulin Gal4 and GFP-tagged Chinmo wild type or each of the mutants in which the lysine (K) residues were mutated to Glutamine (Q) and the nuclear-cytoplasmic distribution of the tagged Chinmo proteins was examined under fed and starvation conditions by immunostaining with GFP(Chinmo) and Dapi. Scale bars represent 5 µm. (**H-I**) Majority of the wild-type GFP: : Chinmo protein redistributes to the cytoplasm upon starvation. (**H, J-L**) K47Q, K55Q double mutant, K348Q single mutant and K47Q, K55Q, K348Q triple mutant predominantly localize to the nucleus even upon starvation. (**M-P**) The K47Q, K55Q double mutant and K348Q single mutant show significant redistribution of Chinmo in the cytoplasm upon overexpression of dSir2. However, combining the three mutations in K47Q, K55Q, K348Q triple mutant abrogates the cytoplasmic export of Chinmo upon dSir2 overexpression. (**Q**) Schematic indicating the three lysine residues that are required for dSir2-mediated redistribution of Chinmo to the cytoplasm. The statistical results in all the bar graphs (**C, E, I-P**) are shown as mean ± SD (n = 3). p Values are calculated using unpaired t-test with Welch's correction and indicated in each of the graphs.

The online version of this article includes the following source data and figure supplement(s) for figure 4:

**Source data 1.** Uncropped western blots with relevant bands labeled for *Figure 4D–F*.
**Source data 2.** Original image files of the unedited for western blots in *Figure 4D*.
**Source data 3.** Original image files of the unedited for western blots in *Figure 4E*.
**Source data 4.** Original image files of the unedited for western blots in *Figure 4F*.
**Source data 5.** Morphology and fluorescence microscopy of Kc167 cells used in *Figure 4*.
**Figure supplement 1.** Evaluation of predicted acetylated lysine residues in Chinmo.
**Figure supplement 1—source data 1.** Uncropped western blots with relevant bands labeled for *Figure 4—figure supplement 1C*.
**Figure supplement 1—source data 2.** Original image files of the unedited for western blots in *Figure 4—figure supplement 1C*.
**Figure supplement 2.** Chinmo protein and mRNA are upregulated in adult and larval fat tissue upon starvation.

presence of dSir2, 24.4 ± 0.83% of the cells had Chinmo localized in the nucleus (*Figure 4C*). These data indicated that dSir2 was required for the export of Chinmo to the cytoplasm. We then examined the acetylation status of Flag-tagged Chinmo in control and starvation conditions (*Figure 4D–E*). Cellular starvation caused a 40 ± 25% decrease in acetylated Chinmo as determined by western blot analysis of immunoprecipitated Flag-Chinmo in control and starvation conditions (*Figure 4D–E*). Consistent with the requirement of dSir2 for the deacetylation of Chinmo during starvation conditions, we found that starvation induced interaction of Myc-dSir2 with Flag-Chinmo (*Figure 4F*). Thus, dSir2-mediated deacetylation of Chinmo is required for the nuclear export of Chinmo (*Figure 4G*).

To identify the specific lysine residues in Chinmo that are the potential sites of Acetylation, we utilized the GPS-PAIL (Prediction of Acetylation on Internal Lysines) online tool (*Deng et al., 2016*). A total of 11 acetylation sites were predicted by the online tool (*Figure 4—figure supplement 1A–B*). The validity of the predicted acetylation sites was assessed by generating Chinmo mutants in which each of the Lysine (K) sites were mutated to Glutamine (Q) (*Figure 4H–P* and *Figure 4—figure supplement 1D*). GFP-tagged Chinmo mutants were co-transfected with Tubulin Gal4 and the subcellular localization of each of the mutants was examined under control and starvation conditions by immunostaining with anti-GFP antibody (*Figure 4H–P* and *Figure 4—figure supplement 1D*). Double mutants were also generated and tested for Lysines that were nearby (K47Q and K55Q). We predicted that mutating the acetylation-specific lysines to Glutamines would result in loss of nuclear export of Chinmo. While the K47Q and K55Q single mutants were able to relocalize to the cytoplasm upon starvation, combining the two mutations in K47QK55Q double mutant as well as the K348Q single mutation led to a significantly diminished nuclear export of Chinmo upon starvation. While only 26 ± 5% of the cells had Chinmo wild type protein localized to the nucleus upon

starvation, 72 ± 4% of the cells transfected with K47QK55Q mutant and 76 ± 4% of the cells transfected with K348Q mutant retained the protein in the nucleus upon starvation (*Figure 4—figure supplement 1D* and *Figure 4H–J,K*). We then tested whether the identified lysine residues were critical for dSir2-mediated relocalization of Chinmo to the cytoplasm. The K47QK55Q and K348Q mutants were co-transfected with Tubulin-Gal4 and Flag dSir2 and cellular localization of Chinmo was analyzed by immunostaining of fixed cells with anti-GFP antibody to detect Chinmo, anti-flag antibody to detect dSir2 and Dapi to detect DNA (*Figure 4M*). Our prediction was that if deacetylation sites were mutated, cells would retain Chinmo in the nucleus even upon dSir2 transfection. For K47QK55Q mutant 51 ± 13% of transfected cells retained Chinmo in the nucleus upon co-transfection of dSir2, while only 20 ± 3% of the cells retained Chinmo in the nucleus upon co-transfection of K348Q with dSir2. Since, both the mutants (K47QK55Q and K348Q) were exported to the cytoplasm upon dSir2 overexpression, we generated a Chinmo triple mutant (K47QK55QK348Q) construct by combining the three mutations and analyzed the subcellular localization upon starvation and upon co-transfection with Flag-dSir2 (*Figure 4H,M,L,P*). 76 ± 3% of the cells retained Chinmo triple mutant (K47QK55QK348Q) in the nucleus upon starvation and 86 ± 2% of the cells retained Chinmo K47QK55QK348Q mutant in the nucleus upon co-transfection with dSir2 (*Figure 4L–P*). These data confirmed that dSir2-mediated deacetylation at all three residues (K47, K55, and K348) was required for redistribution of Chinmo to the cytoplasm upon nutrient deprivation (*Figure 4Q*).

Since our data showed that nutrient deprivation by starvation was responsible for cellular relocalization of Chinmo, we tested whether Chinmo protein and *chinmo* mRNA were regulated by starvation *in vivo* (*Figure 4—figure supplement 2*). Immunostaining of dissected abdominal fat tissue was performed with anti-Chinmo antibody to detect Chinmo, anti-Woc antibody as an immunostaining control, and Dapi for DNA. Fat body was dissected from adult flies and 2nd instar larvae as 2nd instar larvae express Chinmo in their fat body and we predicted that starvation induced Chinmo relocalization maybe utilized for its role during non-feeding larval states. A significantly increased intensity of Chinmo staining was observed in the nuclei of adult (Nuclear Ctrl: 15.3 ± 5.14 AU; Nuclear Stv: 19.13 ± 6.26 AU) and 2nd instar larval fat body (Nuclear Ctrl: 1835.33.19 ± 469.64 AU; Nuclear Stv: 3525.38 ± 904.48 AU) (*Figure 4—figure supplement 2A–B and C–D*). In adult fat tissue, peripheral Chinmo intensity was higher than that of the nuclear region and displayed a statistically significant increase upon starvation (Non-nuclear Ctrl: 282.05 ± 96.45 AU; Non-nuclear Stv: 432.36 ± 129.8 AU) (*Figure 4—figure supplement 2A and C*). In 2nd instar larvae a predominant increase in Chinmo intensity was observed in the nuclear region but not in the peripheral region (Non-nuclear Ctrl: 1114.18 ± 249.63 AU; Non-nuclear Stv: 832.69 ± 292.6 AU) (*Figure 4—figure supplement 2D and F*). One possible explanation for this difference could be the differences in the levels of dSir2 in the fat tissue of second instar larvae versus adult fat tissue and also due to the duration of starvation and a stronger transcriptional induction in the larval fat tissue. A significant increase in *chinmo* mRNA levels was detected by Q-RTPCR of total RNA extracted from 2nd instar larvae (3.98 ± 0.83 fold increase relative to control) and adult flies (3.98 ± 1.66 fold increase relative to control) that were exposed to starvation (*Figure 4—figure supplement 2G–H*). We also examined the effect of starvation in the head tissue versus the decapitated head tissue (*Figure 4—figure supplement 2I*). Our analysis revealed that the magnitude of induction of *chinmo* in decapitated tissue (3.13 ± 0.59 fold increase relative to control) was much higher than the magnitude of induction in head tissue (1.39 ± 0.14 fold increase relative to control), even though *chinmo* is expressed at higher levels in the head tissue (*Figure 4—figure supplement 2G–H*). Taken together, these analyses confirmed that in addition to *miR-125*-mediated post-transcriptional regulation, *chinmo* is independently regulated by nutrient deprivation at the transcriptional and post-translational levels.

## Starvation induces loading of Chinmo into exosomes

Exosomes mediate cell-cell communication by facilitating the intercellular transfer of specific combinations of bioactive molecules such as DNA, miRNAs, mRNAs, proteins, and lipids (*Maas et al., 2017*; *Tkach and Théry, 2016*). These extracellular vesicles are produced by almost all cell types and serve as intercellular messengers and carriers of various signals and molecules and modify the functioning of recipient cells in different contexts. Our data indicate that the deacetylation of Chinmo results in its nuclear export upon starvation. To test whether the long-range non-autonomous effects of Chinmo were associated with its loading onto exosomes, we examined the composition of purified exosomes upon nutrient deprivation. To facilitate biochemical purification and

detection of Chinmo by western blot analysis, we generated a Flag-tagged Chinmo construct (*UAS flag-chinmo*) (*Figure 5A–B*). Consistent with the GFP-tagged Chinmo data, starved Kc167 cells expressing Flag-tagged Chinmo also showed a cytoplasmic redistribution of Chinmo (*Figure 5C–H*). Moreover, this cytoplasmic redistribution of Chinmo was reversible, and shifting the cells back to normal medium (16 hr) resulted in the reappearance of nuclear-localized Chinmo and disappearance of cytoplasmic protein (*Figure 5I–N*). To examine whether non-autonomous expression occurs due

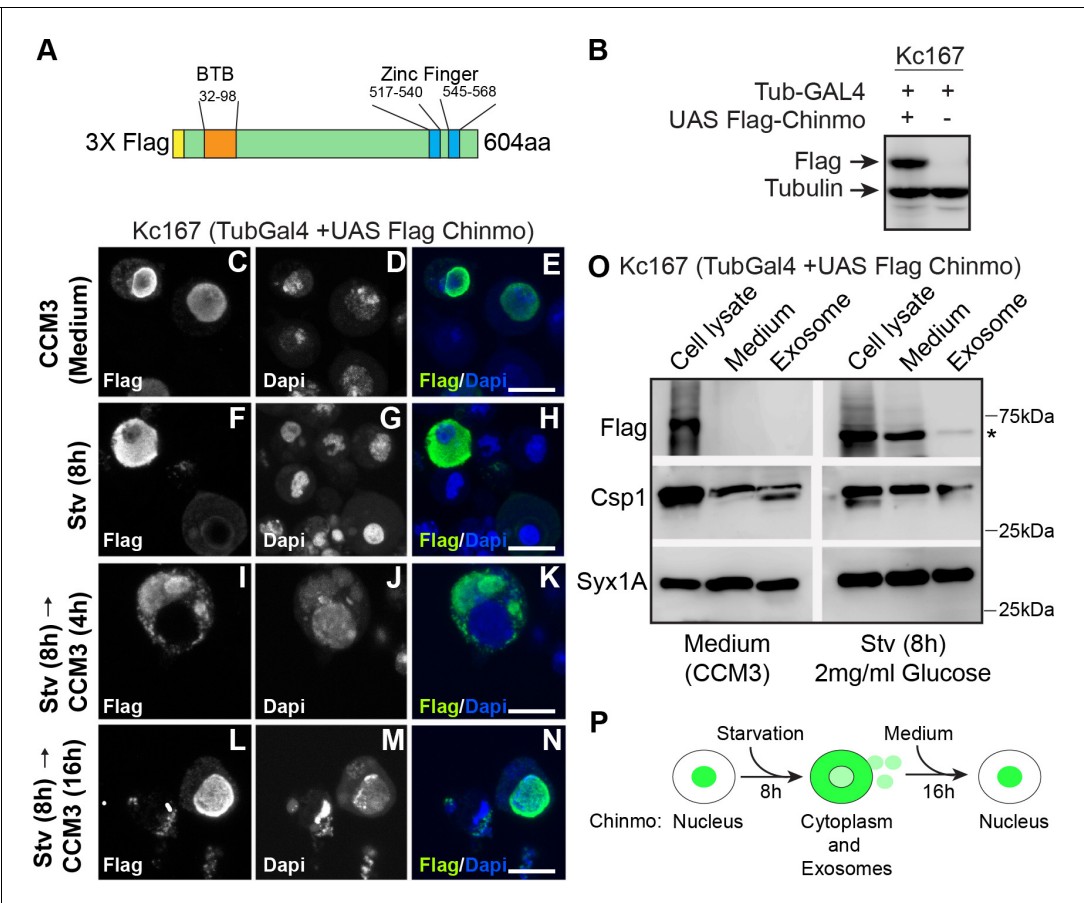

**Figure 5.** Chinmo co-fractionates with exosomal proteins upon nutrient deprivation. (A) Schematic of the 3X Flag-Chinmo protein indicating the BTB and Zinc finger domains. (B) Western blot analysis to confirm expression of Flag-tagged Chinmo in Kc167 cells transfected with Tubulin Gal4 plasmid and either empty vector (pUASTattB) or UAS Flag Chinmo. (C-N) Kc167 cells were transfected with Tubulin Gal4 and Flag Chinmo and distribution of Chinmo was examined in fixed cells after immunostaining with Flag and Dapi. Chinmo is predominantly localized to the nucleus in the fed state (C-E) and redistributes to cytoplasm when Kc167 cells are starved for 8 hr (F-H). (I-N) Chinmo distribution in Kc167 starved cells that are shifted to medium (CCM3) for 4 hr (I-K) and 16 hr (L-N). Chinmo translocates to the nucleus when cells are shifted from starvation to nutrient-rich conditions for 16 hr. Confocal images were acquired at ×100 magnification. Scale bars represent 8 μm. (O) Western blot analysis of cell lysate, concentrated culture medium and purified exosomes from Kc167 cells. Kc167 cells were transfected with Tubulin Gal4 and Flag Chinmo and 72 hr post-transfection cells were divided into flasks with either medium (CCM3) or 2 mg/ml glucose (PBS) for 8 hr. The medium from both conditions was collected and concentrated using Amicon centrifugal columns. Exosomes were purified from concentrated medium by gel exclusion chromatography. The exosomal fraction, concentrated medium and, cell lysate were analyzed by western blotting with Flag (Flag Chinmo), Csp 1 (exosome marker), and Syntaxin 1A (exosome marker) antibodies. Flag Chinmo was detected in exosomes and concentrated medium of Kc167 cells that were starved for 8 hr (Asterisk symbol). (P) Schematic representing dynamic redistribution of Chinmo in Kc167 cells exposed to starvation for 8 hr followed by a shift to medium for 16 hr.

The online version of this article includes the following source data and figure supplement(s) for figure 5:

**Source data 1.** Uncropped western blots with relevant bands labeled for *Figure 5B and O*.

**Source data 2.** Original image files of the unedited western blots in *Figure 5B and O*.

**Figure supplement 1.** Elution of vesicles using gel filtration columns.

**Figure supplement 1—source data 1.** Uncropped western blots with relevant bands labeled for *Figure 5—figure supplement 1B*.

**Figure supplement 1—source data 2.** Original image files of the unedited western blots in *Figure 5—figure supplement 1B*.

to intercellular transfer of exosomes, we examined the protein composition of exosomes derived from Kc167 cells that were exposed to either fed or starvation conditions. Kc167 cells were transfected with Tubulin Gal4 and Flag Chinmo constructs. Seventy-two hr post-transfection, one-half of the cells were shifted to starvation medium for 8 hr and the other half was maintained in fresh complete medium (CCM3) for 8 hr. Exosomes were purified from a concentrated medium from both conditions using gel filtration chromatography (*Figure 5—figure supplement 1A–B*). Western blot analysis was performed with the cell lysate, concentrated medium, and purified exosomes. Flag Chinmo was detected in the medium and exosome fraction of starved Kc167 cells but not in the concentrated medium and exosomal fraction of cells that were maintained in the complete medium (*Figure 5O*). These data uncovered a new mechanism by which Chinmo may be delivered to adjacent or distant cells.

## Overexpression of *chinmo* diminishes the DR-mediated lifespan extension

To gain an understanding of how tissue-specific regulation of *chinmo* results in DR-dependent lifespan extension, we measured the survival of fruit flies that ectopically expressed *chinmo* in adult neurons and fat tissues under AL and DR conditions. Consistent with the *miR-125* single mutant phenotype, overexpression of *chinmo* or *Flag- chinmo* using a neuronal gene switch driver (*3X Elav GS*) or fat-body-specific gene switch driver (*BL8151* also referred to as $S_1106$) resulted in the decrease in the DR dependent lifespan extension (*Figure 6A–E*; *Bolukbasi et al., 2017*; *Gendron and Pletcher, 2017*; *Huang et al., 2014*). A significantly dampened DR-mediated lifespan extension was observed when *chinmo* was overexpressed in the neurons (*Figure 6B* and *Figure 6—source data 1*). Though, a 24% increase in median survival upon DR in uninduced conditions (-RU) and a 37.5% increase in median survival upon DR in induced conditions (+RU) was observed. The p-values (AL- *vs* DR-, 0.00 E + 00; AL + *vs* DR+, 0.0066) and $X^2$ values (AL- *vs* DR-, 70.02; AL + *vs* DR+, 7.37) for comparison of lifespan curves in induced and uninduced conditions indicated that the DR-dependent increase in lifespan was greatly diminished upon overexpression of *chinmo* in adult neurons (*Figure 6B* and *Figure 6—source data 1*). This DR phenotype was further exaggerated in the *FBGS >UAS chinmo* flies where a 50% increase in median survival was observed upon DR in -RU conditions and a 0% increase in median survival was observed upon DR in +RU conditions (*Figure 6C* and *Figure 6—figure supplement 1*). These results were reproduced in crosses of the same GeneSwitch drivers with another transgenic line that overexpressed a Flag-tagged Chinmo (*UAS flag chinmo*). In *3X ElavGS >UAS flag chinmo* flies, a 37% increase in median survival was recorded upon DR in uninduced conditions (-RU), and a 30% increase in median survival upon DR was recorded in induced conditions (+RU). In *FBGS >UAS flag chinmo* flies, a 22% increase in median survival was recorded upon DR in uninduced conditions (-RU) and, a 0% increase in median survival upon DR was recorded in +RU conditions (*Figure 6D–E* and *Figure 6—source data 1*). The *3X ElavGS, UAS-chinmo* and *UAS flag-chinmo* control lifespans were performed (AL+, AL-, DR+, DR-) to confirm that the decrease in DR-mediated lifespan extension was not due to the effect of ligand or genetic background (*Figure 6—figure supplement 1A–C* and *Figure 6—figure supplement 1—source data 1*). The effect of the ligand on FBGS ($S_1106$) lifespan have been published before and referenced here (*Bolukbasi et al., 2017*; *Huang et al., 2014*). In addition, measurement of TAG levels and lipid droplet size indicated that ectopic neuronal expression of *chinmo* or *flag chinmo* (*3X ElavGS*) resulted in decreased TAG levels and a concomitant reduction in lipid droplet diameter in both AL and DR diets (*Figure 6F–I*, *Figure 6—figure supplement 1H–I*). We predicted that overexpression of *chinmo* would result in a decrease of TAG and lipid droplet size irrespective of the diet. Consistent with the *miR-125* single mutant metabolic phenotypes (*Figure 2G–I*), ectopic expression of *chinmo* in the adult fat body with an inducible fat-body-specific GAL4 driver (*Figure 6—figure supplement 1D–G*) resulted in reduced TAG levels (*Figure 6—figure supplement 1E*) and a decrease in lipid droplet diameter (*Figure 6—figure supplement 1F–G*). Together, these data demonstrate that overexpression of *chinmo* in adult neurons mimics the *miR-125* mutant phenotype and confirms a role for *miR-125* in DR pathways via post-transcriptional repression of *chinmo*.

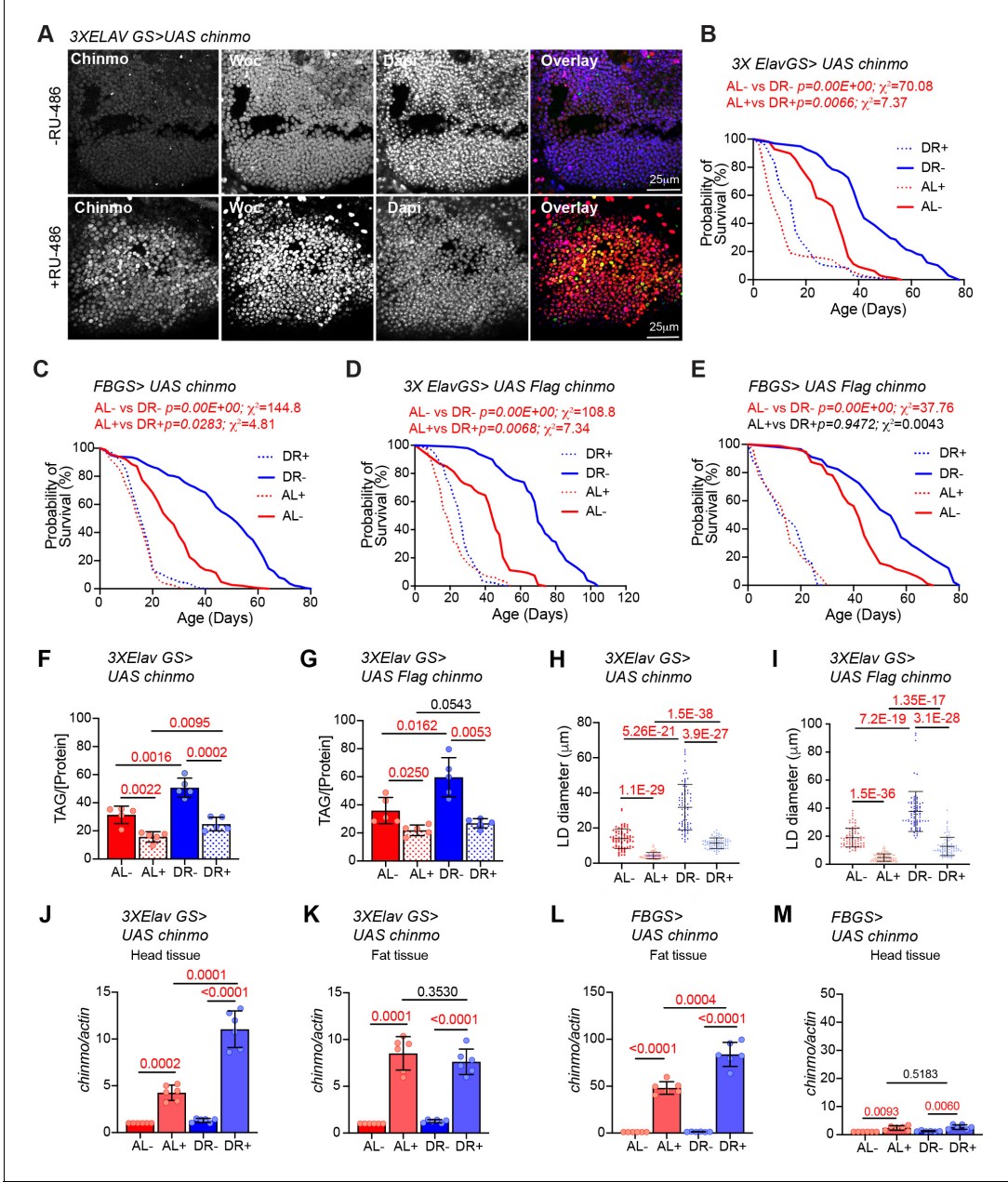

**Figure 6.** Ectopic expression of *chinmo* in adult neurons and fat body mimics *miR-125* mutant DR phenotype. *UAS chinmo* was expressed in adult neurons and adult fat tissue using the steroid (RU-486) inducible gene switch Gal4 drivers. (A) Female flies that were fed an RU-486 supplemented diet for 5 days displayed increased levels of Chinmo in neuronal cells, as detected by Chinmo. Woc (nuclear marker) and Dapi staining of dissected adult fly brains. (B, D) *3X ElavGS >UAS chinmo* (B) or *3X ElavGS >UAS Flag chinmo* (D) flies that were not fed RU-486 show a significant increase in lifespan upon DR (DR, blue line) as compared to *3X ELavGS >UAS chinmo* (B) or *3X ElavGS >UAS Flag chinmo* (D) flies that were fed an *ad libitum* (AL) diet (red line)(3X *ElavGS >UAS chinmo*: AL-, n = 98; median lifespan = 32 d; DR-, n = 99; median lifespan = 42; *3X ElavGS >UAS Flag-chinmo*: AL-, n = 107; median lifespan = 44 d; DR-, n = 99; median lifespan = 70 d). RU-486 fed *3X ElavGS >UAS chinmo* flies or *3X ElavGS >UAS Flag-chinmo* display a significantly dampened DR-dependent increase in lifespan (blue and red dotted lines) (*3X ElavGS >UAS chinmo*: AL+, n = 100; median lifespan = 10 d; DR+, n = 95; median lifespan = 16 d; *3X ElavGS >UAS Flag-chinmo*: AL+, n = 137; median lifespan = 18 d; DR+, n = 100; median lifespan = 26 d). (C, E) *FB GS >UAS chinmo* or *FB GS >UAS Flag-chinmo* flies that were fed RU-486 displayed a stronger DR phenotype relative to *3X ElavGS >UAS Chinmo* or *3X ElavGS >UAS Flag chinmo* (blue and red dotted lines)(*FB GS >UAS chinmo*: AL-, n = 186; median lifespan = 26 d; DR-, n = 187; median lifespan = 52 d; AL+, n = 213; median lifespan = 16 d; DR+, n = 209; median lifespan = 16 d; *FB GS >UAS Flag-chinmo*: AL-, n = 96; median lifespan = 42; DR-, n = 91; median lifespan = 54; AL+, n = 83; median lifespan = 14 d; DR+, n = 73; median lifespan = 14 d) (Compare p values and $X^2$ in panels B/D with C/E) compared to the flies that were fed the Ethanol (blue and red solid lines). For statistical comparison of survival curves, p values and $X^2$ were calculated with log rank test. (F, G) Quantitation of triglyceride (TAG) stored levels in AL-RU-486 (Solid red bars), AL + RU-486 (Red dotted

*Figure 6 continued on next page*

*Figure 6 continued*

bars), DR-RU-486 (Solid blue bars) and DR +RU-486 (Dotted blue bars) fed 20d old (**F**) *3XElavGS > UAS chinmo* or (**G**) *3XElavGS > UAS Flag-chinmo* flies. Increasing Chinmo levels in adult neurons is sufficient to lead to a significant reduction in the systemic triglyceride levels under both AL and DR conditions. This decrease mimics the decrease in TAG levels observed in *miR-125* mutants. The bars represent mean ± SD, n = 5, p value was calculated with unpaired two tailed t-test with Welch's correction. (**H, I**) Fat bodies/abdomens of female flies were dissected and stained for the content and diameter of lipid droplet (LD) (red are lipid droplets stained with Nile red and blue is Dapi). For representative confocal images, please refer to *Figure 6—figure supplement 1H–I*. Quantitation of lipid droplet (LD) diameter in *3XElavGS > UAS chinmo* (**H**) and *3XElavGS > UAS Flag-chinmo* (**I**). A significant reduction is seen in lipid droplet size in the abdominal fat tissue when Chinmo is expressed in adult neurons. Quantitation of 15 largest LDs in five samples per condition. Error bars represent mean ± SD. (**J–K**) *3X ElavGS >UAS chinmo* flies express increased levels of *chinmo* mRNA in both head and fat tissue. (**L–M**) *FBGS >UAS chinmo* flies predominantly express *chinmo* mRNA in the adult fat tissue. (**J–M**) Quantitative RT-PCR of RNA extracted from dissected tissues (head and fat body) of *3XElavGS > UAS chinmo* (**J, K**) and *FBGS >UAS chinmo* (**L, M**) flies that were fed an AL (Red and pink bars) and DR (Dark Blue and Light blue bars) under uninduced (Red and Dark blue bars) and steroid induced (Pink and Light blue bar) conditions. Expression levels were normalized to *Actin5c*. Values are mean ± SD, n = 6. p Values were calculated using unpaired t test with Welch's correction. Genotypes of strains used in this figure: (**A, B, F, H, J-M**) *3X ElavGS >UAS chinmo: P{elav-Switch.O}GS −1A / +; P{elav-Switch.O}GS-3A, P{elav-Switch.O} GSG301/P{w+, UAS-chin::SV40}/+;* (**D, G,I**) *3X ElavGS >UAS Flag chinmo: P{elav-Switch.O}GS −1A / +; P{elav-Switch.O}GS-3A,P{elav-Switch.O} GSG301/P{w+, UAS Flag-chin::SV40} attP2 + /;* (**C**) *FBGS >UAS chinmo: w^{1118}; P {w [+mW.hs]=Switch1}106/+; P{w+, UAS-chin::SV40}/+;* (**E**) *FBGS >UAS Flag-chinmo: w^{1118}; P{w[+mW.hs]=Switch1} 106/+; P{w+, UAS Flag-chin::SV40} attP2 + /.*

The online version of this article includes the following source data and figure supplement(s) for figure 6:

**Source data 1.** Lifespan analysis of strains used in *Figure 6*.
**Source data 2.** Survival proportions of lifespan curves of strains used in *Figure 6B–E*.
**Figure supplement 1.** Lifespan and molecular analysis of strains used in *Figure 6*.
**Figure supplement 1—source data 1.** Lifespan analysis of strains used in *Figure 6—figure supplement 1A–C*.
**Figure supplement 2.** Non-autonomous expression of Chinmo in abdominal fat tissue of *3XElavGS > UAS Chinmo* flies.
**Figure supplement 3.** Non-autonomous expression of Chinmo in abdominal fat tissue of *3XElavGS > UAS Flag chinmo* flies.

## Neuronal *chinmo* upregulation induces *chinmo* expression non-autonomously in the fat tissue

In a previous study, we showed that *let-Complex* miRNAs are induced in the mid-late third larval instar and are predominantly expressed in the nervous system of adult flies (*Chawla et al., 2016*; *Chawla and Sokol, 2012*). However, our current analysis has revealed that derepression of *chinmo* in the nervous system and/or the abdominal fat tissue of *miR-125* mutants results in changes in systemic stored fat levels (*Figure 2B and G–I*, *Figure 6A–I*; *Figure 6—figure supplement 1D–I*). To determine the mechanistic basis for systemic regulation of fat metabolism by the *miR-125-chinmo* regulatory axis operating in the adult brain, we examined the expression of *chinmo* mRNA after inducing its expression with *3X ElavGS* and *FBGS* GeneSwitch Gal4 drivers (*Figure 6J–M*). Quantitative RT-PCR analysis of *chinmo* RNA in the head and dissected abdominal fat tissue of *3X ElavGS >UAS chinmo* flies and *FBGS >UAS chinmo* that were exposed to RU-486 for 5 days was performed. In the brain tissue of *3X ElavGS >UAS chinmo* flies, *chinmo* mRNA levels increased by 4.25 ± 0.8 fold under AL conditions and by 11 ± 1.96 fold in DR conditions upon addition of the ligand (*Figure 6J*). In addition, *chinmo mRNA* levels also increased by 8.5 ± 1.79 fold in AL conditions and by 7.6 ± 1.36 fold in DR conditions in the dissected fat tissue upon addition of the ligand (*Figure 6K*). These data suggested that *chinmo* mRNA was being induced non-autonomously to a similar magnitude in the abdominal fat tissue in *3X ElavGS >UAS chinmo* flies. In *FBGS >UAS chinmo* flies, *chinmo* levels increased by 48 ± 6.7 fold in AL conditions and by 83.9 ± 12.7 fold in DR conditions in the fat tissue (*Figure 6L*). However, a much smaller increase in *chinmo* mRNA levels was detected in the head tissue of *FBGS >UAS chinmo* with a 2.38 ± 0.8 fold increase in AL and 2.69 ± 0.8 fold increase in DR conditions upon addition of ligand (*Figure 6M*). To confirm whether the increase in mRNA also resulted in a non-autonomous increase in Chinmo protein levels, we performed immunostaining of dissected brain and abdominal fat tissue of *3X ElavGS >UAS chinmo* and *FBGS >UAS chinmo* flies (*Figure 6—figure supplement 2*). In parallel, we drove the expression of GFP using the *3X Elav GS* and *FB GS* drivers to verify the specificity of the GeneSwitch drivers (*Figure 6—figure supplement 2A–H and A'-H'*). Immunohistochemistry with GFP antibody confirmed the specificity of both the gene switch GAL4 drivers. GFP protein was detected in the adult brain and neurons innervating the abdominal wall of *3X ElavGS >UAS GFP* flies that were fed an AL + RU or DR +RU diet (*Figure 6—figure supplement 2A–H*). GFP was only detected in the fat body of *FB*

*GS >UAS GFP* flies upon RU treatment (*Figure 6—figure supplement 2A'–H'*). Then we performed immunostaining with anti-Chinmo antibody to detect Chinmo in dissected brain and fat body of *3X Elav GS >UAS chinmo* and *FB GS >UAS chinmo* flies that were fed an AL or DR diet in the presence and absence of RU-486 (*Figure 6—figure supplement 2I–P and I'-P'*). Consistent with the *chinmo* mRNA analysis, *3X Elav GS >UAS chinmo* flies that were fed RU-486 containing food displayed expression of Chinmo in the adult brain and adult fat body. However, *FB GS >UAS chinmo* flies that were fed RU-486 containing food displayed expression of Chinmo specifically in the fat body and no signal was detected in dissected brains (*Figure 6—figure supplement 2I'–P'*). These data were recapitulated using 3X ElavGS >UAS Flag chinmo flies (*Figure 6—figure supplement 3A–E*). Immunostaining was performed with anti-Chinmo and anti-Flag antibody to detect Chinmo in dissected brain, and fat body of *3X Elav GS >UAS Flag chinmo* flies that were fed an AL or DR diet for 10 days in presence or absence of RU-486. As expected Chinmo was detected with both antibodies in the dissected brains of *3X Elav GS >UAS Flag chinmo* flies upon induction. However, a more intense Chinmo signal was detected in DR diet. Chinmo was also detectable with both antibodies in the fat body of *3X Elav GS >UAS Flag chinmo* flies (*Figure 6—figure supplement 3B–C*). Quantitation of the signal intensity indicated a much higher levels of Chinmo in DR +RU-486 conditions (Chinmo AL-: $0.17 \pm 0.06$; Chinmo AL+: $0.59 \pm 0.39$; Chinmo DR-: $0.22 \pm 0.10$; Chinmo DR+: $2.04 \pm 0.75$; Flag AL-: $0.11 \pm 0.06$; Flag AL+:$0.5 \pm 0.35$; Flag DR-: $0.09 \pm 0.025$; Flag DR+:$1.5 \pm 0.74$) (*Figure 6—figure supplement 3D–E*).These data indicated that Chinmo protein and/or mRNA is either capable of loading onto exosomes to facilitate communication with other peripheral tissues or a feed-forward mechanism exists by which upregulation of *chinmo* in the adult neurons promotes its expression in the fat tissue *via* an as yet unidentified signal. This non-autonomous expression of Chinmo is required for mediating its effects on fat metabolism and lifespan.

## Chinmo downregulates expression of genes involved in fat metabolism

To identify potential targets of Chinmo that were responsible for the DR-dependent phenotypes of *miR-125* mutant, we performed semi-quantitative proteomic analysis of extracts prepared from adult flies (whole animals) overexpressing *chinmo* specifically in adult neurons using the 3X *Elav GS* Gal4 driver (*Figure 7A–E*). Since the role of Chinmo as a repressor of gene expression is well established, we examined the downregulated biological processes to identify relevant direct downstream targets of Chinmo (*Figure 7F*). Proteins that were identified to be significantly downregulated were predominantly genes that were involved in metabolism (*Figure 7F*). We successfully validated the expression of eight fat metabolism genes (*FATP, CG2017, CG9577, CG17554, CG5009, CG8778, CG9527,* and *FASN1*) that were identified through this proteomic analysis (*Figure 7G–H*) using RNA extracted from head tissue (*Figure 7G*) or decapitated fly tissue (*Figure 7H*) of *3XElavGS > UAS chinmo* flies that were fed a solvent (Ethanol) or RU-486 containing diet for 10 days (*Figure 7G–H*) or 2 days (*Figure 7—figure supplement 1A–E*). The rationale for examining the expression of the genes in head tissue *versus* decapitated fly tissue was to compare the magnitude of repression of the genes in a tissue where Chinmo was overexpressed with the peripheral tissues. Consistent with the proteomics data, the mRNAs of all the fat metabolism genes were significantly downregulated in the head tissue (51–78% relative to control) and in the decapitated fly tissue (27–50% relative to control) of flies that ectopically expressed *chinmo* in adult neurons (*Figure 7G–H*). These data revealed that neuronal upregulation of *chinmo* led to a repression of genes involved in fat metabolic processes in an autonomous (head) and non-autonomous manner (-head). To verify whether overexpression of *chinmo* in adult neurons led to a decrease in protein levels of the endogenous proteins involved in fat metabolism, we performed western blot analysis of whole fly lysates prepared from 3X *ElavGS >UAS flag chinmo* flies that were fed either an AL + RU-486 or AL + solvent/ethanol diet for 10 days (*Figure 7—figure supplement 3A*). Since, most of the downstream targets identified are not that well-characterized (*CG2107, CG9527, CG17544, CG5009, CG8778,* and *CG9577*), we tested the expression of endogenous Fatp for which antibodies were available. Western blot analysis was performed with anti-flag antibody to detect Chinmo, anti-Fatp antibody to detect endogenous Fatp and anti-tubulin as a normalization control. Consistent with the proteomics data, a reduction in Fatp levels was seen upon upregulation of Flag Chinmo in adult neurons (*Figure 7—figure supplement 3A*). To test whether downregulation of the candidate fat metabolism genes in the adult fat tissue was responsible for modulating lifespan, we measured survival of flies that expressed transgenes to knockdown *fasn1* and *fatp* specifically in the adult fat body (*Figure 7I,J*). We predicted that if

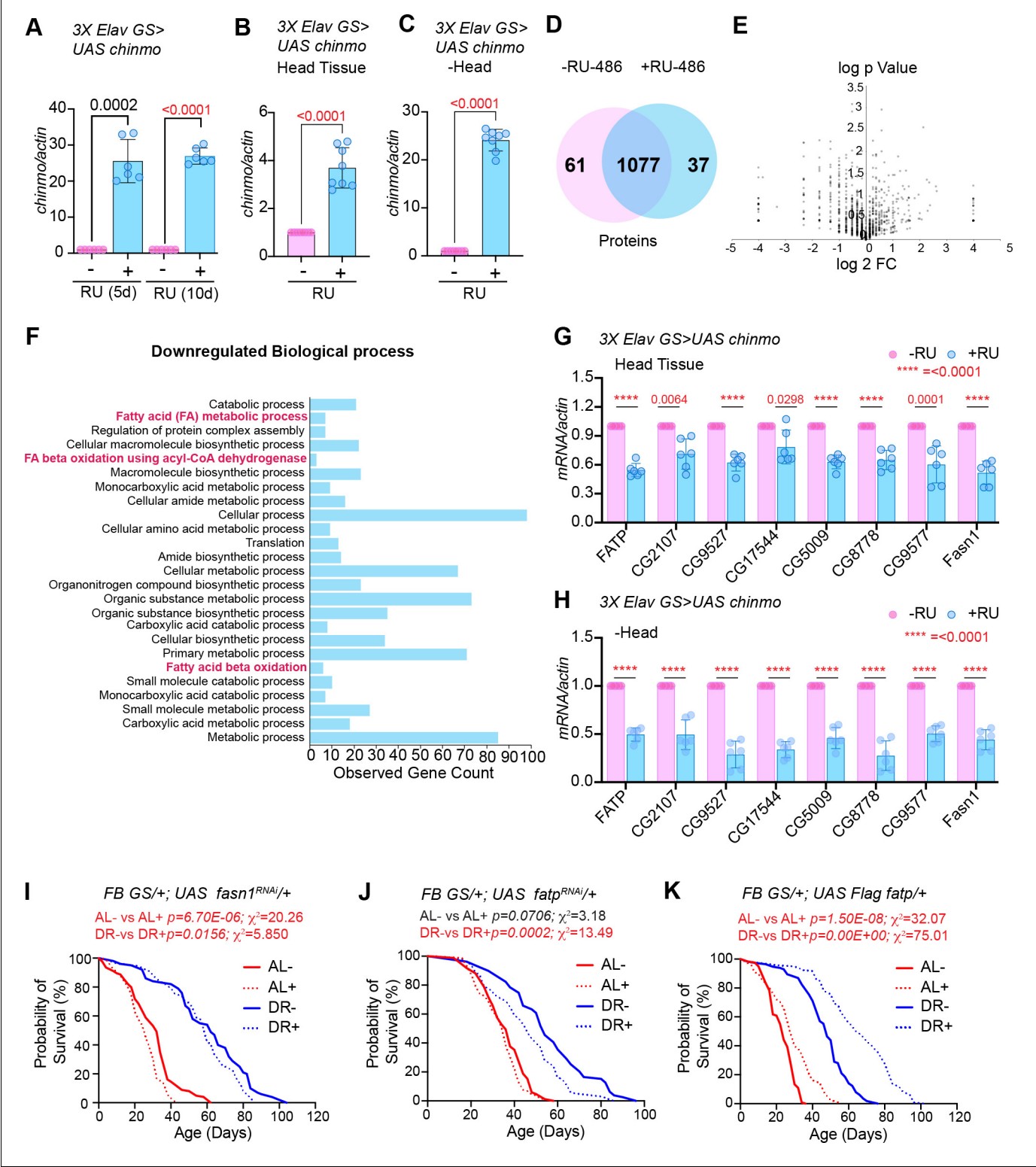

**Figure 7.** Ectopic expression of *chinmo* in adult neurons represses transcription of genes involved in fat metabolic processes. (**A**) Quantitation of *chinmo* mRNA levels in RNA extracted from *3X ElavGS >UAS chinmo* flies that were fed a solvent (light blue bars) or RU-486 (pink bars) diet for 5 and 10 days. (**B, C**) Quantitation of *chinmo* mRNA levels in RNA extracted from head tissue (**B**) and decapitated body tissue (**C**) of *3X ElavGS >UAS chinmo* flies that were fed ethanol (blue bar) or RU-486 (pink bar) diet for 10 days. Expression levels were normalized to *Actin5c*. Values are mean ± SD, n = 3. (**D**) Venn diagram of 1175 proteins identified across the two groups (-RU-486 and +RU-486) and 91% were common between the two groups. (**E**)
*Figure 7 continued on next page*

Figure 7 continued

Volcano plot illustrating significantly differentially abundant proteins. Forty proteins were found to be differentially expressed by using a cutoff on p value ≤ 0.05 and log$_2$FC ≥ 1 (7 upregulated) and ≤−1.0 (33 downregulated) proteins. (F) Twenty-five most significant biological processes that are downregulated upon overexpression of *chinmo* in adult neurons. See *Figure 7—figure supplement 2* for upregulated biological processes. (G, H) Overexpression of *chinmo* in the adult nervous system downregulates genes involved in fat metabolism. RT-PCR quantitation of fold- change in mRNA levels of genes (F) involved in fat metabolism in the head tissue (G) and decapitated body tissue (H) of *3X ElavGS >UAS chinmo* flies that were fed the solvent/ethanol (light blue bars) or RU-486 (pink bars) diet for 10 days. (I) Knockdown of *fasn1* in the adult fat body reduces lifespan more significantly in flies that are fed an AL (compare red solid and dotted lines) compared to flies that are fed a DR diet (compare blue solid and blue dotted lines) conditions (See *Figure 7—source data 1* for p values and median and maximum lifespan for additional experimental replicate). (J) Knockdown of *fatp* in the adult fat body reduces lifespan under DR (compare blue solid and blue dotted lines) conditions (See *Figure 7—source data 2* for p values and median and maximum lifespan for additional experimental replicate). (K) Overexpression of *Flag FATP* increases lifespan in flies that are fed a DR diet (Compare blue dotted line with blue solid line) more significantly than flies that are fed an AL diet (See *Figure 7—source data 3* for p values and median and maximum lifespan for additional experimental replicate). For statistical comparison of survival curves, p values and χ$^2$ were calculated with log-rank test. Genotypes of strains used in this figure: (A-H) *3X ElavGS >UAS chinmo: P{elav-Switch.O}GS −1A / +; P{elav-Switch.O}GS-3A, P{elav-Switch.O} GSG301/P{w+, UAS-chin::SV40}/+*; (I) *FB GS/+; UAS fasn1$^{RNAi}$/+: w$^{1118}$; P{w[+mW.hs]=Switch1}106/+; P{y[+t7.7] v[+t1.8]=TRiP.HMS01524} attP2/+*; (J) *FB GS/+; UAS fatp$^{RNAi}$/+:w[1118]; P{w[+mW.hs]=Switch1}106/+; P{y[+t7.7] v[+t1.8]=TRiP.HMC04206}attP2/+*; (K) *FB GS/+; UAS Flag fatp/+: w[1118]; P{w[+mW.hs]=Switch1}106/+;P{w+, UAS-Flag fatp} attP2/+*.

The online version of this article includes the following source data and figure supplement(s) for figure 7:

**Source data 1.** Lifespan analysis of *FBGS >UAS fasn1$^{RNAi}$* and *+/+; UAS fasn1$^{RNAi}$* strains.

**Source data 2.** Lifespan analysis of *FBGS >UAS fatp$^{RNAi}$* and *+/+; UAS fatp$^{RNAi}$* strains.

**Source data 3.** Lifespan analysis of *FBGS >UAS Flag fatp* and *+/+; UAS Flag fatp* strains.

**Source data 4.** Survival proportions of lifespan curves of strains used in *Figure 7I–K*.

**Source data 5.** Upregulated biological processes from proteomic analysis in *Figure 7D–F*.

**Source data 6.** Downregulated biological processes from proteomic analysis in *Figure 7D–F*.

**Figure supplement 1.** Overexpression of *chinmo* in adult neurons represses transcription of genes involved in fat metabolic processes in the brain within 2 days of induction.

**Figure supplement 2.** Over expression of *chinmo* in adult neurons upregulates cytoplasmic processes.

**Figure supplement 3.** Molecular and survival analysis of strains utilized in *Figure 7*.

**Figure supplement 3—source data 1.** Uncropped western blots with relevant bands labeled for *Figure 7—figure supplement 3A and D*.

**Figure supplement 3—source data 2.** Original image files of the unedited for western blots in *Figure 7—figure supplement 3A*.

**Figure supplement 3—source data 3.** Original image files of the unedited for western blots in *Figure 7—figure supplement 3D*.

**Figure supplement 3—source data 4.** Survival proportions of lifespan curves of strains used in *Figure 7—figure supplement 3E–G*.

increased expression of one of these genes was sufficient to cause increased lipid turnover and consequentially DR-mediated lifespan extension, then knockdown of this gene would result in a reduction in lifespan upon DR. RT-PCR analysis of total RNA extracted from abdominal fat tissue of *FBGS/+; UAS fasn$^{RNAi}$/+* flies exposed to AL or DR conditions was performed to verify the knockdown of *fasn1*. Induction of the *UAS fasn1$^{RNAi}$* led to a 67.1 ± 1.9% decrease in *fasn1* mRNA under AL conditions and a 52.87 ± 3.7% decrease in DR conditions (*Figure 7—figure supplement 3B*). Knockdown of *fasn1* resulted in an 18.7% decrease in median lifespan of flies that were fed an AL diet and a 3.22% decrease in median lifespan on DR diet (*Figure 7I* and *Figure 7—source data 1*). Thus, indicating that knocking down of *fasn1* alone was not sufficient to mediate the DR effect on longevity. Induction of the *UAS fatp$^{RNAi}$* led to a 41.9 ± 5.3% decrease in *fatp* mRNA under AL conditions and a 51.23 ± 2% decrease in *fatp* mRNA under DR conditions (*Figure 7—figure supplement 3C*). Knockdown of *fatp* resulted in a 14.8% decrease in median lifespan upon DR and no change under AL conditions (*Figure 7J* and *Figure 7—source data 2*). Since knockdown of *fatp* resulted in a decrease in the lifespan of flies that were fed a DR diet, we tested whether increasing the levels of *fatp* in the adult fat tissue led to an increase in the lifespan of flies under AL conditions. Western blot analysis of whole fly lysates prepared from *FBGS >UAS Flag fatp* flies that were fed an AL or DR diet in the presence and absence of RU-486 for 10 days revealed that Flag Fatp was expressed in an inducible manner, however, much lower levels of the flag-tagged protein was detected under AL conditions as compared to that in DR conditions (*Figure 7—figure supplement 3D*). Overexpressing *UAS-Flag-FATP* specifically in the adult fat tissue increased median life span by 14.2% under AL conditions and by 25% under DR conditions (*Figure 7K*; *Figure 7—source data 3*). The smaller increase in median lifespan in AL conditions is likely due to the lower induction of the protein, nevertheless, an increase in lifespan in both diets indicates that Fatp functions as a pro-longevity factor and that

DR-dependent increase in lifespan occurs due to an increase in the expression of two or more genes that are regulated by Chinmo. To test whether the changes in survival were not due to an effect of RU-486, control experiments were performed with *+/+, fasn1^RNAi, +/+; fatp^RNAi* and *+/+; UAS Flag fatp* lines (*Figure 7—figure supplement 3E–G* and *Figure 7—source data 1–3*). Taken together, these results indicated that ectopic expression of *chinmo* in the adult neurons causes decreased expression of fat metabolism genes in the fat tissue that resulted in a decreased lifespan.

## Overexpression of human primary *miR-125b-1* in the adult fat body extends lifespan

Given that DR-modulated upregulation of *chinmo* was the likely cause of the decrease in lifespan extension upon DR, we tested whether increasing the levels of the brain enriched *miR-125* in the adult fat body was able to mimic the beneficial effects of *chinmo^RNAi*. Our data indicate that DR induces *miR-125* levels in the fat tissue (*Figure 3—figure supplement 1G* and *Figure 8B*, compare -AL and -DR bars). These data are consistent with a previous study in the mouse model that identified *miR-125b* as one of the miRNAs that increases in the subcutaneous white fat tissue upon caloric restriction (*Mori et al., 2012*). Hence, we tested whether artificially increasing the levels of *miR-125* specifically in the fat tissue would lead to silencing of *chinmo* and consequentially lead to an increase in lifespan under AL conditions. Given the multi-targeting ability of miRNAs, it would be reasonable to expect that other potential mRNA targets of *miR-125* would also be repressed upon overexpression. Since the human and fly processed miR-125 sequences are identical we generated transgenic fly lines that expressed the human primary *miR-125b-1* (*hsa pri miR-125b-1*) transcript (*Figure 8A*). Expression of *hsa primiR-125b-1* was induced in the adult fat body using the steroid inducible gene switch GAL4 driver *FB GS* (*Figure 8B*). Overexpression of *miR-125* in the adult fat body led to a 5.8% increase in median lifespan in the AL diet and a 13.6% increase in median lifespan in the DR diet (*Figure 8C* and *Figure 8—figure supplement 1A–B*). To confirm whether this increase in lifespan was not due to the effects of RU-486, we analyzed the effect of RU-486 in *+/+; hsa miR-125b-1/+* flies (*Figure 8—figure supplement 1* and *Figure 8—source data 1A–B*). While increasing *miR-125* levels in the fat tissue also led to increased stored triglyceride content in flies that were fed either AL or DR diets (*Figure 8D*), a significant increase in the diameter of the lipid droplets was only observed in *FB GS >UAS hsa pri miR-125b-1* flies that were fed a DR +RU-486 diet and no difference was observed in AL diet due to the smaller size of the lipid droplets in both AL- and AL + diets (*Figure 8E*). Finally, to test whether modulating *miR-125* levels led to changes in the expression of genes involved in fat metabolism, we examined the expression of *fasn1* and *fatp* in *FB GS >UAS hsa pri miR-125b-1* flies that were fed AL and DR in the presence and absence of RU-486. Consistent with the stronger increase in lifespan in DR +conditions, *FB GS >UAS hsa pri miR-125b-1* flies expressed higher levels of both *FASN1* and *FATP* in DR plus RU-486 conditions (*Figure 8G–H*). The levels of *FASN1* were also significantly higher in *FB GS >UAS hsa pri miR-125b-1* flies that were fed AL diet in presence of RU-486 (*Figure 8H*). While these data confirmed that artificially modulating the levels of *miR-125* in a tissue where its target was upregulated could result in changes in lifespan by regulation of the same downstream targets, inducing *miR-125* at different levels might be required to minimize the effects of other *miR-125* targets and maximize lifespan upon AL. Nevertheless, these experiments demonstrate that the regulation of fat metabolism by *miR-125* is an evolutionarily conserved mechanism.

## Discussion

### Regulatory RNAs as key regulators of metabolic homeostasis

In contrast to the protein machinery that represents only ~2% of the transcribed genome, the expansion of the noncoding transcriptome in higher eukaryotes reflects greater regulation of cellular processes through control of protein function (*ENCODE Project Consortium, 2012*). Our previous analysis uncovered a role for *let-7* and *miR-125* in aging (*Chawla et al., 2016*). This work was followed up by another study that reported that increasing *Drosophila let-7* levels in the adult nervous system enhanced lifespan and altered metabolism (*Gendron and Pletcher, 2017*). In the current study, we have uncovered a new role for the evolutionarily conserved *miR-125* in DR-dependent extension of lifespan. We utilized hypomorphic and genetic loss of function mutants to examine the

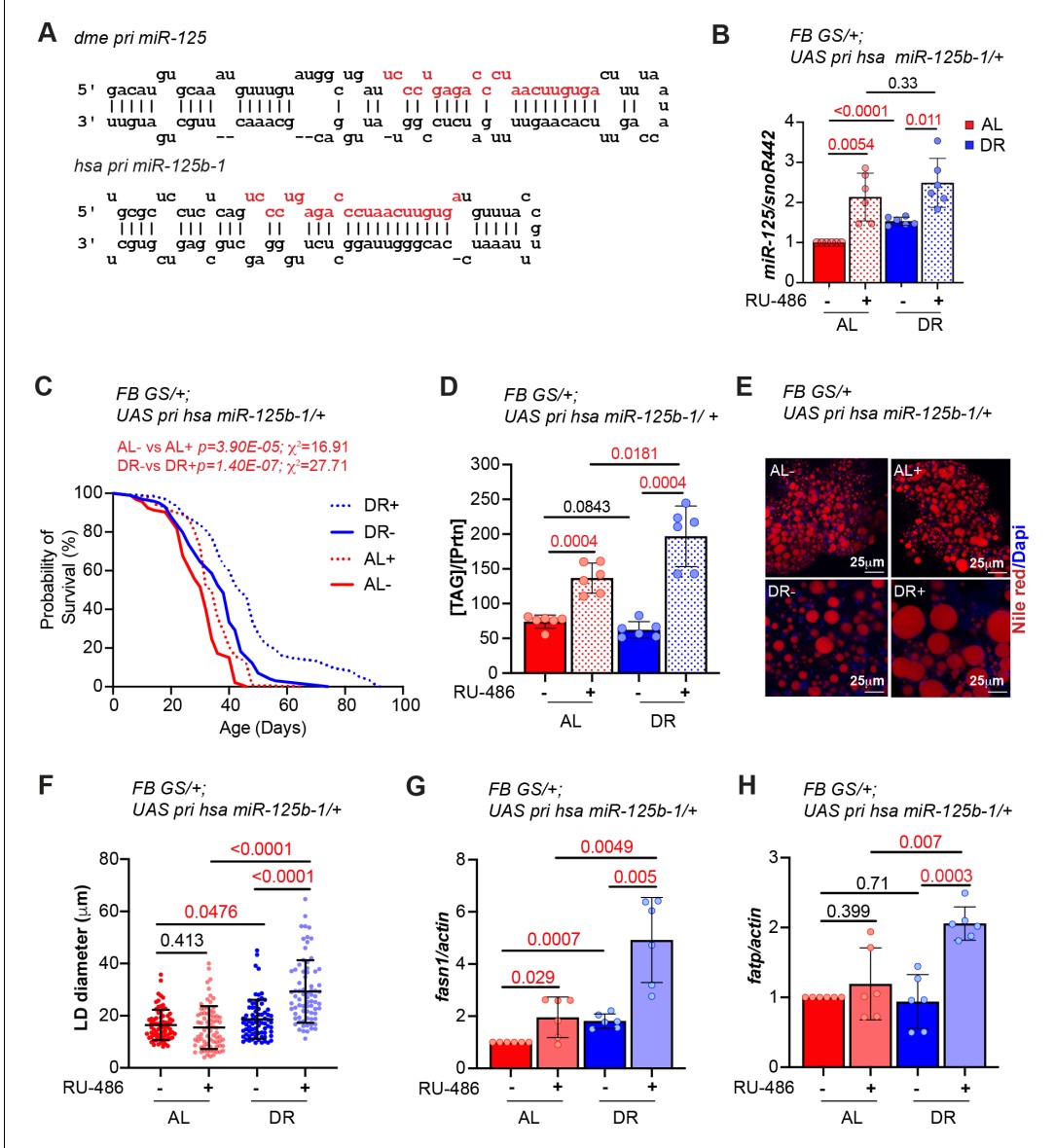

**Figure 8.** Overexpression of human primary *miR-125b-1* in the adult fat tissue promotes longevity and increases TAG levels under both AL and DR conditions. (A) Schematic of *Drosophila melanogaster* primary *miR-125* (*pri miR-125*) transcript and the *Homo sapiens* primary *miR-125b-1* (*pri hsa miR-125b-1*) indicating that the processed miRNA sequence in fruit flies and humans is identical. (B) Quantitative RT-PCR of *miR-125* from abdominal tissue of *FB GS >UAS chinmo* flies in presence of RU-486 (bars with red and blue pattern) or in absence of RU-486 (bars with solid red and blue color) in flies that were fed Ad libitum (AL) (red) or DR diet (blue) for 20 days. Expression levels were normalized to SnoR442. Values are mean ± SD, n = 6. (C) Overexpression of *hsa pri miR-125* increases lifespan in flies that are fed an AL or DR diet (Compare red and blue dotted lines with red and blue solid line) (See *Figure 8—source data 1A–B* for number of flies and median and maximum lifespan). The p values and $X^2$ values indicate that the increase in lifespan is greater under DR conditions. (D) *FB GS >UAS hsa pri miR-125b-1* flies store increased levels of triglycerides (TAG) when fed an AL or DR diet. Quantitation of TAG stored levels in -RU-486 and +RU-486 fed 20 d old *FBGS >UAS hsa pri miR-125b-1 flies.* (E) Fat bodies/abdomens of female flies were dissected and stained for the content and diameter of lipid droplet (LD) (red are lipid droplets stained with Nile red and blue is Dapi). Scale bar, 25 μm. (F) Quantitation of lipid droplet (LD) diameter in (E). Quantitation of 15 largest LDs in five samples per condition. Error bars represent ± SD. (G, H) Overexpression of *hsa pri miR-125* upregulates *fasn1* but not *fatp* under AL conditions. RT-PCR quantitation of mRNA levels of *fasn1* (G) and *fatp* (H) in the fat tissue of *FB GS >UAS hsa pri mir-125* flies that were fed an AL (red and pink bars) or DR diet (dark blue and light blue) under uninduced (red and dark blue bars) and induced (pink and light blue bars) conditions for 20 days. Expression levels were normalized to *Actin5c*. Values are mean ± SD, n ≥ 6. Genotypes of strains used in this figure: (B-H) *FBGS/+; UAS has pri miR-125b-1: w[1118]; P{w[+mW.hs]=Switch1}106/+; P{v+, UAS-hsa miR-125b-1} VK00033/+.*

The online version of this article includes the following source data and figure supplement(s) for figure 8:

**Source data 1.** Lifespan analysis of *FBGS >UAS hsa pri miR-125b-1* strain.

*Figure 8 continued on next page*

*Figure 8 continued*

**Source data 2.** Survival proportions of lifespan curves of strains used in *Figure 8C*.
**Figure supplement 1.** Effect of RU on lifespan of *UAS pri hsa miR-125b-1>*w$^{1118}$ flies.
**Figure supplement 1—source data 1.** Lifespan analysis of *UAS has pri miR-125b-1/+* strain.
**Figure supplement 1—source data 2.** Survival proportions of lifespan curve of strain used in *Figure 8—figure supplement 1*.

contribution of *miR-100*, *let-7*, and *miR-125* in the DR pathway. *Let-7-C$^{hyp}$* mutant strain expresses near wild-type levels of *let-7-C* miRNAs (*miR-100*, *let-7*, and *miR-125*) during development but displays an age-related decline in the levels of these miRNAs during adulthood (*Chawla et al., 2016*; *Chawla and Sokol, 2012*). Expression analysis of the *let-7-Complex* miRNAs in *let-7-C$^{hyp}$* revealed that DR-mediated upregulation of *let-7-C* miRNAs is required for lifespan extension by DR (*Figure 1E–F* and *Figure 1—figure supplement 2B–D*). Lifespan analysis of *Δlet-7* and *ΔmiR-125* flies uncovered a role for these miRNAs in DR-mediated lifespan extension. Thus, this is the first study that identifies conserved miRNAs that are regulated by DR in the *Drosophila* model and demonstrates a role for these two miRNAs in DR-mediated lifespan extension.

## Fat metabolism, DR, and the role of miRNAs

Lipid metabolism plays an important role in the aging process and pharmacological, dietary, and genetic interventions that extend lifespan often cause changes in lipid metabolism (*Barzilai et al., 2012*; *Johnson and Stolzing, 2019*). The adipose tissue has also been linked to metabolic dysfunction and age-related diseases such as heart attacks, stroke, hypertension, diabetes, and cancer (*Tchkonia et al., 2010*). The fat body exerts some of its effects by storage and release of fat under different contexts. A non-autonomous regulatory role for this tissue in aging by regulation of the miRNA biogenesis machinery was reported in the mouse model. This study showed that aging-associated decline in the levels of the miRNA biogenesis factor, Dicer results in the downregulation of several miRNAs including *miR-125b* in the adipose tissue (*Mori et al., 2012*). The authors further showed that knockout of Dicer specifically in the adipose tissue rendered the mice hypersensitive to oxidative stress. More importantly, this decline in multiple miRNAs in the adipose tissue was prevented by caloric restriction (CR). Although this study correlated the age and CR-mediated changes of *miR-125b* with its downstream target p53, the causal role of these changes on lifespan was not examined (*Mori et al., 2012*). Our genetic and molecular analysis in *Drosophila* independently identified *miR-125* as a downstream effector of DR and showed that upregulation of its human ortholog in the fat tissue was able to enhance lifespan in a diet-independent manner. Future work may be required to determine the optimum duration and dose of *miR-125* required to maximize healthspan in higher organisms, nevertheless our analysis provides several avenues that can be explored for testing this miRNA as a dietary restriction mimetic.

## Chinmo as a nutrient-dependent regulator of fat metabolism

Chinmo is a BTB (Bric-a-brac, Tramtrack, Broad complex) domain -Zinc finger (ZF) transcription factor that has been demonstrated to function as a repressor (*Flaherty et al., 2009*). This BTB-ZF protein has been shown to play a critical role in the temporal fate specification of mushroom body neurons during development (*Zhu et al., 2006*). However, more recent studies have highlighted a new role for this protein in the JAK/STAT signaling pathway that regulates stem-cell renewal in *Drosophila* (*Flaherty et al., 2010*). Interestingly, as a mediator JAK/STAT signaling, Chinmo exerts non-autonomous effects on the renewal of germline stem cells in *Drosophila* testis (*Flaherty et al., 2010*). However, the molecular mechanism or the non-autonomous signal that mediates these effects is currently not known. Our analysis indicates that in addition to previously documented transcriptional and post-transcriptional regulation, Chinmo may be reversibly modified post-translationally to play a role in the cytoplasm. Proteomic analysis of flies that ectopically expressed *chinmo* in the adult neurons identified several downregulated biological processes including fat metabolism. In addition, this analysis also identified upregulated processes such as cytoplasmic translation, macromolecule localization, retrograde vesicle-mediated transport, and intra Golgi vesicle-mediated transport, that may aid researchers to assign new cytoplasmic functional roles for this dosage-sensitive factor (*Figure 7—figure supplement 2*). In addition, we show that *chinmo* is transcriptionally regulated by

nutrition deprivation. A significant increase in *chinmo* mRNA levels was observed in flies that were exposed to DR or starvation conditions (*Figure 3* and *Figure 4—figure supplement 2*). These data reveal that Chinmo shares several attributes with the insulin/IGF signaling pathway—it is a nutrient-regulated factor that plays a critical role in the temporal fate specification of neurons during development. Silencing of *chinmo* in the adult brain by *miR-125* is required for normal aging (*Chawla et al., 2016*). More importantly, in this study, we show that reducing *chinmo* in the the adult fat body increases lifespan under AL conditions. Additionally, our results indicate that Chinmo may play a role in fat metabolism through its ability to circulate by loading onto exosomes. Lastly, another plausible hypothesis is that Chinmo exerts a feed-forward effect by inducing a signal that is relayed to the fat tissue to increase its expression in a non-autonomous manner. Future studies focused on the analysis of Chinmo and its interacting partners will likely uncover newer non-nuclear functional roles for Chinmo.

## Role of *miR-125* in aging and late-onset diseases

We previously showed that *miR-125* plays a role in regulating lifespan and maintenance of neuronal integrity (*Chawla et al., 2016*). In this study, we show that overexpression of the human *miR-125* in the adult fat tissue is sufficient to enhance lifespan (*Figure 8*). In addition to its role in modulating lifespan, *miR-125* has also been identified as a circulating diagnostic biomarker in Alzheimer's disease and Type 2 Diabetes (*Ortega et al., 2014*; *Tan et al., 2014*; *Villeneuve et al., 2010*). These and several other studies, highlight the importance of this conserved small RNA as a pro-longevity factor, biomarker and disease-modifier. However, the widespread expression of this miRNA in metazoans calls for a more detailed analysis of the tissue-specific effects of *miR-125* before it can be considered as a relevant therapeutic target for the treatment of age-related diseases.

Taken together, our analyses have identified *miR-125* as an effector of DR pathway. *miR-125* is regulated by dietary signals (DR) and represses *chinmo*, thus leading to de-repression of genes

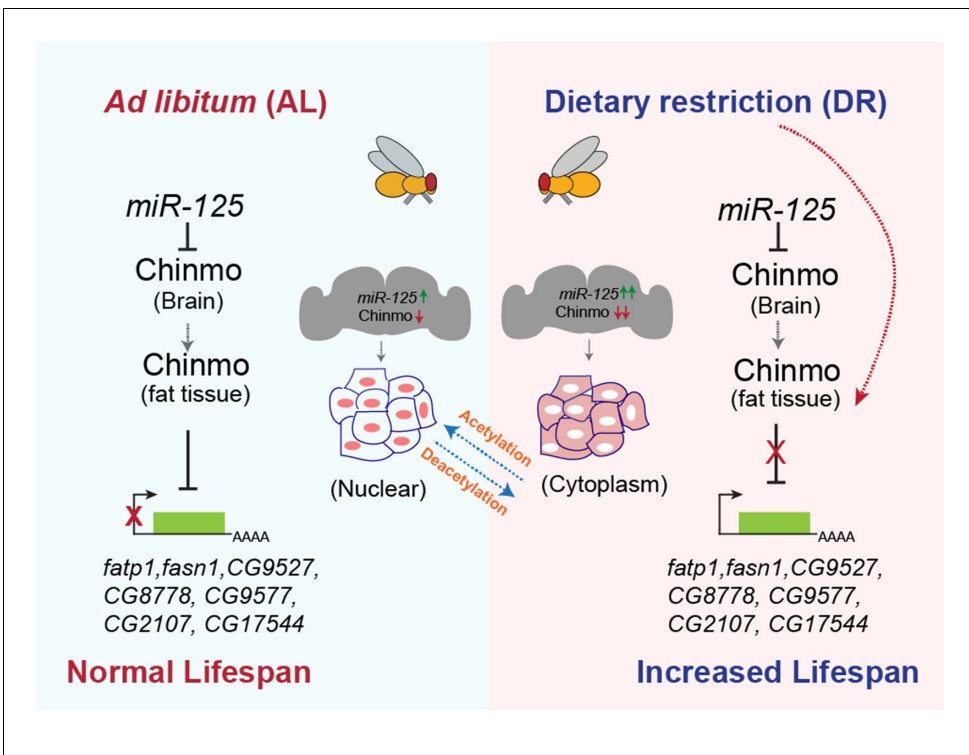

**Figure 9.** *miR-125* regulates DR-dependent lifespan extension by post-transcriptionally silencing *chinmo*. Proposed model summarizing the mechanism by which *miR-125* and *chinmo* regulate lifespan extension by DR. *miR-125* targets *chinmo* mRNA in the brain under AL and DR conditions. In the adult fat tissue, Chinmo transcriptionally represses genes involved in fat metabolism. DR-mediated cytoplasmic relocalization of Chinmo in the fat tissue relieves transcriptional repression of genes involved in fat metabolism, thus increasing lifespan.

involved in fat metabolism in peripheral tissues, which in turn result in the extension of lifespan. (*Figure 9*). This functional analysis sets the stage for evaluation of *miR-125* and other conserved miRNAs as candidates for developing therapeutics that promote healthy aging and prevent/delay late-onset diseases.

# Materials and methods

## Key resources table

| Reagent type (species) or resource | Designation | Source or reference | Identifiers | Additional information |
|---|---|---|---|---|
| Antibody | Anti-GFP(chicken) | Thermo Fisher Scientific | A10262, RRID:AB_2534023 | 1:2500/1:1000 |
| Antibody | Anti-Flag(mouse) | Sigma-Aldrich | F-1804, RRID:AB_262044 | 1: 2500 |
| Antibody | Anti-Fatp(rabbit) | Gift from B. Mollereau *Dourlen et al., 2012* | | 1:200 |
| Antibody | Anti-acetyl lysine | Cell Signaling | #9441; RRID:AB_331805 | 1:750 |
| Antibody | Anti-chinmo (rat) | Gift from Nick Sokol (*Wu et al., 2012*) | | 1:1000 |
| Antibody | Anti-Woc(rabbit) | Gift from Maurizio Gatti (*Raffa et al., 2005*) | | 1:1000 |
| Antibody | Anti-La(rabbit) | Gift from S.L. Wolin (*Yoo and Wolin, 1994*) | | 1:1000 |
| Antibody | Anti-tubulin (mouse) | Sigma-Aldrich | T9026, RRID:AB_477593 | 1:4000 |
| Antibody | Anti-dCsp1 (mouse) | DSHB | DSHB Cat# ab49, RRID:AB_2307340 | 1:1000 |
| Antibody | Anti-Syntaxin 1A (mouse) | DSHB | DSHB Cat# 8C3, RRID:AB_528484 | 1:500 |
| Antibody | Anti-HrsFL(Guinea Pig) | Gift from Hugo Bellen | | 1:2000 |
| Antibody | Anti-mouse HRP | Jackson Immuno Research, Laboratories, Inc | 115-035-003, RRID:AB_10015289 | 1: 2500 |
| Antibody | Anti-chicken HRP | Sigma-Aldrich | A9046, RRID:AB_258432 | 1: 2500 |
| Antibody | Anti-Rabbit HRP | Jackson Immuno Research, Laboratories, Inc | 111-035-003, RRID:AB_2313567 | 1: 2500 |
| Antibody | Alexa Fluor 488 Affinipure Donkey anti-chicken IgY | Jackson Immuno Research, Laboratories, Inc | 703-545-155, RRID:AB_2340375 | 1:1000 |
| Antibody | Alexa Fluor 488 Affinipure Donkey anti-Rat IgG | Jackson Immuno Research, Laboratories, Inc | 712-545-153, RRID:AB_2340684 | 1:1000 |
| Antibody | Alexa Fluor 488 Affinipure Donkey anti-Mouse IgG | Jackson Immuno Research, Laboratories, Inc | 712-545-151; RRID:AB_2340846 | 1:1000 |
| Antibody | Goat-anti-Rabbit IgG, Alexa Fluor 568 | Thermo Fisher Scientific | A-11036, RRID:AB_10563566 | 1:1000 |
| Antibody | Alexa Fluor 568 Goat-anti-Mouse IgG | Thermo Fisher Scientific | A-11031, RRID:AB_144696 | 1:1000 |
| Chemical compound, drug | Nile Red | Sigma-Aldrich | 72485 | 1:250 |

*Continued on next page*

*Continued*

| Reagent type (species) or resource | Designation | Source or reference | Identifiers | Additional information |
|---|---|---|---|---|
| Other | Normal goat serum | Jackson Immuno Research, Laboratories, Inc | 005-000-121, RRID:AB_2336990 | |
| Chemical compound, drug | Deacetylase inhibitor cocktail | ApexBio | K1017 | 1:100 |
| Chemical compound, drug | Phospho Safe Extraction reagent | Merck | 71296 | |
| Other | Anti-Flag M2-Affinity gel | Sigma-Aldrich | A2220, RRID:AB_10063035 | Resin |
| Peptide, recombinant protein | Flag peptide | Sigma-Aldrich | F4799 | |
| Chemical compound, drug | SYBR Premix Ex Taq | Takara Bio | RR420A | |
| Commercial assay or kit | Taqman Fast advanced mix | Thermo Fisher Scientific | 4444557 | |
| Sequence-based reagent | Taqman microRNA assay for miR-100 | Thermo Fisher Scientific | Assay ID 000287 | |
| Sequence-based reagent | Taqman probes for let-7–5 p | Thermo Fisher Scientific | Assay ID 000332 | |
| Sequence-based reagent | Taqman probes for miR-125–5 p | Thermo Fisher Scientific | Assay ID 000449 | |
| Sequence-based reagent | Taqman probes for Sno446 | Thermo Fisher Scientific | Assay ID 001742 | |
| Sequence-based reagent | Taqman probes for 2S rRNA | ThermFisher Scientific | Assay ID 001766 | |
| Other | HyClone CCM3 cell culture media | Cytiva | SH30061.01 | |
| Chemical compound, drug | RNAiso Plus | Takara Bio | 9109 | |
| Chemical compound, drug | Phenol: Chloroform | Sigma-Aldrich | P1944 | |
| Transfection construct | Lipofectamine 3000 Transfection Reagent | Thermo Fisher Scientific | L3000015 | |
| Other | DNAse I (RNAse-free) | New England BioLabs, Inc | NEB Catalog #: M0303 | Enzyme |
| Commercial assay or kit | Triglyceride reagent | Sigma-Aldrich | T2449 | |
| Commercial assay or kit | Free Glycerol reagent | Sigma-Aldrich | F6428 | |
| Recombinant DNA reagent | TopoXL cloning kit | Thermo Fisher Scientific | K804010 | |
| Commercial assay or kit | High-Capacity Reverse Transcription assay kit | Thermo Fisher Scientific | 4368814 | |
| Other | SUPERase In RNAse Inhibitor | Thermo Fisher Scientific | AM2696 | RNAase inhibitor |

*Continued on next page*

*Continued*

| Reagent type (species) or resource | Designation | Source or reference | Identifiers | Additional information |
|---|---|---|---|---|
| Other | dNTP set 100 mM solutions | Thermo Fisher Scientific | R0181 | dNTPs |
| Other | Superscript IV Reverse Transcriptase | Thermo Fisher Scientific | 18090050 | Enzyme |
| Other | Phusion High-Fidelity DNA Polymerase | New England BioLabs, Inc | NEB Catalog# M0530S | Enzyme |
| Recombinant DNA reagent | Quick Ligation Kit | New England BioLabs, Inc | NEB Catalog# M2200L | |
| Genetic reagent (*D. melanogaster*) | $w^{1118}$ | Bloomington *Drosophila* Stock Centre | | |
| Genetic reagent (*D. melanogaster*) | *let-7-C$^{GKl}$* | *Sokol et al., 2008* | | |
| Genetic reagent (*D. melanogaster*) | *let-7-C$^{KO2}$* | *Wu et al., 2012* | | |
| Genetic reagent (*D. melanogaster*) | *chinmo$^1$* | *Zhu et al., 2006* | | |
| Genetic reagent (*D. melanogaster*) | *P{w+, UAS-Chin::SV40}* | *Chawla et al., 2016* | | |
| Genetic reagent (*D. melanogaster*) | *P{w+, UAS-chinmo$^{RNAi\ 148}$} VK00033* | *Chawla et al., 2016* | | |
| Genetic reagent (*D. melanogaster*) | *P{w+, UAS-hsa miR-125b-1} VK00033* | | | |
| Genetic reagent (*D. melanogaster*) | *P{v+, let-7-C$^{\Delta miR-100}$}attP2* | *Chawla et al., 2016* | | |
| Genetic reagent (*D. melanogaster*) | *P{v+, let-7-C$^{\Delta let-7}$}attP2* | *Chawla et al., 2016* | | |
| Genetic reagent (*D. melanogaster*) | *P{v+, let-7-C$^{\Delta miR-125}$}attP2* | *Chawla et al., 2016* | | |
| Genetic reagent (*D. melanogaster*) | *P{w+, let-7-Cp$^{3.3kb}$::cDNA} VK00033* | *Chawla and Sokol, 2012* | | |
| Genetic reagent (*D. melanogaster*) | *3XelavGS* | Kind gift from Scott Pletcher | | |
| Genetic reagent (*D. melanogaster*) | *P{w+, UAS-FASN1$^{RNAi-1}$}attP2* | Bloomington Stock 35775 | | |
| Genetic reagent (*D. melanogaster*) | *P{w+, UAS-FATP$^{RNAi}$}attP2* | Bloomington Stock 55919 | | |
| Genetic reagent (*D. melanogaster*) | *P{w+, UAS-Flag FATP::SV40}* | This study (Chawla Lab) | | G. Chawla (Regional Centre for Biotechnology) |
| Genetic reagent (*D. melanogaster*) | *P{w+, UAS-Flag Chinmo::SV40}* | This study (Chawla Lab) | | G. Chawla (Regional Centre for Biotechnology) |
| Genetic reagent (*D. melanogaster*) | *FB-GS* | Bloomington Stock 8151 | | |
| Genetic reagent (*D. melanogaster*) | *5966 (Gut specific GS)* | Kind gift from David Walker | | |
| Genetic reagent (*D. melanogaster*) | *Da GS* | Kind gift from David Walker | | |

*Continued on next page*

*Continued*

| Reagent type (species) or resource | Designation | Source or reference | Identifiers | Additional information |
|---|---|---|---|---|
| Genetic reagent (*D. melanogaster*) | *UAS GFP* | Bloomington Stock 32198 | | |
| Sequence-based reagent | Primers for RT-PCRs and cloning, see **Supplementary file 2** | This study (Chawla Lab) | | G. Chawla (Regional Centre for Biotechnology) |
| Recombinant DNA reagent | UAS Flag Chinmo plasmid | This study (Chawla Lab) | | G. Chawla (Regional Centre for Biotechnology) |
| Recombinant DNA reagent | UAS GFP Chinmo plasmid | This study (Chawla Lab) | | G. Chawla (Regional Centre for Biotechnology) |
| Software, algorithm | OASIS | *Han et al., 2016* | https://sbi.postech.ac.kr/oasis/ | |
| Software, algorithm | GraphPad Prism | GraphPad | RRID:SCR_000306 | |
| Software, algorithm | GPS-PAIL 2 | *Deng et al., 2016*, *Li et al., 2006* | http://pail.biocuckoo.org | |

## *Drosophila melanogaster* stocks and husbandry

Fly stocks were maintained in standard cornmeal/agar medium (composition provided in the next section) at 25°C with a 12 hr light: 12 hr dark cycle in 60% humidity. Fresh fly food was prepared every 3 days to prevent desiccation. For steroid-mediated UAS-transgene control using the Gene-Switch driver, flies were fed a diet containing 200 µM RU-486 (Mifepristone, Cayman Chemicals, Ann Arbor MI). Unless otherwise noted, adult female flies of indicated ages were used for experiments. All experiments were performed in the *Drosophila* chamber Model DR-36VL (Percival Scientific, Inc, IA, USA). Detailed genotypes of all strains as well as the sources of the genetic mutations and transgenes used in the study are listed in *Figure 6—figure supplement 2* and *Figure 6—figure supplement 3*, respectively. Transgenesis was performed by the Fly Facility at Bangalore Life Science Cluster (CCAMP, Bengaluru, India). Stocks with multiple genetic elements were obtained by genetic recombination and/or crosses. The *let-7-C^{GKI}* mutation contains a 991-base-pair deletion that removes the *miR-100*, *let-7*, and *miR-125*. Additionally, the *let-7-C^{GKI}* mutation (*let-7-C* **G**AL4 **K**nock-**I**n) contains the GAL-4 and *white* coding sequence driven by the *let-7-C* promoter (*Sokol et al., 2008*). The *let-7-C^{KO2}* is identical except that the endogenous *let-7-C* locus was replaced with *white* rather than *white* and *gal4* (*Wu et al., 2012*). All UAS lines used in the study were backcrossed four times. The scheme for the generation of strains used in *Figures 1* and *2* is represented in *Figure 1—figure supplement 1*. The genetic scheme ensured that experimental and control lines had identical genetic backgrounds. Flies that were analyzed were trans-heterozygous for two different *let-7-C* null alleles, ensuring that phenotypes were not due to recessive mutations on either *let-7-C* mutant chromosome. In addition, third chromosomes that contained differing rescuing transgenes were derived in parallel from the same population of flies. Finally, all flies had a common X-chromosome, derived from an isogenized stock. All rescuing transgenes in *Figures 1* and *2*, including the wild-type rescuing transgene as well as, *let-7* and *miR-125* deleted versions, were injected into embryos from the same population of BL#25710. The resulting progeny were backcrossed twice to BL#32261 to select and balance *vermillion+* transformants. Single transformants were subsequently backcrossed to an isogenized version of BL#3703 three times to make balanced stocks with isogenized X chromosomes. Stocks with differing rescuing transgenes were crossed to the same population of a stock that contained the *let-7-C^{KO2}* chromosome, an isogenized X chromosome, and two 3rd chromosome balancers. The *let-7-C^{KO2}* stock was generated in a similar fashion as the rescuing transgenes stocks, by backcrossing three times to an isogenized version of BL#3703. The resulting stocks had common X, 2nd, and 3rd chromosomes and were used to generate the experimental strains. A second *let-7-C* allele, *let-7-C^{GKI}*, was prepared by outcrossing twice to an isogenized stock, and then crossed to an isogenized stock containing a T (2:3) Cyo-TM6b compound chromosome. The *let-7-C* allele was

selected based on mini-white, and the T (2:3) Cyo-TM6b balancer was selected based on the dominant Humoral marker. The resulting stock with a fixed second and third chromosome was amplified and used as the source for all virgins in the crosses that yielded the flies for analysis. For the generation of the *let-7-C* hypomorphic rescue and mutant line, the *let-7-C^Hyp* transgenic was combined with *let-7-C^KO2* stock. The remaining steps were similar to the scheme described (*Figure 1—figure supplement 1*). The genotypes of all the strains used in the study are indicated in the manuscript text, figure legends, and *Supplementary file 1*.

## Diets used in the study

### Cornmeal sugar medium (1L)

Cornmeal (80 g); Yeast extract (15 g); Sucrose (25.85 gm); Dextrose (51.65 gm); 8 g agar; methyl paraben (1 gm) in 5 ml Ethanol; 10 mL of Acid mix (Propionic acid: Orthophosophoric acid).

Ad libitum (AL) diet (1L): Yeast Extract (5%); Corn meal (8.6%); Sucrose (5%); Agar (0.46%); 10 ml of Acid Mix (Propionic acid: Orthophosophoric acid) (1%).

Dietary restriction (DR) diet (1L): Yeast Extract (0.25%); Corn meal (8.6%); Sucrose (5%); Agar (0.46%); 10 ml of Acid Mix (Propionic acid: Orthophosophoric acid) (1%).

Starvation medium: 1.5% agarose in Phosphate Buffered Saline (PBS).

## Survival analysis

Twenty female flies (1–3 days old) were transferred to each vial. Flies were transferred to fresh food every 3 days at which time dead flies were removed. Surviving flies were recorded every 2 days. The survival curves were plotted using Microsoft Excel. Statistical analysis was performed with the *Online Application for the Survival Analysis of lifespan assays* (OASIS) and GraphPad prism (*Han et al., 2016*; *Yang et al., 2011*), and the p-values were calculated using the log-rank (Mantel-cox) test and Cox proportional regression analysis. The number of flies used for each experiment was noted along with the median and maximum lifespans of the tested strains long with p values and $\psi^2$ values of the comparisons made in *Figure 1—source data 1*, *Figure 2—source data 1*, *Figure 3—source data 1*, *Figure 6—source data 1*, *Figure 7—source data 1*, *Figure 7—source data 2*, *Figure 7—source data 3*, *Figure 8—source data 1*, *Figure 8—source data 1*. Experiments usually included two independent controls: $w^{1118}$ as well as a *let-7-C* mutant strain containing a fully rescuing transgene. The $w^{1118}$ survival curve was generated with flies that had been backcrossed five times. To ensure that the DR phenotypes were not due to an effect of RU-486 the genetic background, survival analysis was performed with progeny from crosses between $w^{1118}$ and the relevant stocks used to generate the experimental and control strains used in *Figures 1*, *2*, *3*, *6*, *7* and *8* and have been represented in the respective figure supplements and source data.

## Plasmids and transgenes

The N-terminal Flag-Chinmo cDNA was amplified using primers 109/242 by using RE59755 plasmid (DGRC). The N-Terminal codon-optimized GFP-Chinmo was subcloned as an EcoRI-XbaI fragment by PCR amplification. Optimized GFP was amplified as an EcoRI-Xho1 fragment and cloned in frame with Chinmo using the primers 238/239 by using Addgene plasmid 26229 as a template (*Pfeiffer et al., 2010*). Chinmo cDNA was amplified using primers 241/242 on the RE59755 plasmid (DGRC). GFP-Chinmo mutants were generated by SOE-PCR (Primers listed in *Figure 6—figure supplement 2*). Primers were designed to mutate predicted lysine residues to Glutamine. The putative Lysine residues were identified by utilizing the GPS-PAIL two online tool (*Deng et al., 2016*; *Li et al., 2006*). The outer 5' end and 3' end primers for all the chinmo mutants were 241/242, and these PCRs were sub-cloned as XhoI-XbaI fragments in frame with Optimized GFP into pUAST attB plasmid.

The C-terminal Flag-tagged *FATP* cDNA was sub-cloned as a XhoI-XbaI fragment into pUAST attB using the primers 211 and 209 by using SD05207 plasmid (DGRC).

*Drosophila melanogaster* Sir2 cDNA was amplified from FI19438 plasmid available from the *Drosophila* Genomic Resource Center (DGRC) using primers 586/587. The PCR products were cloned into pENTR/D-TOPO (Thermo Fisher Scientific). Plasmids expressing flag and Myc-tagged versions of dSir2 were generated by recombining pENTR-dSir2 with pAFW and pAMW Gateway plasmids (T. Murphy, Carnegie Institute for Science, Baltimore; obtained from DGRC, Bloomington, IN),

respectively in an LR clonase enzyme (Thermo Fisher Scientific) reaction. The N-terminal Flag dSir2 was sub-cloned as a Not I-Xba I fragment into pUAST attB plasmid by PCR amplification with primers 322/323.

The *hsa mir-125b1* primary transcript was generated by annealing the oligo pair 1070/1071 and cloned into the XbaI site of pUAST attB. All PCRs were performed with High fidelity Phusion enzyme (Thermo Fisher Scientific, USA) and all clones were verified by sequencing. The primers used for cloning are listed in *Supplementary file 2*.

## RNA isolation and quantitative real-time PCR

Total RNA was extracted from whole fly or tissue samples using RNAiso Plus (Takara Bio, Inc). Animals/tissues were homogenized in 0.2 ml of RNAiso Plus with a micropestle (Tarsons) before extraction. The cDNA was generated by using a High-capacity cDNA reverse transcription Kit (Thermo Fisher Scientific, MA, USA). In each reaction, 0.5–1 µg was mixed with random hexamers, $MgCl_2$, 10X RT Buffer, dNTPs, RNAse Inhibitor and MultiScribe Reverse transcriptase in a 10 µl total volume. The cDNA synthesis was performed as per the manufacturer's protocol in a Bio-Rad C1000 Touch Thermal Cycler. The synthesized cDNA was diluted (1:10) and used as template for quantitative real-time PCR (qRT-PCR) using SYBR premix EX-Taq- plus (TaKara) and analyzed on QuantStudio 6 Real-Time PCR machine (Thermo Fisher Scientific, Waltham, Massachusetts, USA). The expression of the target genes was normalized to *actin-5C* or *rp49*. For expression analysis of miRNAs, Taqman miRNA assays (Thermo Fisher Scientific, Waltham, Massachusetts, USA) specific for the miRNA (*dme miR-100*, *dme-let-7* and *dme-miR-125*). Each cDNA sample was diluted 1:25 and real-time quantitative PCR (qPCR) was performed in duplicate using miRNA-specific primers/probe on a Quant six studio Real Time PCR System (Thermo Fisher Scientific, Waltham, Massachusetts, USA). For fold change analysis, individual values were normalized to 2S rRNA or Snu442 for Taqman miRNA assays and *rp49 or Actin 5* c levels for SYBR green assays. For qRT-PCR analysis, oligos 7, 8, 13, 14, 17, 18, 23, 24, 102a, 103a, 179, 180, 181, 182, 1183, 184, 185, 186, 187, 188, 189, 190, 191, 192, 193, 194, 3 and 4 listed in *Supplementary file 2* were used.

## Measurement of protein levels

Protein levels were quantitated using the Bio-Rad protein assay dye reagent concentrate (BIO-RAD, CA, USA) as per the manufacturer's instructions.

## Measurement of triglycerides levels

Triglyceride quantification was performed as described in *Tennessen et al., 2014*.

## Immunostaining and microscopy

Tissues were dissected in PBS and fixed for 30 min in 4% paraformaldehyde. Immunofluorescence was performed as described previously (*Chawla and Sokol, 2012*; *Wu et al., 2012*). Primary antibodies included rat anti-Chinmo (*Wu et al., 2012*), guinea pig anti-Chinmo (gift from Nick Sokol), rabbit anti-Woc (*Raffa et al., 2005*) (gift from Maurizio Gatti 1:1000), rabbit anti-La (gift from S. L. Wolin), mouse anti-Fmr1 (DSHB), mouse anti-Flag (Sigma-Aldrich) and, mouse anti-Dachshund (DSHB). For the staining of lipids, fruit flies were fed AL or DR diets for 20 days and fat bodies were dissected in 1X PBS. Tissues were fixed in 4% PFA for 30 min and three washes of 5 min each were given in 1X PBS + 0.3% Triton. Tissues were stained for 1 hr with Nile Red Staining solution (1 mg/mL in Acetone, 1:250 PBST) and DAPI (1:100 PBST). Samples were washed three times with PBST for 5 min and mounted. For immunostaining of transfected cells, poly-L-Lysine coated coverslips were placed in 6-well plates 48 hr after transfection. After 24 hr incubation, the cells were fixed for 30 min in 4% paraformaldehyde followed by permeabilization in 1X PBS + 0.3% Triton. Cells were blocked for 1 hr and incubated with primary antibody for 12 hr at 4°C. Primary antibodies were used at the following concentration: anti-chicken GFP (1: 2500), mouse anti-Flag (1:2500; Sigma-Aldrich), mouse anti-Fmr1 (1:100; DSHB). The coverslips were rinsed with PBS followed by incubation with Alexa-conjugated secondary antibodies for 1 hr at room temperature. Following incubation, the coverslips were rinsed with PBS and stained with DAPI for 15 min followed by three washes and mounted. Slides were analyzed under Leica SP8 Confocal Microscope and lipid droplet size was measured using Fiji/ImageJ software and graphs were plotted with GraphPad Prism v9. Confocal stacks were

merged using Leica LAS software. Statistical analysis and histograms were generated using Graph-Pad Prism software. p Values were calculated using a two-tailed unpaired t test with Welch's correction. Values are presented as mean ± SD.

For quantitating Chinmo levels in nuclear and peripheral regions, pixel intensity of 10 individual cells in images of five independent dissected abdomens stained with anti-Chinmo antibodies was quantified using ImageJ software. The expression of Chinmo (pixel intensity) was quantitated by measuring the same area between samples of a set (Nuclear or Periphery). Samples whose staining was directly compared were prepared and imaged in parallel and under identical conditions. Slides were analyzed under Leica SP8 Confocal Microscope (Microscopy facility, Regional Centre for Biotechnology, India). Images of abdominal fat tissue were optically sectioned in 1 µm increments keeping the number of sections constant for each set. The image projections were collated from four to six sections and confocal stacks were merged using Leica LAS X software and all images were acquired at the same Laser intensity for a given set. Each experiment was performed three times. Statistical analysis was performed, and histograms generated using Graph-Pad Prism software. p Values were calculated using a two-tailed unpaired t-test with Welch's correction. Values are presented as mean ± SD.

## Exosome isolation

Kc-167 cells were cultured in T75 Flasks in the serum-free CCM3 medium. The cells were transfected with Tubulin-GAL4 and Flag Chinmo pUASTattB at a density of $1.5 \times 10^6$ cells/ml with lipofectamine 3000 (Thermos Fisher Scientific) according to the manufacturer's instruction. After 72 hr of transfection, the cells were starved for 8 hr by culturing in 2 mg/ml Glucose in Phosphate buffered saline (1X PBS). The medium from both control (fed) and starved conditions was collected and concentrated using Amicon Ultra −15 Centrifugal Filter Unit, Ultracel, 3 KDa, 15 ml (Millipore). Exosomes were isolated by size exclusion chromatography with qEV single/35 nm IZON Columns (IZON Science) according to the manufacturer's instructions. The eluted fractions were collected and an indicative elution profile for exosome particles and proteins was obtained by monitoring Absorbance at 600 nm and 280 nm, respectively. Western blot of collected fractions was performed with anti-Syntaxin 1A (1:500; DSHB), anti-Csp (1:1000; DSHB), anti-HrsFL (1:20,000; kindly provided by H. Bellen), and Flag antibody (1:2500; Sigma Aldrich). Western blot analysis of different fractions revealed that the elution of vesicles peaked 200–400 µl after void volume (1 ml) and these fractions were pooled and used for the RNA or protein analysis.

## Immunoprecipitation and western blot analysis

Kc167 cells were cultured in T75 flasks at a density of $1.5 \times 10^6$ cells/ml, transfected with Tubulin Gal4 (15 µg) along with other UAS (15 µg) and/or Gateway expression plasmids (20 µg) along with 50 µL Lipofectamine 3000 per flask and harvested after 72 hr. Transfected Kc167 cells were lysed (Phospho Safe Extraction buffer; Merck) with Protease Inhibitors (Sigma-Aldrich) and Deacetylation inhibitor cocktail (ApexBio). The lysates were sonicated and centrifuged, and the resulting lysate was incubated with 12.5 µL of anti-Flag M2 affinity gel (Sigma) at 4°C with nutation for 3 hr. Beads were washed three times with IP150 buffer (50 mM Tris HCl, pH 7.4, with 150 mM NaCl) and bound proteins were eluted by incubating with 150 ng/mL of 3X FLAG peptide (Sigma) for 30 min at 4°C with nutation. The eluate was then used for Western blot analysis. Western blots were probed with primary antibodies, including anti-Flag antibody (1: 2500; Sigma), anti-Myc antibody (1: 2000; Abcam), anti-Tubulin antibody (1: 4000; Sigma), anti-GFP (1: 1000; Thermo Fisher Scientific), anti-acetyl lysine (1: 750; Cell Signaling). Primary antibodies were detected with HRP-conjugated secondary antibodies (1: 2500; Jackson Immunoresearch Laboratories, Inc), and the chemiluminescent signal was detected using a GE Healthcare LAS 4000 (Central Instrumentation Facility, Regional Centre for Biotechnology).

## Proteomics analysis

Age-matched mated female flies were fed +or + RU-486 food for 5 or 10 days. Protein lysate was prepared by freezing female flies (n = 5 per replicate). Flies were homogenized in 100 µL of Ammonium bicarbonate buffer (100 mM ammonium bicarbonate and 8M urea) after adding appropriate amount of 100X Protease inhibitor cocktail (Sigma). The samples were centrifuged for 15 min at

maximum speed at 4°C in a microfuge and the supernatant was collected, quantitated and processed for Proteomic analysis.

## Trypsin digestion and desalting

Proteins from control and 5 days treated samples were precipitated using 4 volumes of chilled acetone. Precipitated proteins were washed thrice with chilled acetone and dried at room temperature. Precipitates were reconstituted in 100 mM Ammonium bicarbonate and treated with 5 mM DTT for 30 min at 55°C. Reduced disulfide bonds were alkylated using 10 mM Iodoacetamide for 30 min at 55°C. Proteins were digested using MS grade trypsin (Promega Trypsin Gold) in a ratio of 1:50 for 12 hr. After digestion, peptides were dried in a SpeedVac vacuum dryer and reconstituted in 2% Acetonitrile (ACN) and 0.1% Formic acid (FA). The reconstituted peptides were cleaned and desalted using a C18 cartridge (Oasis HLB 1cc). Samples were eluted in 0.1%FA and 50% ACN from the cartridge after desalting with 0.1% FA in water. The eluted peptides were dried in a SpeedVac vacuum dryer and reconstituted in 15 µL of 2% ACN and 0.1%FA.

## LC-ESI MS/MS analysis

Digested peptides were loaded onto a CapTrap C18 trap cartridge Cap-Trap C18 trap cartridge (Michrom Bioresources, Auburn, CA, USA) and desalted for 10 min at the rate of 10 µl/min using 2% ACN and 0.05% trifluoroacetic acid (TFA) in water using Eksigent NanoLC 400. Labeled peptides were separated on Chromolith Caprod RP-18e HR capillary column (150 × 0.1 mm; Merck Millipore) using linear gradient of buffer B (98% ACN and 0.05% TFA in water) in buffer A (2% ACN and 0.05% TFA in water). Peptides were eluted at the rate of 300 nL/min, which were directed to the Sciex 5600 TripleTOF System for MS and MS/MS analysis. MS spectra were acquired from 350 Da to 1250 Da, and the peptides were fragmented and MS/MS spectra acquired using DDA method. In brief, 25 most intense peaks were fragmented using collision-induced dissociation (CID) with rolling collision energy in each cycle. The MS/MS spectra was acquired from 100 Da to 1600 Da.

## MS/MS data analysis

The MS/MS spectra search to identify peptides was performed using the FASTA database of protein sequences from *D. melanogaster*, available at UniProtKB website (https://www.uniprot.org/taxonomy/7227, proteome ID UP000000803). In order to avoid misidentifications, most common contaminants such as human keratin and porcine trypsin were also included in the database FASTA file. The MS/MS peak list data files were analyzed by Mascot ion search engine version 2.3.02, with one fixed modification; carbamidomethylation of cysteine (monoisotopic mass of 57.0215 Da), two variable modifications; methionine oxidation (monoisotopic mass of 15.9949 Da), deamidation (monoisotopic mass of 0.984 Da) and a peptide and MS/MS fragment ion mass tolerance of 0.1 Da. Up to two missed cleavages were allowed along with Mascot automatic decoy database search. The MASCOT DAT files from all fractions of each biological replica of 5 days control (1C1, 1C2, and 1C3) and treated (1E1, 1E2, and 1E3) samples were merged during the search and output file was analyzed using Scaffold Q + S version 4.9.0 (Proteome Software, Portland, OR, USA) to generate a full report of proteomic data, which is provided in *Supplementary files 3* and *4*. Protein identification validation was performed by Scaffold parameters including Mascot ion scores of 30 or higher (for +2, +3, and +4 charges), a minimum of two identified peptides, 100 ppm of parent mass tolerance, 95% peptide identification probability and 95% protein identification probability, (using the Scaffold Local FDR algorithm), resulting in a 0.0% decoy-based FDR. Protein fold change values were calculated on the basis of Normalized Spectral Abundance Factor (NSAF), calculated for each protein. Proteins with fold change values less than or equal to 0.5 were down-regulated and the proteins with fold change values more than or equal to two were up-regulated. Quantitative differences were statistically analysed by a t-test and volcano plot (*Figure 7E*), where differences with p-values lower than 0.05 were considered statistically significant. A two-tailed t test with equal variance between samples was performed to calculate the p-value. In *Figure 6E*, values were log transformed and INF (Infinity) values were changed to four and zero values to −4 (representing +and or over and under expression). Identified proteins were categorized according to gene ontology terms using STRING, an online database of known and predicted protein-protein interactions. Validation of the identified proteins, spectral abundance values and comparative analysis between control and test samples was

done using Scaffold. GO functional categories assigned to the identified proteins have been represented in (*Figure 7F* and *Supplementary files 3* and *4*).

## Statistical analysis

Data representation and statistical analysis were performed using GraphPad Prism eight software and/or Microsoft Excel. Survival curves were compared using log-rank tests, with Bonferroni corrections for p values where multiple comparisons were necessary. Cox proportional Hazard regression analysis was also performed using the Online Application for the Survival Analysis of Lifespan (OASIS) online tool {20 #216}. All survival and lifespan graphs show one representative experiment out of two or three independent repeats with two to three cohorts of 20 female flies per genotype. A two-tailed t test with Welch's correction was used to analyze data in *Figure 1B–G*, *Figure 1—figure supplement 1B-D*, *Figure 2G–H*, *Figure 3B–F,H,J*, *Figure 3—figure supplement 1B–G*, *Figure 4C,E,I–L,N–P*, *Figure 4—figure supplement 2B–C,E–F,G–I*, *Figure 6F–M*, *Figure 6—figure supplement 1E,G*, *Figure 6—figure supplement 3D–E*, *Figure 7A–C,G–H*, *Figure 7—figure supplement 1A–B,D–E*, *Figure 7—figure supplement 3D–E*, *Figure 8B,D,F,G,H*. Other details on statistical analysis can be found in Figure legends. Statistical significance was set at p<0.05. p Values are denoted in red for statistically significant comparisons and in black for comparisons that were not statistically significant.

## Acknowledgements

This work was supported by the DBT/Wellcome Trust India Alliance Fellowship/Grant [grant number IA/I(S)/17/1/503085] awarded to GC. The proteomics analysis was performed at the Mass Spectrometry Facility of the Advanced Technology Platform Centre (ATPC) (Grant No. BT.MED-II/ATPC/BSC/01/2010). AKY was supported by DBT-Big Data Initiative grant (BT/PR16456/BID/7/624/2016) and Translational Research Program (TRP) at THSTI funded by DBT. The authors acknowledge the advice and inputs of Dr. Nirpendra Singh for the proteomics analysis. Fly Facility at Bangalore Life Science Cluster is acknowledged for microinjection of the transgenic line constructs. The author's thank Dr. Arthur Luhur for microinjection of the *hsa miR-125b-1* construct and Jennifer Beck for lifespan analysis to test the effect of RU-486 on 3X ElavGS flies in AL and DR. The authors thank Drs. Scott Pletcher and David Walker and Bloomington *Drosophila* Stock Center (NIH P40OD018537) for fly stocks, *Drosophila* Genomics Resource Center (NIH 2P40OD010949) for plasmids and cell line, Dr. M Gatti for Woc antibody, Dr. SL Wolin for La antibody, Dr. H Bellen for Hrs antibody and Dr. Bertrand Mollereau for Fatp antibody.

## Additional information

### Funding

| Funder | Grant reference number | Author |
| --- | --- | --- |
| Wellcome-Trust DBT India Alliance | IA/I(S)/17/1/503085 | Geetanjali Chawla |

The funders had no role in study design, data collection and interpretation, or the decision to submit the work for publication.

### Author contributions

Manish Pandey, Sakshi Bansal, Sudipta Bar, Investigation, Writing - review and editing; Amit Kumar Yadav, Formal analysis, Statistical analysis, proteomic analysis and motif analysis; Nicholas S Sokol, Resources, Writing - review and editing; Jason M Tennessen, Resources, Investigation; Pankaj Kapahi, Resources, Investigation, Writing - review and editing; Geetanjali Chawla, Conceptualization, Resources, Formal analysis, Supervision, Funding acquisition, Validation, Investigation, Methodology, Writing - original draft, Project administration, Writing - review and editing

## Author ORCIDs

Amit Kumar Yadav (iD) http://orcid.org/0000-0002-9445-8156
Jason M Tennessen (iD) http://orcid.org/0000-0002-3527-5683
Geetanjali Chawla (iD) https://orcid.org/0000-0003-0354-3716

## Decision letter and Author response

Decision letter https://doi.org/10.7554/eLife.62621.sa1
Author response https://doi.org/10.7554/eLife.62621.sa2

## Additional files

### Supplementary files

- Supplementary file 1. Genotypes used in the study.
- Supplementary file 2. Primers used in the study.
- Supplementary file 3. Downregulated biological process.
- Supplementary file 4. Upregulated biological process.
- Transparent reporting form

### Data availability

All data generated or analyzed during this study are included in the manuscript and supporting files. Source data files have been provided for Figures 1, 2, 3, 4, 5, 6, 7, 8. Proteomics analysis data done in Figure 7 is also provided in Supplementary files 3 and 4 (Upregulated and downregulated processes).

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
