## [Decision Letter]

**Acceptance summary:**

Pandey et al., propose that miR-125 and let-7 mediate the lifespan and metabolic response to dietary-restriction via the transcription factor Chinmo in fruit flies. They propose that the enhanced turnover of fatty acids is the critical metabolic adaptation for lifespan extension under DR. These results are novel and connect beneficial systemic effects to the role of a single transcription factor.

**Decision letter after peer review:**

[Editors’ note: the authors submitted for reconsideration following the decision after peer review. What follows is the decision letter after the first round of review.]

Thank you for submitting your work entitled "miR-125-chinmo pathway regulates dietary restriction dependent enhancement of lifespan in *Drosophila*" for consideration by *eLife*. Your article has been reviewed by 3 peer reviewers, and the evaluation has been overseen by a Reviewing Editor and a Senior Editor. The following individual involved in review of your submission has agreed to reveal their identity: Adam Dobson (Reviewer #2).

Our decision has been reached after consultation between the reviewers. Based on these discussions and the individual reviews below, we regret to inform you that your work will not be considered further for publication in *eLife*.

Your work investigates the role of regulatory miRNAs and their targets in mediating the effects of dietary restriction on lifespan, proposing a model where neuronal miR125 represses chinmo expression in both neurons and (somehow) peripheral tissues. Unfortunately, all reviewers raised several issues on the way the experiments were conducted, on the analyses and on the conclusions. It appears that the conclusions and the model are not entirely supported by the data presented. For instance, despite the claim in figure 5, there seems to be no clear causality in the action of chinmo in the fat bodies to regulate lifespan via lipid metabolism. Methodologically, this work would have benefited from more explaining and controls. For instance, the mutant/transgenic lines do not appear to have been back-crossed and there are missing controls (e.g. GAL4- and UAS-alone). The GeneSwitch drivers seem to have ubiquitous expression. Very importantly, Chinmo regulation by miR-125 is not demonstrated in this work, but it is rather assumed to occur based on cited work. Lifespan comparisons are not sufficiently analysed statistically (Cox proportional hazards should be computed). Some of the results seem to contradict the main model, rather than supporting it (see reviewer 2, point 4).

Together, there are too many issues that need to be addressed to make this

manuscript acceptable.

*Reviewer #1:*

The manuscript by Pandey et al., examines the role for microRNAs in the fruit fly's response to dietary restriction. This is a very interesting topic. The authors propose that miR-125 and let-7 mediate the lifespan and metabolic response to DR in the fruit fly. Based on further functional characterization, the authors believe that miR-125 acts though its known target, Chinmo, with effects on both fat metabolism and aging.

While the study is on a very interesting topic, I do not think it's performed to a sufficiently high standard to warrant publication in *eLife*. My key objections (noted below) concern the way the experiments were conducted, as well as their interpretation and presentation. I believe these are too extensive to be addressed by the authors in a timely manner.

1) The key experimental tool used is fly genetics. Unfortunately, based on the information available in the manuscript I do not have the confidence that the genetic analysis was conducted to a sufficiently good level, and this impacts most if not all of the results presented. It appears that none of the mutants/transgenes were backcrossed, and this is very important for studies of aging in flies. Because the genotypes used would have been derived from multiple strains and backgrounds, I am not persuaded that the effects observed are not due to variation in the background. This is specifically relevant for all experiments where flies of different genotypes are compared (not so much for GeneSwitch experiments). Additionally, the genotypes of the experimental flies are very poorly described in the figures/captions and text.

2) I believe there is a number of essential controls missing. Examples include: Figure 2A – here the DR response appears compared to a response in a complexly different experiment – a control should always be run in parallel. Additionally, GAL4-alone and the UAS-alone controls appear not presented (was a GAL4 used to drive the RNAi line in Figure 2 – it's not clear from Sup Table 7 – if not, how is it being driven?). These are essential. Similarly, driver-alone and UAS-alone controls (with/without RU486) should also be run at least once for each GeneSwitch experiment – it's now well established that RU486 alone can sometimes trigger changes in physiology causing artefacts. As these RU486-induced artifacts appear to be context dependent, the controls need to be run by the authors in their experimental system.

3) GeneSwitch drivers are described as having one tissue specificity but then shown to be expressed in other tissues. This is confusing. References for articles describing the drivers are missing for some of the drivers.

4) The regulation of Chinmo by miR-125 is inferred from previous experiments. I could not find any data showing that it occurs in the context of DR in the submitted manuscript.

5) Figure 3 – the ectopic expression of Chinmo is essentially making the flies very sick – how much the lack of DR response in such flies can be interpreted is questionable.

6) Figure 1G – the p value is noted as not significant but the value in Sup Table 1B is, in two experiments for let-7. For miR-125 it's significant 1 out of 2 times.

7) Is the AL regime used by the authors essentially a cholesterol deficient diet (see Q Wu et al., 2020 Aging cell 19 (3), e13120)? If so, how broadly relevant will it be to DR in other organisms?

8) The authors propose that Chinmo can diffuse from one tissue to another – as a transcription factor I think it's very unlikely to do this. I think such a claim should be substantiated with more evidence.

*Reviewer #2:*

This manuscript investigates how dietary restriction extends lifespan in *Drosophila*, and specifically the role of regulatory RNAs and the BTB transcription factor chinmo. The roles of specific tissues and lipid metabolism are also investigated.

There are some good data here and some interesting findings. However these findings are held back by dubious analyses, datasets which appear to contradict the conclusions, and some unjustified logical leaps.

1) The physiological relevance of chinmo is unclear. A leap to studying chinmo is made based on a claim (unsubstantiated by either reference to literature or data) that chinmo is a direct target of the let7C complex. The model is that chinmo repression is required for DR to extend lifespan, and loss of the miRNAs stops this repression. But to really know this, we need to see chinmo levels in WT and miRNA mutants, subjected to AL and DR. And if the miRNAs regulate chinmo at post-translational level, we need to see protein levels by Western blot and IF (using the antibody used in Figure 4?). It would also be nice to have this information specifically in the tissues studied. Currently we have only (A) mRNA levels of chinmo in whole DR flies (in which DR increases chinmo expression, contrary to the concluding model), and (B) IFs in figure 4 which don't appear to show any effect of DR on chinmo levels. A much more convincing case for studying chinmo needs to be made. The genetics in the lifespans might do this (subject to genetics, below), but they follow way later in the paper.

2) Statistical analysis requires considerable work. The model proposed requires evidence from relatively complex epistasis experiments. Some of these data are present, but it is insufficient to say that the magnitude of lifespan extension in one condition is less than another – is it significantly so? More rigorous analysis exploring interactions of experimental factors (e.g. Cox proportional hazards) is required.

3) Numerous conclusions are unsupported. To conclude the epistasis claimed, we'd need to see statistical interactions (i.e. without A, there is no effect of B), but a number of effects look additive (i.e. A affects this thing, and in parallel so does B). E.g. Figure 3F, it looks like chinmo affects mean TAG by a set amount against a baseline set by DR. To really conclude that chinmo was mediating DR, the effect of DR should be blocked (or strongly diminished) when chinmo is manipulated transgenically. We see this in Figure 5, but rarely elsewhere. I encourage the authors to apply proper statistical modelling, testing for interaction effects and specifying pairwise comparisons, for ALL experiments testing epistatic or gene:diet effects.

4) Some figures in fact show results which are contrary to the concluding model. E.g. in Figure 6J: if there is an effect of FATP-RNAi in both AL and DR, does it not argue that FATP is in fact irrelevant for the DR effect? Similarly, the hypothesis tested in 6K is that elevated expression of lipid metabolism genes should create a DR-like state, in which case we expect overexpression to increase AL lifespan but not DR lifespan. However, we see the opposite: FATP expression extends DR lifespan, but not AL lifespan. Altogether, this actually argues against the model that fat metabolism genes in the FB mediate DR's effects.

5) The manuscript would also be more penetrable without jumping so much between tissues, and a more clear explanation of where chinmo and the miRNAs are normally expressed. Probably the most convincing data are in Figure 5, where we have a manipulation of physiological expression patterns (i.e. chinmo RNAi) yielding the sort of interaction patterns we need to conclude epistasis. But then figure 6 jumps back to over-expressing chinmo in neurons. Flyatlas has the expression of chinmo high in neurons and negligible in fat body, and the miRNAs high in the nervous system. No convincing case is made for expressing a neuronal miRNA in the fat body (Figure 7). I am also very cautious in interpreting protein overexpression with a pan-neuronal driver such as ElavGS – brains are complex and this manipulation will force chinmo expression in a lot of neurons where it may never normally be expressed, causing potentially odd effects. This brings the physiological relevance of a number of manipulations into question.

6) The way non-autonomous effects of chinmo are shown is quite nice. But some statements in this section are quite surprising, especially that mRNA can diffuse between tissues. This statement would require quite a lot more data to substantiate. A more parsimonious explanation is a feed-forward loop in which chinmo expression in one tissue promotes its expression elsewhere through a secondary signal.

7) Insufficient genetic information. This is notoriously important when looking at epistasis in lifespan. Number of rounds of backcrossing must be given. The data are not useful if mutants were not freshly backcrossed before recombination. We also are not told whether mutants were heterozygous or homozygous. The Figure 1B qRT-PCR data also don't strongly support the claim that the Let7C mutant is actually a hypomorph – expression levels are not dramatically altered?

8) number of key points are made without reference to literature or data e.g. chinmo is a target of Let7C, Let7C miRNAs are normally expressed in the nervous system.

*Reviewer #3:*

Pandey and colleagues are studying how miRNAs and their targets regulate the extension of lifespan upon dietary restriction (reduced protein content in diet, termed DR). Previous work from the lab of one of the authors has shown that lifespan extension under DR can be prevented when triglyceride synthesis or breakdown is inhibited (Katewa, 2012, 2016). These observations have led to the current model in the field that the enhanced turnover of fatty acids is the critical metabolic adaptation for lifespan extension under DR. In the current study, the authors use established stocks containing deletions and rescue constructs of the let-7-C locus, which contains 3 miRNAs (miR100, miR125 and let-7) and examine lifespan and lipid droplets (fatty acids) in adult *Drosophila*. Ultimately, their data lead them to propose a model in which neuronal miR125 represses chinmo expression in both neurons and (somehow) peripheral tissues; the reduction of chinmo in peripheral tissue leads to increased expression of genes encoding fatty acid metabolism, and this leads to DR-induced lifespan extension. Unfortunately, the current manuscript is lacking critical data to support the model. Additionally, controls are missing and key Gal4 drivers are not lineage-traced, increasing the confusion about the cell autonomy of miR125 and chinmo. The text of the manuscript is lacking sufficient rationale for many experiments, and important introductory facts (let-C-7 locus biology, chinmo biology, DR model and biology and how DR compared to caloric restriction) are either relegated to the discussion or entirely lacking from the manuscript. The unclear and confusing autotomy of chinmo activity greatly reduce enthusiasm. This manuscript could be improved by a large number of new experiments and an editorial overhaul.

Dr. Chawla previously showed that adult flies lacking an upstream regulatory region in the let-7-C locus but containing all three miRNAs (called let-7-Chyp) had a significantly shorter lifespan (Chawla, 2016). In the current study, they find that the shorter lifespan of these flies cannot be extended by DR, in contrast to flies possessing the regulatory let-7-C sequences. This observation forms the foundation for the study. Flies lacking miR125 or let-7 similarly do not extend lifespan under DR. They authors decide to follow up on the miR125 observation. The Sokol lab has previously shown that both miR125 and let-7 degrade transcripts for chinmo, which encodes a putative transcriptional repressor that has only been observed in the nucleus of cells that express it. Indeed, miR125 null flies that are heterozygous for a null chinmo allele now display extended lifespan upon DR and have enlarged LDs. [The experiments using the let-7-C-Gal4 to deplete chinmo in Figure 2 are difficult to interpret because the Gal4 is not regulated by RU486 and is active through the life of the fly, leading to depletion of chinmo throughout development with consequences of defective neuronal patterning (Zhu, Cell 2006 and many other papers from Tzumin Lee and other groups)]. These data suggest that the maintenance of chinmo in miR125 null flies underlies the lack of life extension under DR. In Figure 3, they try to determine in which tissue ectopic expression of chinmo decreases lifespan under DR. They used the gene switch inducible system for these experiments, only to find that ectopic expression of chinmo under Da, 5966 or FB leads to early mortality. In fact, even with the Elav neuronal driver, lifespan is shorter than the AL, no-RU-468 control. I find it difficult to logically connect the relatively early death caused by Chinmo mis-expression with a suppression of lifespan extension under DR. The authors need to find another way to test their hypothesis. In Figure 3, they show that Elav>chinmo under RU-486 leads to a decrease in the size of lipid droplets, which they conclude connects neuronal chinmo to fat body metabolism in peripheral tissues. In Figure 4, they show the expression patterns of the GS Gal4 drivers, and because they see chinmo protein in tissues where the driver is supposedly not expressed, they conclude that chinmo RNA or protein is able to diffuse from neuronal cells to other tissues to induce metabolic changes. This is potentially an interesting model, but there are absolutely no data to support it. First, they need perform both real-time and lineage tracing of these drivers using G-trace. [G-trace should also be used for let-7-C-Gal4]. If these experiments support their hypothesis, then they would need to test how the movement of chinmo RNA or protein between tissues could occur – for example, by exosomes? In Figure 5, they deplete chinmo from fat body using FB GS driver. What is the hypothesis being tested here? It was not clear to me. If neuronal chinmo RNA or protein travels from neurons to fat body, then how would depletion of chinmo from fat body alter this movement? These needs to be explained in the text. Let us assume that the chinmo transcripts being depleted in Figure 5 are produced by fat body cells (and not neurons), what is the prediction of this on lifespan extension under DR, TAG levels and lipid droplet size? In my mind, the upregulation of miR125 upon DR leads to the downregulation of chinmo message and protein in neuronal cells. This effect is somehow transmitted to fat body cells, which alter lipid metabolism to promote lifespan extension. If chinmo levels in fat body were depleted, there should be no effect on DR-induced lifespan extension and no change in TAG levels and no reduction in LD size. Indeed, my predictions are borne out by the data. There is no further increase in lifespan extension under DR when chinmo is depleted from fat body (Figure 5B blue solid and dashed lines); there is no difference in TAG levels (Figure 5C, blue solid and hashed bars) and LD size is not reduced (Figure 5E, light and dark blue dots). Yet the authors write in the title to Figure 5 and in lines 280-290 something to the effect that reducing chinmo levels in the adult fat body increases lifespan and enhances lipid metabolism. Would the authors please explain in the text the predictions for this experiment and the results? Their interpretation and mine are not congruous. In Figure 6, they perform mass spec on adult females in which chinmo has been inducibly mis-expressed in neurons. They find 40 proteins differentially expressed between the induced and uninduced samples, 7 upregulated and 33 downregulated. Of the latter, 7 proteins are involved in fat metabolism and their mRNAs decrease 10 days after induction of chinmo. Do the authors have an earlier time point? If chinmo does directly repress these 7 genes, it presumably would do so in a shorter time frame, for example 1-2 days after induction. 10 days is a long time-frame for gene expression and may represent indirect regulation, and if the authors saw an effect after 2 days, this would be a more meaningful result. Do these genes have chinmo binding sites in regulatory regions? It would also be helpful to validate decreased expression of proteins using antibodies or enhancer traps to characterized genes, for example, FATP for which both antibodies and enhancer traps exist. They then test the functional role for FASN1 and FATP in DR-dependent lifespan extension. Once again, it would be helpful for the authors to explain in the text the predictions for this experiment. To my mind, if downregulation of chinmo in the fat body is important for DR-induced lifespan extension and if chinmo represses the fasn1 and fatp genes, then loss of chinmo should in increase FASN1 and FATP expression and this should aid in fatty acid turnover. Depletion of fasn1 and fatp should then decrease DR-induced lifespan extension, which is confirmed by the results (Figure 6I and J), and mis-expression of these factors should increase this extension, which again is supported by the results (Figure 6L). These are important results for the paper but are insufficiently explained in the text by the authors. In Figure 7, they express human miR125 in fat body and examine lifespan extension. Once again, the rationale for these experiments is totally unclear to me. The model introduced earlier in the paper is that neuronal (not fat body) miR125 extends lifespan under DR. Why did they perform the experiments in Figure 7? How do these results add to the paper? To me, they detract from the paper. The authors need to definitely establish in which cells miR125 is acting and in which cells chinmo is being regulated. Figure 8 is a model of the result, but it does not clarify any of the confusion about the autonomy of chinmo activity.

Figure 1B are missing the appropriate control, which is the "rescue" genotype. *w^1118^* is not the right control for this experiment because it does not contain the let-7-C alleles and the attP-inserted transgene. This is important because it forms the foundation for the entire study.

Figure 2A,B are missing controls (δ miR125 alone and chinmo1/+ alone). Furthermore, the driver in Figure 2A is let-7-CGal4, which is not a GS gal4 and therefore is active through development. The correct experiment here would be elavGS. This should be remedied.

Figure 2C and D – the AD and DR TAG profiles should be displayed on the same graph. Why are they separated? Why are is the Y-axis scale so different (0-7 on the AD and 0-50 on the DR) between AL and DR? I can't determine what happens to TAG levels under DR in the various genotypes. This is a significant issue for me. This experiment needs to be performed again and plotted on the same graph.

Figure 2G-J – why is the chinmo protein not monitored here? Why is dilp6 being monitored? The authors do not discuss the rationale for examining dilp6 or foxo; they need to tell the reader why they are doing this experiment. In panel H, what do the third and fourth pair of bars represent? In other words, why are + head and -head being shown twice? In panel J, why is starvation (absence of protein) introduce here? What is the rationale for this experiment and why is it being shown?

Figure 3A: DaGS>UAS-Chinmo flies with RU-486 all day within ~12 days of adulthood, much earlier ectopic flies without drug treatment. These data indicate that there is toxicity/lethality associated with ectopic chinmo, which obviates any possible connection to lifespan extension.

Figure 4 – please supply monochrome (white on black) all of these. It is very difficult for the human brain to appreciate green on black.

The authors need to better explain the various let-7-C reagents that they use both in the legend to Fig, 1 and in the Materials and Method. In particular, the let-C-7GKI allele is not on Flybase and I had to look up three prior pages to figure this out. Please make the manuscript easier to read and to be appreciated by the reviewer.

---

## [Author Response]

[Editors’ note: The authors appealed the original decision. What follows is the authors’ response to the first round of review.]

Reviewer #1:The manuscript by Pandey et al., examines the role for microRNAs in the fruit fly's response to dietary restriction. This is a very interesting topic. The authors propose that miR-125 and let-7 mediate the lifespan and metabolic response to DR in the fruit fly. Based on further functional characterization, the authors believe that miR-125 acts though its known target, Chinmo, with effects on both fat metabolism and aging.While the study is on a very interesting topic, I do not think it's performed to a sufficiently high standard to warrant publication in eLife. My key objections (noted below) concern the way the experiments were conducted, as well as their interpretation and presentation. I believe these are too extensive to be addressed by the authors in a timely manner.1) The key experimental tool used is fly genetics. Unfortunately, based on the information available in the manuscript I do not have the confidence that the genetic analysis was conducted to a sufficiently good level, and this impacts most if not all of the results presented. It appears that none of the mutants/transgenes were backcrossed, and this is very important for studies of aging in flies. Because the genotypes used would have been derived from multiple strains and backgrounds, I am not persuaded that the effects observed are not due to variation in the background. This is specifically relevant for all experiments where flies of different genotypes are compared (not so much for GeneSwitch experiments). Additionally, the genotypes of the experimental flies are very poorly described in the figures/captions and text.

We agree that controlled genetic background is essential for genetic research and our approach ensured such a controlled genetic background. Moreover, since the genetic approach was previously published by us (Chawla et al., 2016), we did not include the crossing scheme in this submission but noted the reference in the text. To make our approach as transparent as possible, we have included our entire genetic scheme in the revised version as Figure 1—figure supplement 1. As the scheme indicates, our approach relies on the comparison between strains with two different rescuing transgenes in a background as close to identical as possible. This procedure and the strategy should clarify that the strains we have analyzed have a common genetic background. The attached scheme also includes a description of the balancer strains used. The same isogenized balancer stock was used in multiple crosses, thereby ensuring that there was no genetic variability introduced as part of the strain construction.

The use of such rescuing transgenes is a "gold standard" in genetic research and all our rescuing transgenes were generated by injecting them into the same population of flies and once recovered, resulting strains were treated identically and in parallel. In addition, it is not clear whether the recommendation of backcrossing the transgenes/mutants will solve the issue: after backcrossing the strains, we would still need to follow the 8- generation scheme (F1 to F8) in Figure 1 figure supplement-1 to generate the final strains and spontaneous background mutations could be introduced and fixed in the population at any of these steps. As an example of a study with a similar genetic strategy as ours, please see Verma et al., Nature Neuroscience 2015. Additionally, the same scheme was used by us and published earlier (Chawla et al., 2016).

We have included the genotypes of the experimental flies in the revised figure legends and text. Additionally, the manuscript Supplementary File 1 has been retained in the revised version to enable readers to review a compiled list of all strains used in this manuscript.

2) I believe there is a number of essential controls missing. Examples include: Figure 2A – here the DR response appears compared to a response in a complexly different experiment – a control should always be run in parallel.

We acknowledge that genetically similar controls are essential for any functional analyses, and these were run in parallel with the experiment but were represented as Supplementary Figure 1A-B and Supplementary Tables 2A and 2B and referenced in the main text in the previously submitted version. This data is now included as panels in Figure 2C, 2E and Figure 2-source data 1-2 of the revised manuscript. The genotypes of the controls and the experimental strains are now included in the figure legends. The only difference between the genetic background of the control and experimental strains is the *let-7-Complex* rescuing transgenes that is *let-7-C^wild type^* (Ctrl) or *let-7-C^ΔmiR-125^* (Exp).

Additionally, GAL4-alone and the UAS-alone controls appear not presented (was a GAL4 used to drive the RNAi line in Figure 2 – it's not clear from Sup Table 7 – if not, how is it being driven?). These are essential.

We thank Reviewer 1 for pointing out the missing controls and agree that GAL4-alone and UAS-alone controls are essential. The lifespan for GAL4-alone (*let-7-C GAL4*) is represented as Figure 1—figure supplement 2F and Figure 1—figure supplement 2-source data 1-2 of the revised manuscript. The survival analysis of *UAS chinmo^RNAi^* is represented as Figure 2—figure supplement 1A and Figure 2—figure supplement 1-source data 1-2.

The *let-7-C* GAL4 (*let-7-C^GKI^*) was used to drive expression of the *UAS chinmo^RNAi^* and this was noted in line 124 of the previously submitted manuscript. The following information regarding *let-7-C GAL4* is included in the Materials and methods section of the revised manuscript (Lines 1173-1176). The *let-7-C^GKI^* mutation contains a 991-basepair deletion that removes the *miR-100*, *let-7*, and *miR-125*. Additionally, the *let-7-C^GKI^* mutation (*let-7-C GAL4 Knock-In*) contains the GAL4 coding sequence driven by *let-7-C* promoter (Sokol et al., 2008).

Similarly, driver-alone and UAS-alone controls (with/without RU486) should also be run at least once for each GeneSwitch experiment – it's now well established that RU486 alone can sometimes trigger changes in physiology causing artefacts. As these RU486-induced artifacts appear to be context dependent, the controls need to be run by the authors in their experimental system.

We agree with Reviewer 1 that RU-486 alone can sometimes trigger changes in physiology causing artifacts. The driver-alone and UAS-alone controls (with/without RU486) for *UASchinmo* (Figure 6—figure supplement 1B and Figure 6—figure supplement 1-source data 1), *UAS chinmo^RNAi^
*(Figure 2—figure supplement 1A and Figure 2—figure supplement 1-source data 1-2), *UAS Flag fatp* (Figure 7—figure supplement 3C and Figure 7-source data-3), *UAS fatp^RNAi^
*(Figure 7—figure supplement 3B and Figure 7-source data 2C-D), *UAS fasn1^RNAi^
*(Figure 7 —figure supplement 3A and Figure 7-source data-1C-D), *UAS Flag chinmo* (Figure 6—figure supplement 1C and Figure 6—figure supplement 1source data 1C-D) and, *UAS hsa pri miR-125* (Figure 8—figure supplement 1 and Figure 8—figure supplement 1-source data 1-2) have been included in the revised manuscript. We have cited the references for Gene switch driver strains (*S_1_106*) that have been previously tested and reported to have no survival effects in presence of RU-486 (Huang et al., 2014; Balukbasi et al., 2017). The effect of RU-486 on the lifespan of *3X ElavGS* strain in AL and DR was tested in this study and the survival curve has been included in Supplementary Figure 6—figure supplement 1A and Figure 6—figure supplement 1-source data-1). The *DaGS* and *5966 GS* have also been tested before but we have removed the data from the revised manuscript to retain the focus on the *miR-125* regulatory axis in the brain and fat tissue.

3) GeneSwitch drivers are described as having one tissue specificity but then shown to be expressed in other tissues. This is confusing. References for articles describing the drivers are missing for some of the drivers.

We apologize for the confusion with regards to Chinmo non-autonomous expression. We have revised the text so that the non-autonomous expression is not attributed to nonspecific expression of the GeneSwitch drivers. For clarity, the references for articles describing the GeneSwitch GAL4 drivers are included in the main text of the revised manuscript. We have removed data pertaining to *DaGS* and *5966* GeneSwitch drivers to add clarity. The specificity of the two GeneSwitch drivers used in the revised manuscript (*FBGS* and *3X ElavGS*) was validated by setting up crosses with UAS GFP and analyzing GFP expression in the brain and abdominal fat body of the progeny (Figure 6—figure supplement 2 A-H and 2A’-H’). As expected, *3X ElavGS>UAS GFP* flies expressed GFP specifically in the adult neurons and *FBGS> UAS GFP* flies expressed GFP specifically in the fat body in an inducible manner. Consistent with the GFP expression, Chinmo was specifically expressed in the fat body in *FBGS> UAS chinmo* flies upon induction with RU486. However, Chinmo was detected in the fat body in addition to being expressed in neurons in *3X ElaGS >UAS chinmo* flies (Figure 6—figure supplement 2I-P and 2I’-P’). We attribute this to a non-autonomous expression of Chinmo and have presented some data in support of our hypothesis.

4) The regulation of Chinmo by miR-125 is inferred from previous experiments. I could not find any data showing that it occurs in the context of DR in the submitted manuscript.

The regulation of *chinmo* by *miR-125* in flies that were exposed to AL and DR diet for 5 days (adult brain and fat body) is now represented in Figure 2B of the current manuscript. In addition, a schematic showing differential regulation of *chinmo* by *let-7* and *miR-125* in our previously published study is represented as figure 2A (Chawla et al., 2016).

5) Figure 3 – the ectopic expression of Chinmo is essentially making the flies very sick – how much the lack of DR response in such flies can be interpreted is questionable.

We agree with Reviewer 1 that overexpression of c*hinmo* makes the flies sick. This is expected for proteins that play dosage-sensitive essential roles like Chinmo. We believe that it was important to analyze the consequences of ectopic expression of *chinmo* to assess whether its derepression in adult brain mimicked *miR-125* mutant phenotypes and to confirm that *chinmo* was the functionally relevant target that was mediating the DR phenotype. Moreover, it was important to identify the tissues where this silencing of *chinmo* was critical for DR-mediated lifespan extension and hence an inducible GeneSwitch Gal4 system seems most appropriate. The other approach that we utilized was to examine whether reducing chinmo dosage by c*hinmo^RNAi^* or genetic mutation (*chinmo^1^*) in *miR-125* mutants was able to rescue *miR-125* mutant DR phenotype (presented in Figure 2). We have edited figure 3 (current Figure 6 of the revised manuscript) and removed the analysis done with *DaGS* and *5966* GeneSwitch drivers for clarity. The figure now includes an analysis of ectopic expression of *chinmo* and *flag chinmo* in adult neurons (*3X ElavGS*) and in adult fat tissue (*S_1_106*). Comparison of survival curves of flies that overexpress Chinmo or flag Chinmo in adult neurons (Chinmo: AL+ vs DR+ p=0.0066 and Χ^2^= 7.37; Flag Chinmo p=0.0068 and Χ^2^=7.34) with flies that overexpress Chinmo or Flag Chinmo in adult fat tissue (Chinmo: AL+ vs DR+ p=0.0283 and Χ^2^= 4.81; Flag Chinmo p=0.9472 and Χ^2^=0.0043) indicates that overexpression of this protein in the adult fat tissue dampens the DR dependent increase in lifespan more significantly (Figure 6B-E). The p-values and Χ^2^ values indicate that there is a significant difference between survival curves in AL and DR conditions. These experiments confirm that deregulation of *chinmo* is the cause of *miR-125* mutant DR phenotypes. Analysis of tissue-specific inducible expression of Chinmo was also important to uncover the non-autonomous effect of *chinmo* in fat tissue in *3X ElavGS >UAS chinm*o flies.

6) Figure 1G – the p value is noted as not significant but the value in Sup Table 1B is, in two experiments for let-7. For miR-125 it's significant 1 out of 2 times.

We thank the reviewer for pointing out the typographical error and apologize for missing the error in Figure 1. We have corrected the p-values and have noted the Χ^2^ values in all the figure panels with survival analysis. We have corrected all typographical errors and the source data tables have information regarding the maximum and median lifespan as well as the number of flies. Analysis was reperformed with OASIS and GraphPad Prism and corrected.

7) Is the AL regime used by the authors essentially a cholesterol deficient diet (see Q Wu et al., 2020 Aging cell 19 (3), e13120)? If so, how broadly relevant will it be to DR in other organisms?

The AL regimen that we used in this study is a cholesterol-deficient diet and the AL and DR diets essentially vary in the yeast extract (protein source). Protein restriction and more specifically amino acid restriction have been reported to significantly enhance lifespan in yeast and flies (Mirisola et al., 2014; Wei et al., 2008; Mair et al., 2005; Minor et al., 2010). In rodents, restriction of protein intake or of specific amino acids is associated with improved healthspan and increased lifespan (Mirzaei et al., 2014; Fontana et al., 2010; Brandhorst and Longo., 2019; Lopez-Torres and Barja., 2008). In humans, ongoing and future nutritional studies are required for the definition of what constitutes a healthy diet with pro-longevity benefits, however, short-term clinical trials and epidemiological studies indicate that protein restriction leads to reduced activity of the GH/IGF-1 regulatory axis, thus leading to a delay in age-related pathologies (Levine et al., 2014). From all these studies it is reasonable to conclude that healthspan effects observed with calorie restriction can at least partially be explained by the restriction of protein or more specifically amino acids. However, we thank Reviewer 1 for referring Wu et al., 2020, and agree that including cholesterol in the dietary regimen might yield additional conserved miRNA-mediated DR pathways that we would like to investigate in future studies.

8) The authors propose that Chinmo can diffuse from one tissue to another – as a transcription factor I think it's very unlikely to do this. I think such a claim should be substantiated with more evidence.

We agree with Reviewer 1 that more evidence is required to demonstrate that Chinmo is not predominantly functioning as a transcription factor in the nucleus. Cytoplasmic translocation of Chinmo would be a prerequisite for any model that claims a nonautonomous role for this protein. We agree with the need for a better explanation in support of experiments performed. Hence, we would like to highlight some new data that we have included in the revised manuscript as new figures (Figure 4 and Figure 5) and other panels that strengthen the hypothesis for non-autonomous effects of Chinmo and other non-nuclear roles for this protein.

i. Chinmo redistributes to the cytoplasm upon nutrient deprivation.

To ascertain whether Chinmo translocates to the cytoplasm upon nutritional deprivation we made use of a *Drosophila* cell line and tagged versions of Chinmo (GFP and Flag). Kc167 is an embryonic cell line that expresses very low levels of Chinmo (modEncode data). GFP-tagged Chinmo translocates to the cytoplasm in starved Kc167 cells that are transfected with Tubulin Gal4 and UAS Chinmo::GFP (Figure 4A). Consistent with the GFP-tagged Chinmo data, starved Kc167 cells expressing Flag-tagged Chinmo also show cytoplasmic redistribution of Chinmo (Figure 5C-H). Moreover, this cytoplasmic redistribution of Chinmo upon starvation was reversible, and shifting the cells back to normal medium (16h) resulted in reappearance of nuclear-localized Chinmo and disappearance of cytoplasmic protein (Figure 5I-N). Since, UAS-chinmo constructs were used in these experiments, we could conclude that nutrient-dependent post-translational modification of the protein was the mechanism involved in redistribution of the protein to the cytoplasm upon nutrient deprivation.

ii. Deacetylation of Chinmo by dSir2 is required for its nuclear export upon nutrient deprivation.

Increasing the cellular levels of dSir2 led to redistribution of the exogenous GFP-tagged Chinmo to the cytoplasm in fed conditions (Figure 4B). Thus, indicating that nutrient dependent post-translational modification of Chinmo by dSir2 is required for its export from the nucleus. Since, dSir2 is a nutrient-dependent deacetylase, we performed mutational analysis of Chinmo and identified three Lysine (K47, K55, and K348) residues that were critical for relocalization of Chinmo by dSir2 upon nutrient deprivation. This analysis is represented in Figure 4 and Figure 4—figure supplement 1 of the revised manuscript. These data confirm that dSir2 mediated deacetylation of Chinmo is required for its nuclear export.

iii. Chinmo fractionates with exosomes upon starvation.

Since exosomes are well-known players in intercellular communication and have been shown to carry protein, mRNAs and miRNAs, we examined whether Chinmo is loaded onto these extracellular vesicles in response to nutrient deprivation. To test whether Chinmo protein is loaded onto exosomes, we purified exosomes from Kc167 cells expressing flag-tagged Chinmo by size exclusion chromatography. Kc167 cells expressing Flag-Chinmo were starved for 8h (2mg/ml glucose in PBS) and exosomes were purified from cell culture medium and starvation medium. Western blot analysis was performed with the cell lysate, concentrated medium, and purified exosomes from Kc167 cells expressing Flag Chinmo for detection of exosomal proteins and Flag-Chinmo. Flag-Chinmo was detected in medium and exosome fraction of starved Kc167 cells but not in the concentrated medium and exosomal fraction of cells that were in normal medium (Figure 5O). Together these data provide evidence that Chinmo translocates to the cytoplasm and is loaded onto exosomes upon nutrient deprivation, and this might be a plausible mechanism for its non-autonomous expression (Figure 5P). A similar nutrient dependent translocation has been previously shown for the predominantly nuclear localized protein LC3/Atg8a (Huang et al., 2014).

iv. DR induces c*hinmo* mRNA levels in the hemolymph.

Hemolymph is responsible for circulating nutrients and exosomes. *Chinmo* mRNA levels increased by 2.6 ± 0.19-fold in the hemolymph of *w^1118^* adult flies that were fed a DR diet for 10 days (Figure 3E). A DR-dependent increase in the levels of circulating *chinmo* mRNA hints at a possible mechanism by which *chinmo* mRNA could exert nonautonomous effects.

v. Nutrient deprivation leads to an increase in the non-nuclear Chinmo in the fat tissue of *w^1118^*.

Quantitation of Chinmo protein levels in fat tissue of adult *w^1118^* flies exposed to AL and DR flies indicates that Chinmo protein levels increased in the non-nuclear region of the cells in flies that were fed a DR diet (Figure 3A-C and Figure 3—figure supplement1A-E). These data provide *in vivo* evidence for nutrient-dependent relocalization of Chinmo.

vi. Neuronal Chinmo upregulation induces Chinmo expression non-autonomously in the fat tissue.

Quantitative RT-PCR analysis of *chinmo* RNA in the head and dissected abdominal fat tissue of *3X ElavGS> UAS chinmo* flies and *FBGS> UAS chinmo* flies that were exposed to RU-486 for 10 days was performed. In the brain tissue of *3X ElavGS> UAS chinmo* flies, *chinmo* mRNA levels increased by 4.25 ± 0.8-fold under AL conditions and by 11 ± 1.96-fold in DR conditions upon addition of the ligand (Figure 6J). In addition, *chinmo mRNA* levels also increased by 8.5 ± 1.79-fold in AL conditions and by 7.6 ± 1.36-fold in DR conditions in the dissected fat tissue upon addition of the ligand (Figure 6K). These data suggested that *chinmo* mRNA was being induced non-autonomously to a similar magnitude in the abdominal fat tissue in *3X ElavGS> UAS chinmo* flies. To confirm whether the increase in mRNA also resulted in a non-autonomous increase in Chinmo protein levels, we performed immunostaining of dissected brain and abdominal fat tissue of *3X ElavGS> UAS chinmo* and *FBGS>UAS chinmo* flies (Figure 6—figure supplement 2). Consistent with the *chinmo* mRNA analysis, *3X Elav GS > UAS chinmo* flies that were fed RU-486 containing food displayed expression of Chinmo in the adult brain and adult fat body. However, *FB GS > UAS chinmo* flies that were fed RU-486 containing food displayed expression of Chinmo specifically in the fat body and no signal was detected in dissected brains (Figure 6—figure supplement 2I’-P’). These data were recapitulated in *3X ElavGS > UAS Flag chinmo* flies by quantitating the Chinmo protein expression (Figure 6—figure supplement 3A-E). Together these data highlight the existence of mechanisms for the non-autonomous effects of Chinmo. We hypothesize that Chinmo protein and/or mRNA is either capable of loading onto exosomes to facilitate communication with other peripheral tissues or a feed-forward mechanism exists by which upregulation of *chinmo* in the adult neurons promotes its expression in the fat tissue *via* an unidentified signal.

Reviewer #2:This manuscript investigates how dietary restriction extends lifespan in *Drosophila*, and specifically the role of regulatory RNAs and the BTB transcription factor chinmo. The roles of specific tissues and lipid metabolism are also investigated.There are some good data here and some interesting findings. However these findings are held back by dubious analyses, datasets which appear to contradict the conclusions, and some unjustified logical leaps.1) The physiological relevance of chinmo is unclear. A leap to studying chinmo is made based on a claim (unsubstantiated by either reference to literature or data) that chinmo is a direct target of the let7C complex. The model is that chinmo repression is required for DR to extend lifespan, and loss of the miRNAs stops this repression. But to really know this, we need to see chinmo levels in WT and miRNA mutants, subjected to AL and DR. And if the miRNAs regulate chinmo at post-translational level, we need to see protein levels by Western blot and IF (using the antibody used in Figure 4?). It would also be nice to have this information specifically in the tissues studied. Currently we have only (A) mRNA levels of chinmo in whole DR flies (in which DR increases chinmo expression, contrary to the concluding model), and (B) IFs in figure 4 which don't appear to show any effect of DR on chinmo levels. A much more convincing case for studying chinmo needs to be made. The genetics in the lifespans might do this (subject to genetics, below), but they follow way later in the paper.

We are not in complete agreement with the reviewer regarding the unsubstantiated claims and had referenced our previously published study where we showed that *chinmo* is differentially regulated by *miR-125* and *let-7* along with three other references in Lines 126-127 of the previously submitted manuscript (Chawla et al., 2016; Wu, Chawla, and Sokol, 2020; Wu, Chen, Mercer, and Sokol, 2012; Zhu et al., 2006). To further increase clarity regarding the targeting of c*hinmo* by *miR-125*, we have included a panel (Figure 2A) representing a schematic of the previously published study (Chawla et al., 2016). In addition, we show the immunostaining of Chinmo in the rescue and *miR-125* mutant dissected brain and fat body under AL and DR conditions (Figure 2B). The antibody used does not recognize denatured Chinmo protein and hence we and other labs that have used this antibody have not had success with this antibody in detecting Chinmo by western blot analysis.

2) Statistical analysis requires considerable work. The model proposed requires evidence from relatively complex epistasis experiments. Some of these data are present, but it is insufficient to say that the magnitude of lifespan extension in one condition is less than another – is it significantly so? More rigorous analysis exploring interactions of experimental factors (e.g. Cox proportional hazards) is required.

The statistical analysis for all the lifespan data has been repeated by performing Log Rank analysis using OASIS and GraphPad Prism. In addition to the p-values, we have included the Χ^2^ values to aid in the direct comparisons between different conditions. Both p values as well Χ^2^ values are represented in the Figure panels and Figure supplement tables for all the lifespan curves in the revised manuscript. Cox proportional analysis has been performed for experiments where diet and ligand interactions were tested and the p values are noted in the Figure table supplements.

3) Numerous conclusions are unsupported. To conclude the epistasis claimed, we'd need to see statistical interactions (i.e. without A, there is no effect of B), but a number of effects look additive (i.e. A affects this thing, and in parallel so does B). E.g. Figure 3F, it looks like chinmo affects mean TAG by a set amount against a baseline set by DR. To really conclude that chinmo was mediating DR, the effect of DR should be blocked (or strongly diminished) when chinmo is manipulated transgenically. We see this in Figure 5, but rarely elsewhere. I encourage the authors to apply proper statistical modelling, testing for interaction effects and specifying pairwise comparisons, for ALL experiments testing epistatic or gene:diet effects.

We agree with the reviewer that the effects of overexpression of *chinmo* can be seen in both AL and DR. However, this is not unexpected and may be due to other DR-mediated effector mechanisms operating in parallel. The complex metabolic reprogramming that occurs because of Dietary restriction is mediated by several effectors including mTOR, Sirtuins, AMPK, FOXO, and circadian regulators. Our study adds non-coding RNAs to the list of effectors of the DR pathway. Hence, it is not surprising that due to other operating mechanisms, DR is not completely blocked. However, our TAG analysis indicates a greater reduction in TAG levels upon DR as compared to that in AL, upon overexpression of *chinmo* and *flag chinmo* (Figure 6F-G)(compare p-values between AL- vs DR- with AL+ vs DR+). As pointed by Reviewer 2, Figure 5 (Figure 3 in the revised manuscript) is the only experiment where DR response is blocked—in this experiment *chinmo* is knocked down in the fat body. Consistent with RNAi analysis, we would predict a stronger reduction in TAG levels in *FBGS>UAS chinmo*. As predicted a stronger effect on survival is seen in *FBGS>UASchinmo* flies as compared to the *ElavGS>UAS chinmo* flies (Figure 6B-E). TAG analysis of *FBGS>UAS chinmo* was not done in AL and DR diets but on normal food, a much stronger reduction in TAG levels was observed as compared to *3XElavGS> UASchinmo* flies (Figure 6—figure supplement 1E).

Pairwise comparisons have been specified in each of the graphs and the p-values are depicted in the figure panels for all graphs.

4) Some figures in fact show results which are contrary to the concluding model. E.g. in Figure 6J: if there is an effect of FATP-RNAi in both AL and DR, does it not argue that FATP is in fact irrelevant for the DR effect? Similarly, the hypothesis tested in 6K is that elevated expression of lipid metabolism genes should create a DR-like state, in which case we expect overexpression to increase AL lifespan but not DR lifespan. However, we see the opposite: FATP expression extends DR lifespan, but not AL lifespan. Altogether, this actually argues against the model that fat metabolism genes in the FB mediate DR's effects.

We agree that the concluding model in the previous submission did not accurately reflect the results. We have edited the proposed model (Figure 9) and have provided an explanation in the Results section in the revised manuscript. Based on the proposed model, *miR-125* dependent regulation of *chinmo* in the adult brain, as well as DR-mediated post-translational regulation of Chinmo in the fat tissue are required for extension of lifespan upon DR. The proposed model summarizes the mechanism by which *chinmo* regulates lifespan extension by dietary restriction. *miR-125* targets *chinmo* mRNA in the brain under AL and DR conditions. In the adult fat tissue, Chinmo transcriptionally represses genes involved in fat metabolism. Dietary restriction mediated cytoplasmic relocalization of Chinmo in the fat tissue relieves transcriptional repression of genes involved in fat metabolism, thus increasing lifespan. Thus, regulation of chinmo at the transcriptional (nutrition-dependent), post-transcriptional (miR-125-dependent) and post-translational (dSir2-dependent) ensures DR-mediated extension of lifespan.

We do not agree with Reviewer 2 regarding the conclusions drawn from the results in Figure 6J and Figure 6K of the previous submission (currently Figure 7J and Figure 7K). The proteomic analysis of *3X ElavGS > UAS chinmo* flies identified 8 genes and it is likely that modulation of more than one gene would be required to alter fat metabolism. We predicted that if increased expression of one of these genes was sufficient to cause increased lipid turnover and consequentially DR-mediated lifespan extension, then knockdown of this gene would result in a reduction in lifespan upon DR. Knockdown of *fatp* resulted in a 14.8% decrease in median lifespan upon DR and no change under AL conditions (Figure 7J and Figure 7-source data 2). Since knockdown of *fatp* resulted in a decrease in the lifespan of flies that were fed a DR diet, we tested whether increasing the levels of *fatp* in the adult fat tissue led to an increase in the lifespan of flies under AL conditions. Western blot analysis of whole fly lysates prepared from *FBGS > UAS Flag fatp* flies that were fed an AL or DR diet in the presence and absence of RU-486 for 10 days revealed that Flag Fatp was expressed in an inducible manner, however, much lower levels of the flag-tagged protein was detected under AL conditions as compared to that in DR conditions (Figure 7—figure supplement 3D). Overexpressing *UAS-Flag-FATP* specifically in the adult fat tissue increased median life span by 14.2% under AL conditions and by 25% under DR conditions (Figure 7K). The smaller increase in median lifespan in AL conditions is likely due to the lower induction of the protein, nevertheless, an increase in lifespan in both diets indicates that Fatp functions as a pro-longevity factor and that DR dependent increase in lifespan occurs due to an increase in the expression of two or more genes that are regulated by Chinmo. We believe that these data do not argue against the proposed model but indicate that a combination of two more genes likely function together to mediate the effects and a combination of genes would need to be tested to attain the predicted increase in longevity.

5) The manuscript would also be more penetrable without jumping so much between tissues, and a more clear explanation of where chinmo and the miRNAs are normally expressed. Probably the most convincing data are in Figure 5, where we have a manipulation of physiological expression patterns (i.e. chinmo RNAi) yielding the sort of interaction patterns we need to conclude epistasis. But then figure 6 jumps back to over-expressing chinmo in neurons. Flyatlas has the expression of chinmo high in neurons and negligible in fat body, and the miRNAs high in the nervous system. No convincing case is made for expressing a neuronal miRNA in the fat body (Figure 7). I am also very cautious in interpreting protein overexpression with a pan-neuronal driver such as ElavGS – brains are complex and this manipulation will force chinmo expression in a lot of neurons where it may never normally be expressed, causing potentially odd effects. This brings the physiological relevance of a number of manipulations into question.

We thank the reviewer for the suggestion to restrict the analysis to the relevant tissues and to improve clarity and retain focus, we have removed the lifespan data and expression analysis of *DaGS> UAS chinmo* and *5966> UAS chinmo* flies. We have examined the expression of *miR-125* and *chinmo* in the brain and fat tissue under AL and DR conditions and our data indicated that DR induces both *miR-125* and *chinmo* in the fat tissue (Figure 3—figure supplement 1G and Figure 8B, compare -AL and -DR bars). However, the lower expression levels of *miR-125* in the fat body as well as higher induction of *chinmo* mRNA limit *miR-125*-mediated downregulation of Chinmo in this tissue. We have also examined and quantitated tissue-specific and sub-cellular localization of Chinmo protein in the *w^1118^* adult fat body and brain (Figure 3A-C and Figure 3—figure supplement 1A-E, H). We thank Reviewer 2 for appreciating the analysis done with *FBGS>UAS chinmo^RNAi^* flies and would like to highlight other experiments that also depict convincing data. In figure 1 we show that a *let-7-C^hyp^* strain has a dampened DR-mediated lifespan extension and go onto identifying *let-7* and *miR-125* to be responsible for the phenotype. In figure 2, we show that reducing the dosage of *chinmo* completely rescues the DR phenotype of *miR-125* mutants. We also believe that examining the effect of overexpression of *chinmo* was important to pinpoint the tissue where *chinmo* functions to cause DR-mediated lifespan extension.

We believe that it was important to analyze the consequences of ectopic expression of *chinmo* to assess whether its upregulation in the adult brain mimicked *miR-125* mutant phenotypes. This experiment was complementary to the experiments where we show that knockdown of *chinmo* in the fat body promotes a DR-like state under AL conditions (current Figure 3F-J). Moreover, since a given miRNA can target several mRNAs, this experiment was important for confirming that *chinmo* was the functionally relevant target that was mediating the DR phenotype. It was important to identify the tissues where this silencing of *chinmo* was critical for DR-mediated lifespan extension and hence an inducible GeneSwitch Gal4 system seems most appropriate. The FlyAtlas reports expression of genes under normal dietary conditions and hence, we don’t think any conclusions can be drawn regarding expression of genes under AL and DR diets which are significantly different from the normal food used for rearing flies in those experiments. We have examined expression of *miR-125* and *chinmo* in the brain and fat tissue under AL and DR conditions and our data indicated that DR induces both *miR-125* and *chinmo* in the fat tissue (Figure 3—figure supplement 1G and Figure 8B, compare -AL and -DR bars). However, the lower expression levels of *miR-125* in the fat body as well as higher induction of *chinmo* mRNA limit *miR-125*-mediated downregulation of Chinmo in this tissue. By expressing human *miR-125* in the adult fat body, we addressed four questions: (i) Is the role of *miR-125* in fat tissue conserved i.e., can human primary *miR-125* substitute for *Drosophila miR-125*; (ii) *Can miR-125* function as a DR-mimetic i.e., can modulating its level provide benefits of DR, irrespective of diet (i.e., in both AL and DR). (iii) Can increasing levels of the miRNA silence *chinmo* more efficiently and provide additional benefits. (iv) Can increasing the levels of a miRNA in a tissue where it is expressed at low levels be used as a strategy to redirect the role of its downstream targets. Consistent with our expression analysis of *miR-125*, a previous study in the mouse model identified *miR-125b* as one of the miRNAs that increase in the subcutaneous white fat tissue upon caloric restriction (Mori et al., 2012). However, the beneficial effects of increasing *miR-125b* in the fat tissue were not examined in the mouse model. Hence, we tested whether artificially increasing the levels of *miR-125* specifically in the fat tissue would lead to silencing of *chinmo* and consequentially lead to an increase in lifespan. Our results revealed that overexpression of *miR-125* in fat tissue does increase lifespan of flies that are fed an AL diet but more so in DR. Since, only one dose/condition of induction was utilized, further optimization of induction duration or RU486 would likely lead to a greater enhancement in lifespan upon AL. Nevertheless, these initial results are promising because the data indicates that the modulation of this miRNA in the fat tissue is beneficial at an organismal level. In addition, this analysis also shows that the functional role of *miR-125* as a pro-longevity factor is conserved and that human *pri miR-125* can extend lifespan. In the revised manuscript, we have edited the text to clarify that both *miR-125*-dependent and nutrition-dependent post-translational control of Chinmo mediate the DR-mediated increase in lifespan.

We believe that it was important to analyze the consequences of ectopic expression of *chinmo* to assess whether its upregulation in the adult brain mimicked *miR-125* mutant phenotypes. This experiment was complementary to the experiments where we show that knockdown of *chinmo* in the fat body promotes a DR-like state under AL conditions (current Figure 3F-J). Moreover, since a miRNA can target several mRNAs, this experiment was important for confirming that *chinmo* was the functionally relevant target that was mediating the DR phenotype. It was important to identify the tissues where this silencing of *chinmo* was critical for DR-mediated lifespan extension and hence an inducible GeneSwitch Gal4 system seems most appropriate. The other approach that we utilized was to examine whether reducing *chinmo* dosage by c*hinmo^RNAi^* or genetic mutation (*chinmo^1^*) in *miR-125* mutants was able to rescue *miR-125* mutant DR phenotype (presented in Figure 2). Given the broad pattern of Chinmo derepression in *miR-125* mutants, we chose the ElavGS to drive the expression of *chinmo* (Figure 2B) (Chawla et al., 2016).

6) The way non-autonomous effects of chinmo are shown is quite nice. But some statements in this section are quite surprising, especially that mRNA can diffuse between tissues. This statement would require quite a lot more data to substantiate. A more parsimonious explanation is a feed-forward loop in which chinmo expression in one tissue promotes its expression elsewhere through a secondary signal.

We agree with Reviewer 2, for the need for more experiments and explanations in support of the non-autonomous effects of Chinmo. This statement has been substantiated with the addition of new data (Figure 4, Figure 5, and Figure 3A-C). For a more detailed explanation, we would like to request Reviewer 2 to refer to our response to Reviewer 1 comment 8). Some of the experiments that have been performed and included in the revised manuscript are listed below:

i. Chinmo redistributes to the cytoplasm upon nutrient deprivation (Figure 4A and Figure 5C-N)

ii. Deacetylation of Chinmo by dSir2 is required for its nuclear export upon nutrient deprivation (Figure 4 and Figure 4—figure supplement 1).

iii. Chinmo fractionates with exosomes upon starvation (Figure 5).

iv. DR induces c*hinmo* mRNA levels in the hemolymph (Figure 3E).

v. Nutrient deprivation leads to an increase in non-nuclear Chinmo in the fat tissue of *w^1118^* (Figure 3A-C, and Figure 3—figure supplement 1A-E).

vi. Neuronal Chinmo upregulation induces Chinmo expression non-autonomously in the fat tissue (Figure 6J-K, Figure 6—figure supplement 2, Figure 6—figure supplement 3A-E).

We thank the reviewer for suggesting a more parsimonious explanation for the nonautonomous role of Chinmo and have included the suggestion as a plausible mechanism to explain our data in addition to the experiments performed in the revised manuscript (Lines 834-836).

7) Insufficient genetic information. This is notoriously important when looking at epistasis in lifespan. Number of rounds of backcrossing must be given. The data are not useful if mutants were not freshly backcrossed before recombination. We also are not told whether mutants were heterozygous or homozygous. The Figure 1B qRT-PCR data also don't strongly support the claim that the Let7C mutant is actually a hypomorph – expression levels are not dramatically altered?

The genetic scheme for generation of strains used in Figure 1 and 2 is included as Figure 1—figure supplement 1 of the revised manuscript. The same scheme was previously used in our previously published study (Chawla et al., 2016). Our approach relies on the comparison between strains with two different rescuing transgenes in a background as close to identical as possible. This procedure and the strategy should clarify that the strains we have analyzed have a common genetic background. It is not clear whether the recommendation of backcrossing the transgenes/mutants will solve the issue: after backcrossing the strains, we would still need to follow the 8- generation scheme (F1 to F8) in Figure 1 figure supplement-1 to generate the final strains and spontaneous background mutations could be introduced and fixed in the population at any of these steps. The rescue and the mutant lines were generated by expressing a single copy of the wild type, Δ*miR-100*, *let-7* or Δ*miR-125* transgene in a transheterozygous *let-7-C^null^* mutant background (*let-7-C^KO2^/let-7-C^GKI^*). The *let-7-C* rescuing (wild type) transgene encodes a 17983 base-pair genomic fragment containing the *let-7-C* locus. The Δ*miR100*, Δ*let-7* and Δ*miR-125* are derivatives of the *let-7-C* transgenes that lack the mature *miR-100*, *let-7,* and *miR-125* sequences, respectively (Lines 174-179). For a more detailed explanation of the genetic scheme, we would like to request the reviewer to refer to our response to Reviewer 1 comment 1.

The lifespan and RT-PCR analysis for the *let-7-C^hyp^* and *let-7-C^hyp^/Rescue* line has been repeated with another set of strains that express a single copy of the *let-7-C^hyp^*. These edits have been incorporated to ensure that a genetically identical control is used for the analysis of the *let-7-C^hyp^* in response to comment 11 by Reviewer 3 and the main text of the revised manuscript regarding the new RT-PCR data and lifespan analysis. Since the tested strain encodes a single copy of the *let-7-C^hyp^* transgene (as opposed to two copies that was presented earlier), a stronger decrease is seen in the hypomorph as opposed to the control even in younger flies (Figure 1E-G). As expected, the expression is dramatically reduced with age (Figure 1—figure supplement 2B-D).

8) number of key points are made without reference to literature or data e.g. chinmo is a target of Let7C, Let7C miRNAs are normally expressed in the nervous system.

We have included the references regarding the expression of *let-7-C* miRNAs (Chawla and Sokol., 2012). References pertaining to the targeting of *chinmo* by *let-7-C* miRNAs have been included in the main text (Wu et al., Chawla et al., 2016). Additionally, we have included a panel representing a schematic (Figure 2A) and immunostaining (Figure 2B) of Chinmo in the rescue line and *miR-125* mutant in AL and DR conditions (Figure 2A-B)(Chawla et al., 2016).

Reviewer #3:Pandey and colleagues are studying how miRNAs and their targets regulate the extension of lifespan upon dietary restriction (reduced protein content in diet, termed DR). Previous work from the lab of one of the authors has shown that lifespan extension under DR can be prevented when triglyceride synthesis or breakdown is inhibited (Katewa, 2012, 2016). These observations have led to the current model in the field that the enhanced turnover of fatty acids is the critical metabolic adaptation for lifespan extension under DR. In the current study, the authors use established stocks containing deletions and rescue constructs of the let-7-C locus, which contains 3 miRNAs (miR100, miR125 and let-7) and examine lifespan and lipid droplets (fatty acids) in adult *Drosophila*. Ultimately, their data lead them to propose a model in which neuronal miR125 represses chinmo expression in both neurons and (somehow) peripheral tissues; the reduction of chinmo in peripheral tissue leads to increased expression of genes encoding fatty acid metabolism, and this leads to DR-induced lifespan extension. Unfortunately, the current manuscript is lacking critical data to support the model. Additionally, controls are missing and key Gal4 drivers are not lineage-traced, increasing the confusion about the cell autonomy of miR125 and chinmo. The text of the manuscript is lacking sufficient rationale for many experiments, and important introductory facts (let-C-7 locus biology, chinmo biology, DR model and biology and how DR compared to caloric restriction) are either relegated to the discussion or entirely lacking from the manuscript. The unclear and confusing autotomy of chinmo activity greatly reduce enthusiasm. This manuscript could be improved by a large number of new experiments and an editorial overhaul.1) Dr. Chawla previously showed that adult flies lacking an upstream regulatory region in the let-7-C locus but containing all three miRNAs (called let-7-Chyp) had a significantly shorter lifespan (Chawla, 2016). In the current study, they find that the shorter lifespan of these flies cannot be extended by DR, in contrast to flies possessing the regulatory let-7-C sequences. This observation forms the foundation for the study. Flies lacking miR125 or let-7 similarly do not extend lifespan under DR. They authors decide to follow up on the miR125 observation. The Sokol lab has previously shown that both miR125 and let-7 degrade transcripts for chinmo, which encodes a putative transcriptional repressor that has only been observed in the nucleus of cells that express it. Indeed, miR125 null flies that are heterozygous for a null chinmo allele now display extended lifespan upon DR and have enlarged LDs. [The experiments using the let-7-C-Gal4 to deplete chinmo in Figure 2 are difficult to interpret because the Gal4 is not regulated by RU486 and is active through the life of the fly, leading to depletion of chinmo throughout development with consequences of defective neuronal patterning (Zhu, Cell 2006 and many other papers from Tzumin Lee and other groups)]. These data suggest that the maintenance of chinmo in miR125 null flies underlies the lack of life extension under DR.

We thank the reviewer 3 for a very thorough feedback of the data in the manuscript, however, we do not agree with the reviewer regarding the interpretation of *let-7-C* Gal4 for the analysis in Figure 2C-D. *let-7-C* Gal4 is not active throughout development, and its activity coincides with the expression of *let-7-C* primary transcript. The neuronal patterning of *chinmo* occurs much earlier during development- and *let-7-C* miRNA mediated downregulation of *chinmo* only affects the later-born neurons (Author response image 1). Our previously published analyses revealed that *let-7* represses *chinmo* during development and *miR-125* silences *chinmo* in adulthood (Chawla et al., 2016). We acknowledge Reviewer 3 for raising this relevant point and agree that more explanation should be included in support of the use of *let-7-C Gal4* in the revised manuscript and we have summarized the differential mode of regulation of *chinmo* by *let-7* and *miR-125* in Figure 2A of the revised manuscript.

**Author response image 1. respfig1:** Chinmo is silenced by let-7 and miR-125 in a biphasic manner. (A) Schematic of a 1.4kb Chinmo 3’UTR fragment indicating number and arrangement of let-7 and miR-125 binding sites. (B) *Chinmo* derepression is significantly higher in let-7 mutant brains dissected from staged pupae (24h and 96h after puparium formation)(red dashed lines indicates the stage at which let-7 represses *chinmo*) (C) *Chinmo* is predominantly regulated by *miR-125* in adult flies as the degree of depression of *chinmo* in *ΔmiR-125* flies is more widespread and higher in *miR-125* mutants than in *Δlet-7* adult fly brains (Day-3 after eclosion)(red dashed lines indicates late pupal stage).

chinmo is abundantly expressed in the nervous system of Embryo, L1, L2, and early L3 (Zhu et al., 2006). Let-7-C miRNAs are induced by the ecdysone pulse in mid-late third larval instar (Chawla and Sokol., 2012) and this induction of let-7-C miRNAs coincides with repression of chinmo after mid-third larval stages. The 3’UTR of chinmo has multiple miRNA binding sites for both let-7 and miR-125 and the Chinmo gradient (downregulation) that is required for neuronal transitions during metamorphosis is mediated predominantly by let-7. The genotype of the strain analyzed in Figure 2D is: w^1118^; let-7-C^GKI^ / let-7-C^KO2^, P{neoFRT}40A; {v+, let-7-C ΔmiR-125} attP2 / P{w+, UAS-chinmo^RNAi^ 148}VK00033.

The ∆*miR-125* transgene on chromosome 3 has a single wild-type copy of *let-7* miRNA that predominantly silences *chinmo* during the 3^rd^ larval instar. Our previously published study has shown that *let-7* and *miR-125* miRNAs silence *chinmo* in a biphasic manner, with *let-7* being the predominant repressor of *chinmo* during development and *miR-125* being the predominant silencer of *chinmo* during late pupal stages into adulthood. Quantitation of Chinmo protein (by immunofluorescence) in *let-7* and *miR-125* mutants (Chawla et al., 2016) during different pupal stages indicates that *chinmo* is predominantly repressed by *let-7* during development and by *miR-125* during adulthood (Schematic above). This expression analysis is further corroborated by our previously published phenotypic analysis of ∆*let-7,* ∆*miR-125* double mutants, and ∆*let-7* single mutants that show that the loss of *miR-125* does not exaggerate the *let-7* mutant developmental phenotypes. Thus, indicating that ∆*miR-125* mutants do not display any defects in neuronal patterning during development. Moreover, wild type *let-7* in the ∆*miR-125* mutants and control line ensure that *chinmo* is already repressed during metamorphosis.

2) In Figure 3, they try to determine in which tissue ectopic expression of chinmo decreases lifespan under DR. They used the gene switch inducible system for these experiments, only to find that ectopic expression of chinmo under Da, 5966 or FB leads to early mortality. In fact, even with the Elav neuronal driver, lifespan is shorter than the AL, no-RU-468 control. I find it difficult to logically connect the relatively early death caused by Chinmo mis-expression with a suppression of lifespan extension under DR. The authors need to find another way to test their hypothesis.

We agree with Reviewer 3 that ubiquitously driving *chinmo* expression with *DaGS* leads to early mortality. This is expected for proteins like Chinmo, which play dosage-sensitive essential roles. These experiments were not aimed at connecting overexpression of *chinmo* with an extension of lifespan by DR but we expected to observe a reduction in the DR-mediated extension of lifespan. We believe that it was important to analyze the consequences of ectopic expression of *chinmo* to assess whether its upregulation in the adult brain mimicked *miR-125* mutant phenotypes. This experiment was complementary to the experiments where we show that knockdown of *chinmo* in the fat body promotes a DR-like state under AL conditions (current Figure 3F-J). Moreover, since a miRNA can target several mRNAs, this experiment was important for confirming that *chinmo* was the functionally relevant target that was mediating the DR phenotype. It was important to identify the tissues where this silencing of *chinmo* was critical for DR-mediated lifespan extension and hence an inducible GeneSwitch Gal4 system seems most appropriate. The other approach that we utilized was to examine whether reducing *chinmo* dosage by c*hinmo^RNAi^* or genetic mutation (*chinmo^1^*) in *miR-125* mutants was able to rescue *miR125* mutant DR phenotype (presented in Figure 2). We have edited figure 3 (current Figure 6 of the revised manuscript) and removed the analysis done with *DaGS* and *5966* GeneSwitch drivers for clarity. The figure now includes analysis of ectopic expression of *chinmo* and *flag chinmo* in adult neurons (*3X ElavGS*) and in adult fat tissue (*S_1_106*). Comparison of survival curves of flies that overexpress Chinmo or flag Chinmo in adult neurons (Chinmo: AL+ vs DR+ p=0.0066 and Χ^2^= 7.37; Flag Chinmo p=0.0068 and Χ^2^=7.34) with flies that overexpress Chinmo or Flag Chinmo in adult fat tissue (Chinmo: AL+ vs DR+ p=0.0283 and Χ^2^= 4.81; Flag Chinmo p=0.9472 and Χ^2^=0.0043) indicates that overexpression of this protein in the adult fat tissue dampens the DR dependent increase in lifespan more significantly (Figure 6B-E). We believe that the statistical comparisons (p-values and Χ^2^ values) indicate that there is a significant difference between survival curves in AL and DR conditions and these should help the readers in connecting reduction in c*hinmo* with extension of lifespan upon DR. These experiments confirm that deregulation of *chinmo* is the cause of *miR-125* mutant DR phenotypes. Analysis of tissue-specific inducible expression of Chinmo was also important to uncover the non-autonomous effect of *chinmo* in fat tissue in *3X ElavGS >UAS chinm*o flies. Based on the lifespan analysis, the maximum lifespan of *3X ElavGS >UAS chinmo* flies in (AL-) conditions ranged between 56-58 days and upon RU mediated induction (AL+) varied between 50-54 days. The differences in DR- and DR+ conditions were expected if Chinmo were to play a role in DR-mediated lifespan extension.

Lastly, we would like to emphasize that the decision to use of GeneSwitch system to drive overexpression of *chinmo* was based on the requirement for adult-specific, tissue-specific, and inducible expression of *chinmo*. We are not aware of any other strategy that can fulfill the above criteria and be used to drive expression of dosage-sensitive regulators for examining the organismal level changes.

3) In Figure 3, they show that Elav>chinmo under RU-486 leads to a decrease in the size of lipid droplets, which they conclude connects neuronal chinmo to fat body metabolism in peripheral tissues.

In Figure 3 (Current figure 6), quantitation of stored fat (TAG levels) as well as lipid droplet size, were decreased (Figure 6F-G). The conclusion that ectopic expression of *chinmo* in adult neurons lead to a reduction in expression of genes involved in fat metabolism was arrived at by the proteomic analysis of *3X ElavGS> UAS chinmo* flies and validation of downstream genes by RT-PCR analysis of the downregulated proteins that are involved in fat metabolism. These data are presented in Figure 7 of the revised manuscript. Additional supplementary data that connects neuronal *chinmo* to fat body metabolism in peripheral tissues is that overexpression of *chinmo* or *Flag chinmo* in adult neurons leads to an increase in Chinmo protein in the fat tissue (Figure 6—figure supplement 2 and Figure 6—figure supplement 3).

4) In Figure 4, they show the expression patterns of the GS Gal4 drivers, and because they see chinmo protein in tissues where the driver is supposedly not expressed, they conclude that chinmo RNA or protein is able to diffuse from neuronal cells to other tissues to induce metabolic changes. This is potentially an interesting model, but there are absolutely no data to support it. First, they need perform both real-time and lineage tracing of these drivers using G-trace. [G-trace should also be used for let-7-C-Gal4]. If these experiments support their hypothesis, then they would need to test how the movement of chinmo RNA or protein between tissues could occur – for example, by exosomes?

We agree with Reviewer 3 that more data needs to be presented in support of our hypothesis regarding the movement of *chinmo* mRNA or protein and here we list the new experiments that have been performed to show that *chinmo* mRNA and/or protein is expressed in a non-autonomous manner. For a more detailed explanation, we would like to request Reviewer 3 to refer to our response to Reviewer 1 comment 8.

i. Chinmo redistributes to the cytoplasm upon nutrient deprivation (Figure 4A and Figure 5C-N)

ii. Deacetylation of Chinmo by dSir2 is required for its nuclear export upon nutrient deprivation (Figure 4 and Figure 4—figure supplement 1).

iii. Chinmo fractionates with exosomes upon starvation (Figure 5).

iv. DR induces c*hinmo* mRNA levels in the hemolymph (Figure 3E).

v. Nutrient deprivation leads to an increase in non-nuclear Chinmo in the fat tissue of *w^1118^* (Figure 3A-C and Figure 3—figure supplement1A-E).

vi. Neuronal Chinmo upregulation induces Chinmo expression non-autonomously in the fat tissue (Figure 6J-K, Figure 6—figure supplement 2, Figure 6—figure supplement 3A-E).

In addition, we would like to cite an example of the study by Ulgherait et al., 2014, in which the authors utilized GeneSwitch drivers to provide evidence for a non-autonomous role for AMPK in organismal aging.

5) In Figure 5, they deplete chinmo from fat body using FB GS driver. What is the hypothesis being tested here? It was not clear to me. If neuronal chinmo RNA or protein travels from neurons to fat body, then how would depletion of chinmo from fat body alter this movement? These needs to be explained in the text. Let us assume that the chinmo transcripts being depleted in Figure 5 are produced by fat body cells (and not neurons), what is the prediction of this on lifespan extension under DR, TAG levels and lipid droplet size? In my mind, the upregulation of miR125 upon DR leads to the downregulation of chinmo message and protein in neuronal cells. This effect is somehow transmitted to fat body cells, which alter lipid metabolism to promote lifespan extension. If chinmo levels in fat body were depleted, there should be no effect on DR-induced lifespan extension and no change in TAG levels and no reduction in LD size. Indeed, my predictions are borne out by the data. There is no further increase in lifespan extension under DR when chinmo is depleted from fat body (Figure 5B blue solid and dashed lines); there is no difference in TAG levels (Figure 5C, blue solid and hashed bars) and LD size is not reduced (Figure 5E, light and dark blue dots). Yet the authors write in the title to Figure 5 and in lines 280-290 something to the effect that reducing chinmo levels in the adult fat body increases lifespan and enhances lipid metabolism. Would the authors please explain in the text the predictions for this experiment and the results? Their interpretation and mine are not congruous.

We agree with Reviewer 3, that more explanation should be provided to explain the premise for doing this experiment. We have included additional data in support of the hypothesis and have made edits in the manuscript that we explain here. Immunostaining of adult brains and fat tissue with Chinmo antibody indicates that *miR-125* mutant flies express higher levels of Chinmo in the brain and fat tissue as compared to the control/rescue flies. These data indicate that Chinmo is derepressed in the adult brain and to some extent in the fat body in the *miR-125* mutants (Figure 2B). We predicted that silencing of *chinmo* by *miR-125* in either brain tissue and/or fat tissue is required for DR- mediated lifespan extension in wild-type flies. Since highly expressed *miR-125* ensures that c*hinmo* is repressed in adult brain in wild type flies/rescue (Figure 2B), we tested the effect of reducing *chinmo* in the fat tissue on lifespan extension by DR. The hypothesis being tested was that reduction in c*hinmo* levels in the fat tissue is required for DR=mediated lifespan extension. Our prediction was that knockdown of *chinmo* in adult fat tissue would lead to extension of lifespan, increase in TAG levels and increase in lipid droplet size in AL conditions but no further increase of lifespan, no increase in TAG and no increase in lipid droplet diameter under DR conditions. Indeed, the data obtained from these experiments matched our prediction. These results indicate that knockdown of *chinmo* in the adult fat body is a prerequisite for DR-mediated lifespan extension and hence, a further knockdown of *chinmo* in DR conditions does not increase the lifespan further. However, knocking down *chinmo* in AL conditions recapitulates a DR like state and leads to an increase in lifespan.

We thank Reviewer 3 for pointing out the incorrectness in the title. We have edited the title to reflect that knockdown of *chinmo* in fat tissue leads to an increase in lifespan and enhances lipid metabolism in AL conditions. These data are now represented as Figure 2F-J of the revised manuscript.

6) In Figure 6, they perform mass spec on adult females in which chinmo has been inducibly mis-expressed in neurons. They find 40 proteins differentially expressed between the induced and uninduced samples, 7 upregulated and 33 downregulated. Of the latter, 7 proteins are involved in fat metabolism and their mRNAs decrease 10 days after induction of chinmo. Do the authors have an earlier time point? If chinmo does directly repress these 7 genes, it presumably would do so in a shorter time frame, for example 1-2 days after induction. 10 days is a long time-frame for gene expression and may represent indirect regulation, and if the authors saw an effect after 2 days, this would be a more meaningful result.

We thank reviewer 3 for the suggestion to analyze expression at an earlier time point to determine whether the effects are direct. We analyzed the expression of the 8 genes identified after inducing expression of chinmo for 2 days in *3X ElavGS>UAS chinmo* flies and the data from this analysis is represented in Figure 7—figure supplement 1. Induction of *UASchinmo* for 2 days was sufficient to lead to a statistically significant reduction of 7 of the 8 genes analyzed in the head tissue. 3 out of 8 genes analyzed also displayed a statistically significant albeit lower reduction in the decapitated body tissue. These data confirmed that Chinmo represses these genes directly in the head tissue and longer induction (10d) of *uas chinmo* in the adult neurons is required for the genes to be repressed in the peripheral tissues.

7) Do these genes have chinmo binding sites in regulatory regions? It would also be helpful to validate decreased expression of proteins using antibodies or enhancer traps to characterized genes, for example, FATP for which both antibodies and enhancer traps exist.

A recently published study by Grmai et al., 2021 describes the transcriptomic analysis of *chinmo*-deficient cells from *Drosophila* testis. This study reports the following consensus Chinmo binding site (G/A)ATGCAC(T/C)(T/N)NN. We examined the noncoding regions (1500 bp upstream of the transcription start site, introns and 1500bp downstream of the termination site) of the 8 genes involved in fat metabolism that we validated to be Chinmo targets. Two binding sites were identified in *CG9527* and one each in *CG9577* and *CG5009*. We also searched the reverse complement sequence of the genes for the consensus binding and found that 7 of the 8 genes had 1 site and the remaining 1 (*CG17544)* had 3 sites.

As a second approach to identify motifs in the promoter sequences of the 8 genes that were validated to be transcriptionally repressed by Chinmo, we performed MEME analysis. The promoter sequences (50 base pairs) were downloaded from the Eukaryotic promoter database (EPD) and used for analysis. The MEME Suite (Bailey et al., 2009; Bailey et al., 2015) is a motif-based sequence analysis tool that can fetch motifs hidden in biological sequences – DNA, RNA, or proteins. We searched the promoter sequences of the genes, that we validated to be downregulated. The MEME tool (Version 5.3.3) from the MEME Suite web server (https://meme-suite.org/meme/tools/meme) was used with the default parameters. MEME was instructed to report the top ten motifs from the search. No motifs could be identified with this approach.

As suggested by reviewer 3, we have verified that overexpression of Flag-tagged chinmo in adult neurons using a *3X ElavGS* driver (*3X ElavGS> UAS Flag chinmo*) reduced endogenous Fatp protein levels under AL conditions. The western blot analysis was performed on whole fly lysates and is currently represented as Figure 7—figure supplement 3A and following explanation has been included in the Results section:

“To verify whether overexpression of *chinmo* in adult neurons led to a decrease in protein levels of the endogenous proteins involved in fat metabolism, we performed western blot analysis of whole fly lysates prepared from 3x *Elavgs>UAS flag chinmo* flies that were fed either an AL+ RU-486 or AL+ solvent/ethanol diet for 10 days. Since, most of the downstream targets identified are not that well-characterized (*CG2107, CG9527, CG17544, CG5009, CG8778,* and *CG9577*), we tested the expression of endogenous Fatp for which antibodies were available. Western blot analysis was performed with anti-flag antibody to detect Chinmo, anti-Fatp antibody to detect endogenous Fatp, and antitubulin as normalization control. Consistent with the proteomics data, a reduction in Fatp levels was seen upon upregulation of Flag Chinmo in adult neurons (Figure 7—figure supplement 3A).”

8) They then test the functional role for FASN1 and FATP in DR-dependent lifespan extension. Once again, it would be helpful for the authors to explain in the text the predictions for this experiment. To my mind, if downregulation of chinmo in the fat body is important for DR-induced lifespan extension and if chinmo represses the fasn1 and fatp genes, then loss of chinmo should in increase FASN1 and FATP expression and this should aid in fatty acid turnover. Depletion of fasn1 and fatp should then decrease DR-induced lifespan extension, which is confirmed by the results (Figure 6I and J), and mis-expression of these factors should increase this extension, which again is supported by the results (Figure 6L). These are important results for the paper but are insufficiently explained in the text by the authors.

We agree with Reviewer 3 and thank the reviewer for the suggestion. The text in the Results section of the revised manuscript has been edited to explain the predictions for these experiments. To test whether downregulation of the candidate fat metabolism genes in the adult fat tissue was responsible for modulating lifespan, we measured survival of flies that expressed transgenes to knockdown *fasn1* and *fatp* specifically in the adult fat body (Figure 7I, J). We predicted that if overexpression of one of these genes was sufficient to cause increased lipid turnover and consequentially DR-mediated lifespan extension, then knockdown of these genes would result in a reduction in lifespan upon DR. Knockdown of *fasn1* resulted in an 18.7% decrease in median lifespan of flies that were fed an AL diet and a 3.22% decrease in median lifespan on DR diet (Figure 7I and Figure 7-source data 1). Knockdown of *fatp* resulted in a 14.8% decrease in median lifespan upon DR and no change under AL conditions (Figure 7J and Figure 7-source data 2). Since knockdown of *fatp* resulted in a decrease in the lifespan of flies that were fed a DR diet, we tested whether increasing the levels of *fatp* in the adult fat tissue led to an increase in the lifespan of flies under AL conditions. Western blot analysis of whole fly lysates prepared from *FBGS > UAS Flag fatp* flies that were fed an AL or DR diet in the presence and absence of RU-486 for 10 days revealed that Flag Fatp was expressed in an inducible manner, however, much lower levels of the flag-tagged protein was detected under AL conditions as compared to that in DR conditions (Figure 7—figure supplement 3D). Overexpressing *UAS-Flag-FATP* specifically in the adult fat tissue increased median life span by 14.2% under AL conditions and by 25% under DR conditions (Figure 7K). The smaller increase in median lifespan in AL conditions is likely due to the lower induction of the protein, nevertheless, an increase in lifespan in both diets indicates that Fatp functions as a pro-longevity factor and that DR dependent increase in lifespan occurs due to an increase in the expression of two or more genes that are regulated by Chinmo.

9) In Figure 7, they express human miR125 in fat body and examine lifespan extension. Once again, the rationale for these experiments is totally unclear to me. The model introduced earlier in the paper is that neuronal (not fat body) miR125 extends lifespan under DR. Why did they perform the experiments in Figure 7? How do these results add to the paper? To me, they detract from the paper. The authors need to definitely establish in which cells miR125 is acting and in which cells chinmo is being regulated.

We apologize for the lack of clarity in the Results section of Figure 8 (Figure 7 in the previously submitted version). We have edited the Results section to include a rationale for performing the experiments in Figure 8. We have examined the expression of *miR-125* and *chinmo* in the brain and fat tissue under AL and DR conditions and our data indicated that DR induces both *miR-125* and *chinmo* in the fat tissue (Figure 3—figure supplement 1G and Figure 8B, compare -AL and -DR bars). However, the lower expression levels of *miR-125* in the fat body as well as higher induction of *chinmo* mRNA limit *miR-125*mediated downregulation of Chinmo in this tissue. By expressing human *miR-125* in the adult fat body, we addressed four questions: (i) Is the role of *miR-125* in fat tissue conserved i.e., can human primary *miR-125* substitute for *Drosophila miR-125*; (ii) *Can miR-125* function as a DR-mimetic i.e., can modulating its level provide benefits of DR, irrespective of diet (i.e., in both AL and DR). (iii) Can increasing levels of the miRNA silence *chinmo* more efficiently and provide additional benefits. (iv) Can increasing the levels of a miRNA in a tissue that it is not expressed/expressed at low levels be used as a strategy to redirect the role of its downstream targets. Consistent with our expression analysis of *miR-125*, a previous study in the mouse model reported *miR-125b* as one of the miRNAs that increase in the subcutaneous white fat tissue upon caloric restriction (Mori et al., 2012). However, the beneficial effects of increasing *miR-125b* in the fat tissue were not examined in the mouse model. Hence, we tested whether artificially increasing the levels of *miR-125* specifically in the fat tissue would lead to silencing of *chinmo* and consequentially lead to an increase in lifespan. Our results revealed that overexpression of *miR-125* in fat tissue does increase lifespan of flies that are fed an AL diet but more so in DR. Since, only one dose/condition of induction was utilized, further optimization of induction duration or RU-486 would likely lead to a greater enhancement in lifespan upon AL. Nevertheless, these initial results are promising because the data indicates that the modulation of this miRNA in the fat tissue is beneficial at an organismal level. In addition, this analysis also shows that the functional role of miR-125 as a pro-longevity factor is conserved and that human *pri miR-125* can extend lifespan. In the revised manuscript, we have edited the text to clarify that both *miR-125*-dependent and nutrition-dependent post-translational control of Chinmo mediate the DR-mediated increase in lifespan.

10) Figure 8 is a model of the result, but it does not clarify any of the confusion about the autonomy of chinmo activity.

We have edited the model (Figure 9) to clarify that both *miR-125* dependent regulation of *chinmo* in the adult brain, as well as DR-mediated post-translational regulation of Chinmo in the fat tissue, are required for extension of lifespan upon DR. The proposed model summarizes the mechanism by which *chinmo* regulates lifespan extension by dietary restriction. *miR-125* targets *chinmo* mRNA in the brain under AL and DR conditions. In the adult fat tissue, Chinmo transcriptionally represses genes involved in fat metabolism. Dietary restriction mediated cytoplasmic relocalization of Chinmo in the fat tissue relieves transcriptional repression of genes involved in fat metabolism, thus increasing lifespan. Thus, regulation of *chinmo* at the transcriptional (nutrition-dependent), post-transcriptional (*miR-125*-dependent) and post-translational (dSir2-dependent) ensures DR-mediated extension of lifespan.

11) Figure 1B are missing the appropriate control, which is the "rescue" genotype. W^1118^ is not the right control for this experiment because it does not contain the let-7-C alleles and the attP-inserted transgene. This is important because it forms the foundation for the entire study.

We agree with Reviewer 3 that *w^1118^* is not the appropriate control for the experiment. In the previously submitted manuscript, the strain encoded two copies of the *let-7-C^hyp^* transgene (*let-7-C^null^; let-7-C^hyp^/let-7-C^hyp^*). To make comparisons with a control having an identical genetic background, we have done the analysis again with a strain with a single copy of the *let-7-C^hyp^* and a single of the *let-7-C* transgene lacking all three miRNAs (*w^1118^; let-7-C^GKI^/ let-7-C^KO2^* in a *let-7-C^null^* transheterozygous background (*P{neoFRT}40A; P{w+, let-7-Cp^3.3kb^::cDNA} / {v+, let-7-C ^Δlet-7-C miRNAs^}attP2*). The experiment has been repeated with a genetically identical control strain that has the transheterozygous *let-7-C* null alleles, a single copy of the hypomorph transgene, and a wild type *let-7-C* transgene (*let-7-C^GKI^/ let-7-C^KO2^, P{neoFRT}40A; P{w+, let-7Cp^3.3kb^::cDNA} / {v+, let-7-C }attP2*). Both rescue and experimental lines have been tested again and are represented as Figure 1H-I and figure 1-source data 1-2. Since these strains are different from the hypomorph used earlier, the RT-PCR analysis for quantitation of *let-7-C* miRNAs has also been repeated with the strains used for lifespan analysis and the data are presented as Figure 1E-G and Figure 1—figure supplement 2BD of the revised manuscript.

12) Figure 2A,B are missing controls (δ miR125 alone and chinmo1/+ alone). Furthermore, the driver in Figure 2A is let-7-CGal4, which is not a GS gal4 and therefore is active through development. The correct experiment here would be elavGS. This should be remedied.

We thank Reviewer 3 for pointing out the missing controls. The survival curves for testing the effect of genetic background have been represented in Figure 1—figure supplement 2K, Figure 2—figure supplement 1B, Figure 1—figure supplement 2-source data 1, and Figure 2—figure supplement 1-source data 1 of the revised manuscript.

We agree with Reviewer 3 that the Gal4 used is *let7-C* Gal4 and is not GeneSwitch. Our premise for using *let-7-C* Gal4 is based on our previous analysis of *miR-125* mediated regulation of *chinmo*. Chinmo is regulated by *let-7* and *miR-125* and wild type *let-7* in the *miR-125* mutants ensures that *chinmo* is repressed during development. In the Figure 2C and D panels (Previous Figure 2A) we are trying to assess whether knockdown of *chinmo* in *miR-125* expressing cells rescues the *miR-125* mutant phenotype. Our previous analysis (Chawla et al., 2016) of Chinmo protein levels in adult brains indicates that Chinmo is predominantly repressed by *miR-125* in adult fly brains and as a consequence, *miR-125* mutants display upregulated Chinmo protein in the brain (Author response image 2). *Let-7* miRNA ensures that *chinmo* is repressed during development (late third instar onwards). Since, both the control and experimental strains in Figure 2C-D have wildtype *let-7* miRNA, *chinmo* is repressed during development even when *let-7-C Gal4* is driving *chinmo^RNA^*^i^. However, the absence of *miR-125* in Figure 2D leads to the derepression of *chinmo* in adult brains. Hence, to examine the effect of *chinmo^RNAi^* specifically in *miR-125* expressing cells, *let-7-C Gal4* is more appropriate. In addition, we would like the reviewer to refer to Figure 5 in Chawla et al., 2016, where we have analyzed the expression of *chinmo* mRNA and protein in *let-7* and *miR-125* mutant lines during different developmental stages and in adult flies. In addition, we have also provided functional evidence for the differential regulation of *chinmo* by *let-7* and *miR-125* in the previously published study. *miR-125* mutants do not display any developmental phenotypes associated with *chinmo* overexpression (Chawla et al., 2016). We have included a schematic of the differential regulation of *chinmo* by *let-7* and *miR-125* as Figure 2A in the revised manuscript.

**Author response image 2. respfig2:** miR-125 is the predominant miRNA that silences chinmo in adult flies. (A-D) Confocal images of 3d adult brains immunostained for Chinmo (green). No Chinmo expression was detected in brains of flies harboring either the wild type of the ΔmiR-100 transgene (panels A and B). The level of Chinmo expression in ΔmiR-125 mutants is much higher than in Δlet-7 mutant adult flies (compare panels C and D).

13) Figure 2C and D – the AD and DR TAG profiles should be displayed on the same graph. Why are they separated? Why are is the Y-axis scale so different (0-7 on the AD and 0-50 on the DR) between AL and DR? I can't determine what happens to TAG levels under DR in the various genotypes. This is a significant issue for me. This experiment needs to be performed again and plotted on the same graph.

The data represented in Figure 2C and D of the previously submitted manuscript were normalized to protein and hence lysates prepared from flies that were fed a DR diet had lower amounts of proteins and hence higher TAG/[protein] (y-axis) values as compared to lysates from flies that were fed an AL diet. As suggested by Reviewer 3, the experiment was performed again and we didn’t see much change in the values. The AL and DR data are now plotted on the same graph (Figure 2G).

14) Figure 2G-J – why is the chinmo protein not monitored here?

We thank the reviewer for pointing this out, In the revised manuscript we have examined Chinmo protein levels under AL and DR conditions (Figure 3A-C; Figure 3—figure supplement 1A-E and 1H) and starvation conditions (Figure 4—figure supplement 2A-C) in *w^1118^* adult flies. Chinmo is not abundantly expressed in adult flies and hence, we were not so confident about immunostaining to detect Chinmo protein levels in the previously submitted version. However, we have standardized the protocol and the data is presented in the revised version of the manuscript.

15) Why is dilp6 being monitored? The authors do not discuss the rationale for examining dilp6 or foxo; they need to tell the reader why they are doing this experiment.

We agree with Reviewer 3 that a rationale should be provided for examining dilp6 and foxo. We examined *dilp6* and *foxo* because the mRNAs of these genes are induced upon nutrient shortage/starvation (Slaidina et al., 2009; He et al., 2020). Thus, we utilized dilp6 and foxo mRNA as positive controls for the starvation state and to compare the magnitude of increase of these mRNAs with that of *chinmo* mRNA upon starvation. The *foxo* and *dilp6* data have been removed from the revised manuscript to retain the focus on *chinmo* expression.

16) In panel H, what do the third and fourth pair of bars represent? In other words, why are + head and -head being shown twice?

We thank the reviewer for pointing out the error. The panel has been removed and only one time-point is represented in Figure of the revised manuscript.

17) In panel J, why is starvation (absence of protein) introduce here? What is the rationale for this experiment and why is it being shown?

Starvation is a form of nutrient deprivation that recapitulates the non-feeding states during development. Since *chinmo* is highly expressed during development, we wanted to test whether transcriptional upregulation of *chinmo* occurs during starvation as well. There are other regulators such as dSir2 and insulin-like peptide mRNAs that are regulated both by dietary restriction and starvation and hence, we wanted to examine tissue-specific regulation of *chinmo* upon nutrient deprivation. We have included additional data to support our preliminary analysis. *Chinmo* mRNA and protein are transcriptionally and post-translationally regulated upon starvation. In the revised manuscript we show that dSir2 deacetylates Chinmo and facilitates its distribution to the cytoplasm upon starvation. This nutrient-dependent post-translational modification of Chinmo may also play a role in relieving the repression of Chinmo targets. This panel is now represented as Figure 4—figure supplement 2I in the revised manuscript.

18) Figure 3A: DaGS>UAS-Chinmo flies with RU-486 all day within ~12 days of adulthood, much earlier ectopic flies without drug treatment. These data indicate that there is toxicity/lethality associated with ectopic chinmo, which obviates any possible connection to lifespan extension.

We agree that overexpression of *chinmo* ubiquitously using DaGS in adult tissues causes lethality. These data further support our model that silencing of *chinmo* by *miR-125* is critical for DR. Our model proposes that repression of *chinmo* in the brain and fat tissue is required for DR mediated lifespan extension and hence, we would expect that increasing mRNA or protein levels of this dosage-sensitive factor would be deleterious. We believe that *chinmo* is an example in support of the Antagonistic pleiotropy hypothesis that proposes that a gene that controls multiple traits, where at least one of the traits is beneficial to the organism’s fitness early on in life and at least one is detrimental to the organism’s fitness later-on (during aging). Hence, multiple mechanisms (transcriptional, post-transcriptional, and post-translational) exist to regulate the levels of this key temporal regulator. In our opinion, these data do not in any way obviate a connection of the *miR125-chinmo* axis to lifespan extension. Since, the manuscript is focused on examining the *miR-125-chinmo* pathway in the brain and fat tissue, we have removed the *DaGS>UASchinmo* lifespan data as most of the supporting data to show the effect on TAG levels is in *3X ElavGS >UAS chinmo* and *FBGS >UAS chinmo*, we have retained these panels only.

19) Figure 4 – please supply monochrome (white on black) all of these. It is very difficult for the human brain to appreciate green on black.

We have edited all figures with images to monochrome and retained the colors only in the merged/overlay images.

20) The authors need to better explain the various let-7-C reagents that they use both in the legend to Fig, 1 and in the Materials and Method. In particular, the let-C-7GKI allele is not on Flybase and I had to look up three prior pages to figure this out. Please make the manuscript easier to read and to be appreciated by the reviewer.

The *let-7-C* reagents that are used in Figures 1 and 2 are described in the results, material methods, and figure legends of the revised manuscript. Additionally, we have included the genetic scheme for the generation of the strains in Figure 1—figure supplement 1. The description of the let-7-C^GKI^ allele is included in the methods section. We have tried our best to make the manuscript easier to read.